# Fabrication of *p*-type 2D single-crystalline transistor arrays with Fermi-level-tuned van der Waals semimetal electrodes

Seunguk Song [1,2,7], Aram Yoon[1,3,7], Sora Jang[1,7], Jason Lynch[2], Jihoon Yang[1], Juwon Han[1], Myeonggi Choe[1,3], Young Ho Jin[1], Cindy Yueli Chen[4], Yeryun Cheon [5], Jinsung Kwak[1,6], Changwook Jeong[1], Hyeonsik Cheong [5], Deep Jariwala [2], Zonghoon Lee [1,3] ✉ & Soon-Yong Kwon [1] ✉

High-performance *p*-type two-dimensional (2D) transistors are fundamental for 2D nanoelectronics. However, the lack of a reliable method for creating high-quality, large-scale *p*-type 2D semiconductors and a suitable metallization process represents important challenges that need to be addressed for future developments of the field. Here, we report the fabrication of scalable *p*-type 2D single-crystalline 2H-MoTe$_2$ transistor arrays with Fermi-level-tuned 1T'-phase semimetal contact electrodes. By transforming polycrystalline 1T'-MoTe$_2$ to 2H polymorph via abnormal grain growth, we fabricated 4-inch 2H-MoTe$_2$ wafers with ultra-large single-crystalline domains and spatially-controlled single-crystalline arrays at a low temperature (~500 °C). Furthermore, we demonstrate on-chip transistors by lithographic patterning and layer-by-layer integration of 1T' semimetals and 2H semiconductors. Work function modulation of 1T'-MoTe$_2$ electrodes was achieved by depositing 3D metal (Au) pads, resulting in minimal contact resistance (~0.7 kΩ·μm) and near-zero Schottky barrier height (~14 meV) of the junction interface, and leading to high on-state current (~7.8 μA/μm) and on/off current ratio (~10$^5$) in the 2H-MoTe$_2$ transistors.

Extensive research has been conducted on novel transistor nanomaterials, such as two-dimensional (2D) van der Waals (vdW) semiconductors[1-3]. However, most 2D semiconductors have *n*-type, or in rare cases, ambipolar characteristics, and the scarcity of unipolar *p*-type 2D semiconductors severely limits the extensive use of 2D electronics in complementary metal-oxide-semiconductor (CMOS) inverters for energy-efficient circuits with a low power-delay product per bit[3]. Exploring 2D semiconductors with high electrical performance,

which may be grown at low temperatures (550 °C or less)[4] or easily transferable onto desired substrates, is critical for potential device fabrication with a limited thermal budget that can target CMOS back-end-of-line (BEOL) compatible process. Because of its low valence band maximum ($E_{VBM}$ ≈ 4.9–5.1 eV; the smallest among group-VI transition metal dichalcogenides (TMDs)[5]), 2D 2H-phase molybdenum ditelluride (MoTe$_2$) has been proposed as a promising unipolar *p*-type semiconductor with a suppressed electron transport compared to

[1]Department of Materials Science and Engineering & Graduate School of Semiconductor Materials and Devices Engineering, Ulsan National Institute of Science and Technology (UNIST), Ulsan 44919, Republic of Korea. [2]Department of Electrical and Systems Engineering, University of Pennsylvania, Philadelphia, PA 19104, US. [3]Center for Multidimensional Carbon Materials (CMCM), Institute for Basic Science (IBS), Ulsan 44919, Republic of Korea. [4]Department of Chemistry, University of Pennsylvania, Philadelphia, PA 19104, US. [5]Department of Physics, Sogang University, Seoul 04107, Republic of Korea. [6]Department of Physics, Changwon National University, Changwon 51140, Republic of Korea. [7]These authors contributed equally: Seunguk Song, Aram Yoon, Sora Jang. ✉e-mail: zhlee@unist.ac.kr; sykwon@unist.ac.kr

other TMDs. Hence, the development of a reliable and scalable method to synthesize a few-layer, high-quality p-type 2H-MoTe$_2$ is expected to enable next-generation 2D electronics for both front and back-end applications.

However, chemical vapor deposition (CVD) growth of the single-crystalline 2H-MoTe$_2$ polymorph with 100% coverage on wafer scales is challenging[6,7] because it only forms within a narrow growth window, and the uncontrolled Te flux (with low equilibrium vapor pressure) during growth hinders polymorphic control between the metallic 1T' and semiconducting 2H structure[8–10] owing to their small free energy difference (~35 meV per formula unit)[9]. The partial 1T' residual products produced by CVD across the 2H-MoTe$_2$ surface[11,12] as well as oxygen-related impurities[13,14], Te vacancies[11,15], and grain boundaries (GBs)[1,16,17] could lead to an increase in sheet resistance and inadequate functioning of electronic devices, resulting in the degradation of electrical properties[12–14]. Recent pioneering studies have been made to synthesize 2H-MoTe$_2$ through solid-to-solid phase transitions[18,19] or 2D seed growth methods[20]. However, the practical application of these methods is limited because of the use of powder-based horizontal CVD[18,19] or the need for an AlO$_x$ passivation layer[20], which makes the production of uniform, high-quality 2H-phase thin films on a large scale with high throughput difficult. Moreover, the electrical conductance of 2H-MoTe$_2$ synthesized using these methods[18–20] has been restricted (e.g., on-state sheet conductance less than ~10 μS and on-to-off current ratio of ~10$^4$; Supplementary Table 1), indicating a need for further exploration of novel growth techniques.

In terms of the fabrication of electrical contacts, the formation of interfacial defects such as vacancies, glassy layers, and alloys[1] at 2H-MoTe$_2$-based heterojunctions may increase during 3D metallization due to the small electronegativity differences between Mo and Te, resulting in weaker bonds[6,7]. Because such defects generate new interfacial states while pinning the Fermi level ($E_F$) to a specific energy level, the contact properties or Schottky barrier height (SBH) cannot be effectively modulated by selecting metals with different work functions (WFs). In this regard, the reported p-type transport performances of 2H-MoTe$_2$ field-effect transistors (FETs) was primarily limited by their on-to-off current ratios ($I_{on}/I_{off} < 10^4$), on-state currents ($I_{on} < 1$ μA/μm at $V_{ds} = -1$ V), and field-effect hole mobilities ($\mu_h < 10$ cm$^2$V$^{-1}$s$^{-1}$), even when using high-WF 3D metal contact electrodes, such as Pd[18,21–23] and Pt[24]. Alternatively, vdW integration of 2D polymorphic metallic electrodes, that is, 1T'-phase MoTe$_2$, may provide an ultrasharp and pristine interface at the vertical 2D/2D metal-semiconductor junction (MSJ). However, no systematic studies on the impact of 3D metallization on 2D metals (as opposed to 2D semiconductors) in 2D/2D MSJs have been conducted, despite the fact that 3D contact pads are required for all 2D devices. This is particularly important because the use of 2D metals as an electronic component in all-2D circuits become more frequent (e.g., graphene[25,26], and VSe$_2$[27], NbSe$_2$[28] as contact electrodes for WSe$_2$ transistors); however, the 2D semimetal WF tuned by 3D metal is often overlooked. Moreover, the vast majority of vdW-integrated 2D metals have been fabricated with mechanically exfoliated[29–31] or CVD-grown irregular flakes[32,33], which are impractical for high-yield manufacturing. Furthermore, significant differences in reported performance metrics[34,35] or FET switching polarities[10], raise questions about the reproducibility of the polymorphic junction using 1T'-MoTe$_2$[10,18,20,34–36].

In this study, we report the fabrication of high-performance p-type FET arrays consisting of single-crystal 2D semiconductors and Fermi-level-tuned vdW contact electrodes using a 4-inch-scale phase-controlled growth technique. When subjected to heat treatment with a uniform vertical Te flux, the solid-phase 2H-MoTe$_2$ domain grew more rapidly in a matrix of finer 1T'-MoTe$_2$ poly-grains. This produces a large-area of 2H-MoTe$_2$ thin film with ultra-large single-crystalline domains. In addition, the introduced seed growth mode facilitates the lateral abnormal and epitaxial grain growth of 2H single-crystal conformal patterns with controlled nucleation

sites at a low temperature of ~500 °C, even on an amorphous SiO$_2$ substrate. Because of the large-area control over polymorph growth, fabrication of on-chip arrays of 1T'/2H MoTe$_2$ MSJs was easily accomplished using standard photolithographic patterning techniques and layer-by-layer assembly. Through Fermi-level tuning of 1T'-MoTe$_2$ semimetal using a 3D metal (Au), we fabricated and characterized p-type MoTe$_2$ FET arrays that offer contact resistances of ~0.7 kΩ·μm, near-zero SBHs of ~14 meV, $I_{on}/I_{off}$ ratios that exceed ~10$^5$ and $\mu_h$ of ~29.5 cm$^2$V$^{-1}$s$^{-1}$. The combinations of the low-resistance 2H-MoTe$_2$ single crystals and defect-free contact interface permitted the highly efficient hole transport in FET arrays with high $I_{on}$ values approaching ~7.8 μA·μm$^{-1}$, which outperformed most of the chemically synthesized p-type 2D semiconductors.

## Results

### Phase-engineered synthesis of MoTe$_2$

The schematic of a bottom-up approach for obtaining MoTe$_2$ is shown in Fig. 1a. The synthesis was carried out in a Te-confined reactor containing the face-to-face stacked precursors of Mo and the eutectic alloy of Ni$_x$Te$_y$, with the Ni$_x$Te$_y$ as the Te source[17]. The Te vapor constantly evaporated from the eutectic alloy and was confined at a tiny gap between the precursor films, assisting the entire tellurization process in forming MoTe$_2$. Additionally, the growth method enabled uniform nucleation of MoTe$_2$ flakes owing to the homogeneous evaporation of Te from the surface, which was not possible via powder-based horizontal CVD[2]. As a result, a large-area MoTe$_2$ thin film with the 1T'-phase was obtained directly on SiO$_2$/Si at a growth temperature ($T$) of 500 °C, as shown in the optical microscopy (OM) images (Fig. 1b and Supplementary Fig. 1). A higher growth temperature (e.g., $T > 600$ °C) results in the in-situ phase conversion of 1T' to 2H during the reaction, as indicated by the circular shapes of the 2H crystal in Fig. 1c. The domain of the 2H phase expanded as the growth temperature ($T$) and time ($t$) increased. For instance, the 2H phase was able to expand up to ~300–1000 μm for $T = 700$ °C and $t = 30$ min, and a fully converted 2H phase thin film (~1 × 1 cm$^2$) was achieved at $t = 75$ min with the same $T$ (i.e., a high production rate of ~50.5 mm/h) (Supplementary Fig. 2). Meanwhile, the defective 2H phase starts to form at $T > 750$ °C (Supplementary Fig. 2l–o). The polymorphic phase diagram in Fig. 1d was successfully extracted, which illustrates the fraction of the 2H phase covered by the thin film as a function of $T$ and $t$. By utilizing this information, we were able to obtain the first 4-inch wafer-scale of MoTe$_2$ with a completely covered 2H or 1T' phase (Fig. 1e). Comparisons of the production rates of the resulting films with those reported suggest the substantial advantage of our approach toward mass production over other studies[18–22,35,37] (Supplementary Fig. 3 and Supplementary Table 1).

The Raman spectra of the MoTe$_2$ grown at different $T$ revealed representative signals from each phase (Fig. 1f), which were consistent with reports published previously[9,36]. The X-ray photoelectron spectroscopy (XPS) images (Fig. 1g and Supplementary Fig. 4) further confirm the successful phase transition as indicated by the shift in binding energies[10] as well as the near-ideal stoichiometry of the crystal [at.%(Te/Mo) ≈ 2.0 and 1.9 for 1T' and 2H phases, respectively] devoid of oxidation-related peaks (i.e., Te-O or Mo-O bonds). The atomic force microscopy (AFM) images (insets in Fig. 1e) demonstrated that the thin films were uniform in thickness ($H$) for both phases, with roughness ($R_a$) values less than the interlayer distances (<0.9 nm). Notably, the $H$ of the MoTe$_2$ was precisely controlled by the $H$ of the Mo precursor, and films as thin as ~3.5 nm (~4 layers) were obtained (Fig. 1h and Supplementary Fig. 5). Regardless of the 2H or 1T' phase, the $H$ was identical once the same precursor was used for the growth, as evident from the homogenous morphology (Supplementary Fig. 5). In addition, when MoO$_x$ was applied as a metal precursor instead of Mo, our growth technique could be extended to construct thinner atomic layers of MoTe$_2$ (~1.6 nm; bilayers) on a large scale (~20 mm) without notable micro-voids or impurities (Supplementary Fig. 6).

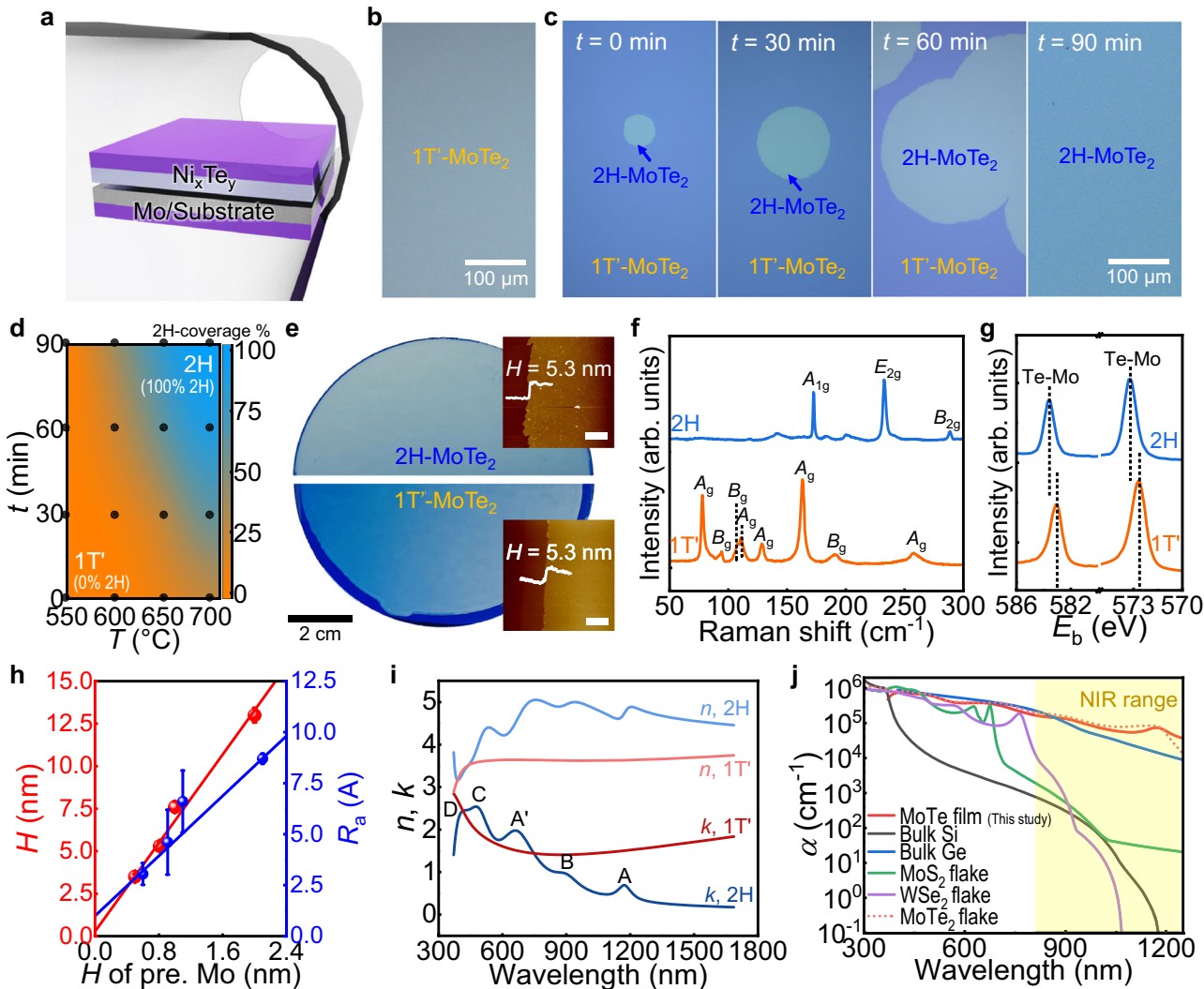

**Fig. 1 | In situ control of MoTe₂ polymorphs during wafer-scale synthesis.**
**a** Schematic of the Te-gas confined reactor consisting of $Ni_xTe_y$ on top of a Mo precursor film within the furnace. **b**, **c** Optical microscopy (OM) images of $MoTe_2$ at the growth temperatures of (**b**) $T = 500\,°C$ for the growth time, $t = 20$ min and (**c**) $T = 700\,°C$. Rounded shapes in (**c**) indicate the 2H-phase $MoTe_2$. **d** Polymorphic phase diagram of $MoTe_2$ as a function of $T$ and $t$, determined by the difference of optical contrast in the OM images (Supplementary Fig. 2). Linear interpolation was performed for the contour plot using data points (bullets). **e** Optical image of the synthesized 1T'- and 2H-phase $MoTe_2$ on a 4-inch wafer, with atomic force microscopy (AFM) images showing their thickness ($H$) (scale bar: 2 μm). **f** Raman spectra of the resultant 2H- (blue) and 1T'-$MoTe_2$ (orange). **g** X-ray photoelectron

spectroscopy (XPS) spectra of the Te 3$d$ level for 2H- (blue) and 1T'-$MoTe_2$ (orange) grown at $T = 700\,°C$ for $t = 30$ min and $T = 500\,°C$ for $t = 10$ min, respectively. Dashed lines indicate Te-Mo binding energies ($E_b$) of $MoTe_2$. **h** Summary of the $H$ and surface roughness ($R_a$) of $MoTe_2$, depending on the $H$ of the pre-deposited Mo precursor ($H$ of pre. Mo). The average and standard deviation of five areas (~10 × 10 μm²) are represented as data points ± error bars. **i** Refractive index ($n$; lighter color) and extinction coefficient ($k$; darker color) values of 1T'- (red) and 2H-phase $MoTe_2$ (blue) obtained *via* ellipsometer measurements. **j** Absorption coefficient ($α$) of our $MoTe_2$ film in the broad wavelength range (~300–1300 nm) compared with bulk semiconductors (e.g., Si[76] and Ge[77]) and 2D transition-metal dichalcogenides (e.g., $MoS_2$, $MoSe_2$, and $MoTe_2$ flakes)[50], demonstrating its higher absorption efficiency.

Polymorphic structural deformation occurs in tandem with band structure variation, as indicated by the different optical contrasts between two phases in OM images (Fig. 1e). Investigations using ultraviolet photoelectron spectroscopy (UPS; in Supplementary Fig. 7) enabled the estimation of band structures of each phase. The 1T'-$MoTe_2$ was found to be a gap-less semimetal with a WF of ~4.51 eV, which was comparable to the value computed using the density functional theory (DFT)[38]. For the 2H-phase $MoTe_2$, the calculated $E_F$ was higher than the mid-gap (i.e., $E_F - E_{VBM} \approx 0.57$ eV) value, implying that as-synthesized crystal was not doped as *p*-type, which is consistent with earlier reports on mechanically exfoliated crystals (WF ≈ 4.35–4.42 eV)[39,40] or DFT simulations (WF ≈ 4.29 eV)[41]. For the comparison, we summarized the WF and ($E_F - E_{VBM}$) for $MoTe_2$ with the values obtained from literatures[18,19,21,22,35,41–43] (Supplementary Table 2). This trivial shift of $E_F$ to $E_{VBM}$ is advantageous for achieving its intrinsic

carrier transport[44,45], which is exceptional because the chemically synthesized $MoTe_2$ generally possesses a large WF (~4.85 eV)[42] or small $E_F - E_{VBM}$ (~0.15 eV)[43], most likely attributed to its inevitable oxide-related defects produced during the growth[13,14].

Furthermore, the complex refractive index ($n + ik$) of 2H-$MoTe_2$ (Fig. 1i) revealed multiple excitonic band transitions (labeled A to D). This feature is absent in 1T'-$MoTe_2$ because of its semi-metallic properties. The optical bandgap ($E_g$) of the as-prepared 2H-$MoTe_2$, extracted by the optical absorbance, was ~0.89 eV (Supplementary Fig. 7c), which was in good agreement with the values in the literatures[46,47]. We discovered that the significant difference between the refractive indices ($n$) of the two polymorphs ($\Delta n > 1$) was comparable to that of the well-investigated phase change material Ge-Sb-Te[48]. This suggests its potential use as a phase-change material in 2D photonics (Supplementary Fig. 7e). In addition, the $n$ value of 2H-$MoTe_2$ (~4.45) was

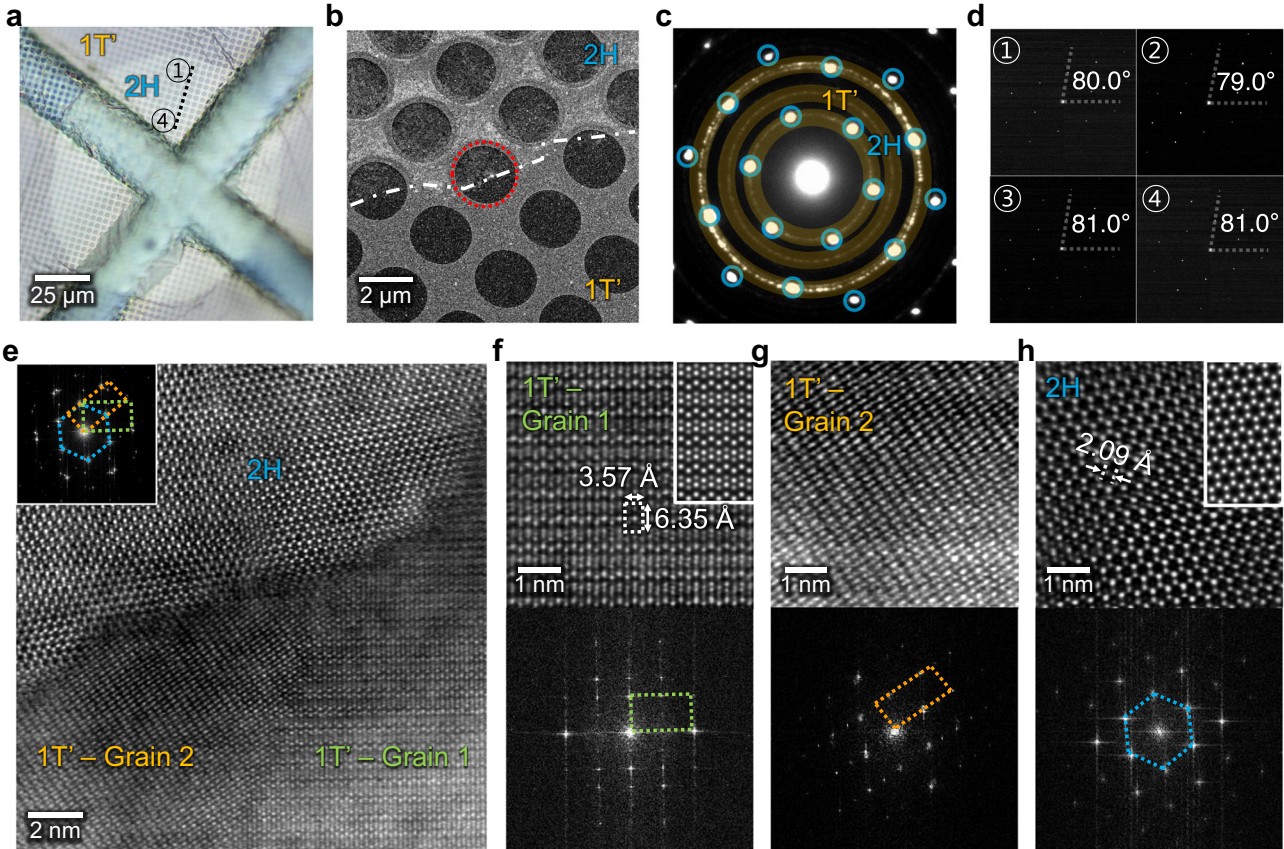

**Fig. 2 | Transmission electron microscopy (TEM) analysis of the abnormal grain growth of MoTe₂ accompanied by the phase transition. a** Optical microscopy image and **b** low-magnification scanning-TEM (STEM) image of the MoTe₂ homojunction transferred to the TEM grid, where the two polymorphs are distinguished by the contrast. **c** Selected area electron diffraction (SAED) pattern captured at the interface between 2H- (blue circles) and 1T′-MoTe₂ (yellow rings). Three-fold diffraction spots of 2H-MoTe₂ suggest that the corresponding structure is single-crystalline. In contrast, 1T′-MoTe₂ is polycrystalline, as demonstrated by the ring shape of the patterns. **d** SAED patterns of the 2H-MoTe₂ measured at the marked

regions in (**a**). **e** High-resolution STEM image of the homojunction interface (marked as a red circle in (**b**)). Inset shows the corresponding fast-Fourier-transform (FFT) pattern, showing the three distinct lattices of MoTe₂ (orange and green rectangles for the 1T′ and blue hexagon for the 2H structure). **f**–**h** (top) Atomic-resolution STEM images of the different grains in (**e**). The inset on the right shows the computed atomic images for corresponding structures. (bottom) Corresponding FFT patterns of the top images represent the different structures of each grain. The orientation of the three grains was random (Supplementary Fig. 9).

considerably below the bandgap where the loss was low ($k = 0$). This sub-gap index was higher than those of 3D semiconductors with comparable bandgaps ($E_g \approx 0.72–1.34$ eV) such as Si and III-V compounds (i.e., GaSb and InP) ($n \approx 3.5–3.7$)[49]. The absorption coefficient of the as-grown MoTe₂ ($\alpha > 4 \times 10^4$) was also higher than those of the other group-VI TMDs ($\alpha \approx 10^0–10^4$)[50] in the near-infrared spectral range ($\lambda > 800$ nm) (Fig. 1j). Given the tunability of 2D thin films to external stimuli such as electrostatic gating and doping[51], the high $n$ and $\alpha$ values in 2H-MoTe₂ show great promise as a highly tunable, electro-optic material and optical communication device (e.g., saturable absorbers, modulators, and photodetectors) in the telecommunications regime (~1.3–1.5 μm) (Supplementary Fig. 7).

**Abnormal grain growth of MoTe₂ with its phase transition**
We investigated the as-synthesized in-plane 1T′−2H MoTe₂ heterostructure using transmission electron microscopy (TEM), as shown in Fig. 2 (and Supplementary Figs. 8 and 9). The single circular domain of the 2H phase is a single crystal, as demonstrated in the OM (Fig. 2a) and low-magnification scanning TEM (STEM) images (Fig. 2b). Conversely, the 1T′-MoTe₂ demonstrated a polycrystalline nature. For instance, the selected-area electron diffraction (SAED) pattern measured at the heterostructure (red mark in Fig. 2b) revealed the three-fold symmetry of the planes of the 2H phase (blue circles in Fig. 2c) and the ring

shapes representing the multiple grains of the 1T′ structure (yellow circles in Fig. 2c). Additionally, the SAED patterns captured in the various regions indicated in Fig. 2a (positions 1–4) suggest the presence of similar orientated planes with the three-fold symmetry (Fig. 2d), implying that the 2H domain is a single crystal spanning a large area (>100 μm). Conversely, the 1T′ structure possessed a small grain size (<100 nm, Supplementary Fig. 8a). The MoTe₂ exhibits an excellent atomic structure, as demonstrated by the atomic-resolution STEM images of each phase (Fig. 2e–h). The measured lattice spacings of 3.57 and 6.35 Å also correspond well to the predicted values for the (010) and (001) planes of the 1T′ structure (Fig. 2f, g), respectively. Additionally, the 2H phase demonstrated the Mo-Te lattice distance of ~2.09 Å in Fig. 2h (Mo-Mo distance of 3.47 Å in Supplementary Fig. 8g, h), which is consistent with a previous study[36].

The grain size difference between the polymorphs is greater than 1000 times, indicating that the phase transition from 1T′ to 2H occurred via recrystallization and atomic rearrangement during high-temperature growth. For instance, few of the junctions between 1T′ and 2H crystals shared an atomic orientation with the [2-1-10]₂H// [020]₁T′ interfaces (Supplementary Fig. 9a), which may be the energetically most favorable interface for the phase transition. However, not all in-plane 1T′−2H junctions exhibited similar orientations (as depicted in Fig. 2e and Supplementary Fig. 9b, c). This suggests that

our growth mode may assist in overcoming the energy barrier to arrange the random crystallographic orientation of polycrystals, resulting in a grain growth for 2H-MoTe$_2$. Characteristically, this process is referred to as "abnormal grain growth", as opposed to "ordinary grain growth", in which the formation of a single crystal results in a relatively uniform grain size distribution[52].

According to previous reports on MoTe$_2$ synthesis[8–10], the amount of Te largely dictates the MoTe$_2$ phase during growth. When a large amount of Te is present, the 2H phase is thermodynamically stable. In contrast, an insufficient Te encourages the formation of non-stoichiometric MoTe$_{2-x}$, triggering the formation of the metastable 1T' polycrystalline structure[9,53]. In this study, the Ni$_x$Te$_y$ source was located in the hot zone of the furnace, where the vapour pressure of Te at $T$ = 500 °C (~0.77 Torr) was ~18 times lower than that at $T$ = 650 °C (~13.88 Torr)[54]. Thus, the early synthesis phase (i.e., low-temperature growth) may result in a Te-deficient environment to form the 1T' structure. The formation of 1T'-MoTe$_2$ may be energetically favorable owing to the existence of a considerable strain induced by the sudden volume expansion from the Mo precursor (~15.6 Å$^3$) to MoTe$_2$ (~74.9 Å$^3$)[37], as predicted by the DFT calculations of the strain effect[37,55]. In our 1T' structure thin film, the micro-strain reached ~2.22 × 10$^{-3}$, as determined by the Williamson–Hall method for X-ray diffraction (XRD) patterns (Supplementary Fig. 10).

As the $T$ and $t$ increase, the Te enables the phase transition and nucleation of the 2H-phase MoTe$_2$, which consumes the small 1T'-phase sub-grains and grows rapidly because the 2H phase is energetically more favorable. In recrystallization theory[56], the migration of defects (e.g., GBs, dislocations, and vacancies) in a material can relieve the internal strain energy by producing large strain-free grains. Comparably, under Te-rich conditions, the Te atoms may adsorb and desorb onto the film, promoting the migration of GBs and vacancies while reducing the tensile strain via grain growth. Because the strain state determines the equilibrium phase of MoTe$_2$[37,55], the grain growth simultaneously transformed the 1T' to 2H phase; hence, the micro-strain of the thin film consisting of single-crystal 2H MoTe$_2$ could undergo a three-fold decrease (~0.65 × 10$^{-3}$) compared with that of 1T'-MoTe$_2$ polycrystals (~2.22 × 10$^{-3}$; Supplementary Fig. 10). Furthermore, the produced 2H nucleus distributed along the 1T' matrix facilitated the abnormal grain growth of 2H domains, allowing the production of fully converted 2H single crystals with enormous domain sizes over a large area as the growth time increased. The Te-confined growth mode used in this study also enabled homogeneous growth of the 2H-phase and promoted abnormal grain growth across the wafer by providing a uniform and substantial flow of Te vapor to the Mo precursor (Supplementary Fig. 11). This is in contrast to the horizontally supplied, non-uniform Te sources using powder-based horizontal CVD[18,19], which inevitably led to a compositional gradient across the substrate, thus limiting the growth scalability.

## Low-temperature, position-controlled seed growth of MoTe$_2$

The nucleation of abnormal grains is made possible by using a seed crystal of 2H-MoTe$_2$; the process is shown in Fig. 3a. We deposited "2H-seed" single crystals onto pre-synthesized polycrystalline 1T'-MoTe$_2$ film, which was then annealed in Te-rich atmosphere using the Ni$_x$Te$_y$ stack (the same Te source for our study in Fig. 1). The phase transition from 1T'-MoTe$_2$ to 2H occurred at $T$ = 500 °C, as optically distinguishable shapes formed along the square-like seed patterns in the OM image (Fig. 3b, c). The corresponding Raman spectra validated the phase transition (Fig. 3d). Notably, 2H-phase formation via seed growth was possible at $T$ = 500 °C, which is below the minimum temperature required for 2H crystal nucleation (>550 °C) in previous synthesis experiments (Fig. 1d). With the exception of the transitioned-2H areas surrounding the 2H-seed patterns, random 2H phase nucleation was inhibited, especially compared with the seed growth at $T$ = 700 °C (Supplementary Fig. 12a). This suggests that the

low-temperature (<500 °C) seed growth mode can be employed for the position-controllable synthesis of 2H, 2D chalcogenide semiconductors by expediting the 1T'-to-2H phase transition.

The STEM analysis proved that the newly achieved 2H structure is a single crystal for all the distinguished domains in the OM image (Fig. 3c), produced by the lateral grain growth of the 2H-MoTe$_2$ seed layers ("2H-seed") (Fig. 3f–h). The SAED pattern captured in the transitioned 2H region ("tr-2H") showed distinct three-fold symmetric plane sets with a well-specified alignment with respect to those of the seeded single crystal, whereas that of the 1T' structure had the ring-shaped feature for polycrystals (Fig. 3f). The atomic-resolution STEM images for 2H structures also showed the same crystallographic orientation between the 2H-seed and transitioned regions, indicating its solid-phase epitaxial and abnormal grain growth (Fig. 3g, h; note that the brighter intensity in the 2H-seed area is attributed to its higher thickness). In contrast, the 1T'/tr-2H interface revealed a random orientation between them (Supplementary Fig. 12b), similar to the 1T'-to-2H phase transition (Fig. 2e).

We evaluated the effects of different carrier gases (i.e., H$_2$ and mixed H$_2$/Ar) and Te precursors (i.e., Te powder or Ni$_x$Te$_y$ stack layer) on abnormal 2H-grain growth by measuring the dimensions of the tr-2H attributed to the controlled 2H nucleation by seed growth (Fig. 3e and Supplementary Fig. 12c–f). Heat treatment without a Te source or with Te powder did not induce the 1T'-MoTe$_2$-to-2H phase transition (Supplementary Fig. 12c, d); the opposite is true when the Te-confined reactor, Ni$_x$Te$_y$ (Supplementary Figs. 12e, f), was used. This implies the importance of a sufficient Te concentration for the abnormal grain growth of the 2H phase. The use of H$_2$ as a carrier gas resulted in a larger converted 2H area (~5.20 ± 0.46 μm) compared with that annealed under only Ar gas (~1.84 ± 0.15 μm; averaged for 10 different patterns of the "tr-2H" region in Fig. 3e and Supplementary Fig. 12e, f). Because the H$_2$ gas reacted with Te to form the thermodynamically favorable gas-phase H$_2$Te[57], the H$_2$ may act as an efficient Te carrier. The EDS analysis of the 2H-MoTe$_2$ sample prepared by rapid cooling at 500 °C revealed that its Te content was slightly higher than its ideal stoichiometry (i.e., at.%(Te/Mo) = 2.10 ± 0.12). This also indicates that the Te adatom along the surface played an important role in GB migration (Supplementary Figs. 12g–i). Considering these findings, we conclude that the high Te concentration in our Te-gas confined system provided a substantial driving force for the abnormal grain growth of the 2H structure.

This technique can regulate both the domain size and crystallographic orientation of Te-based 2D semiconductors, which is a major challenge for conventional CVD methods on amorphous substrates. Each conformal pattern is single-crystalline by nature, which benefits electronic or optoelectronic devices that require high-quality single crystals after isolating individual crystals by etching away unnecessary areas. In particular, large-scale single-crystal films can be fabricated without GBs or with ultrasmall-angle GBs by using 2H seed patterns with the same crystal orientation (obtained from mechanically exfoliated single-crystalline flakes, see Supplementary Fig. 13). The results show that there is further potential for developing wafer-scale single-crystal films by "2H-seed" engineering using CVD process at low temperatures (550 °C or less), that is, centimeter-scale or larger grain sizes in the resulting 2D films can be achieved by producing 2H seed patterns with the same crystal orientation sufficiently separated from one another with high controllability although there might still be a 1D defect if there is a slight misalignment. Furthermore, the generation of 2H-MoTe$_2$ single crystals via seed growth at a low $T$ of ~500 °C ensures the compatibility of the process with modern silicon integrated circuits, in which the material preparation $T$ should not exceed 550 °C for CMOS BEOL fabrication[4]. Given that single-crystal semiconductors at relatively low $T$ are exceedingly challenging to obtain using conventional crystal growth methods, our seeded abnormal grain growth technique is a promising method for the production of high-quality 2D

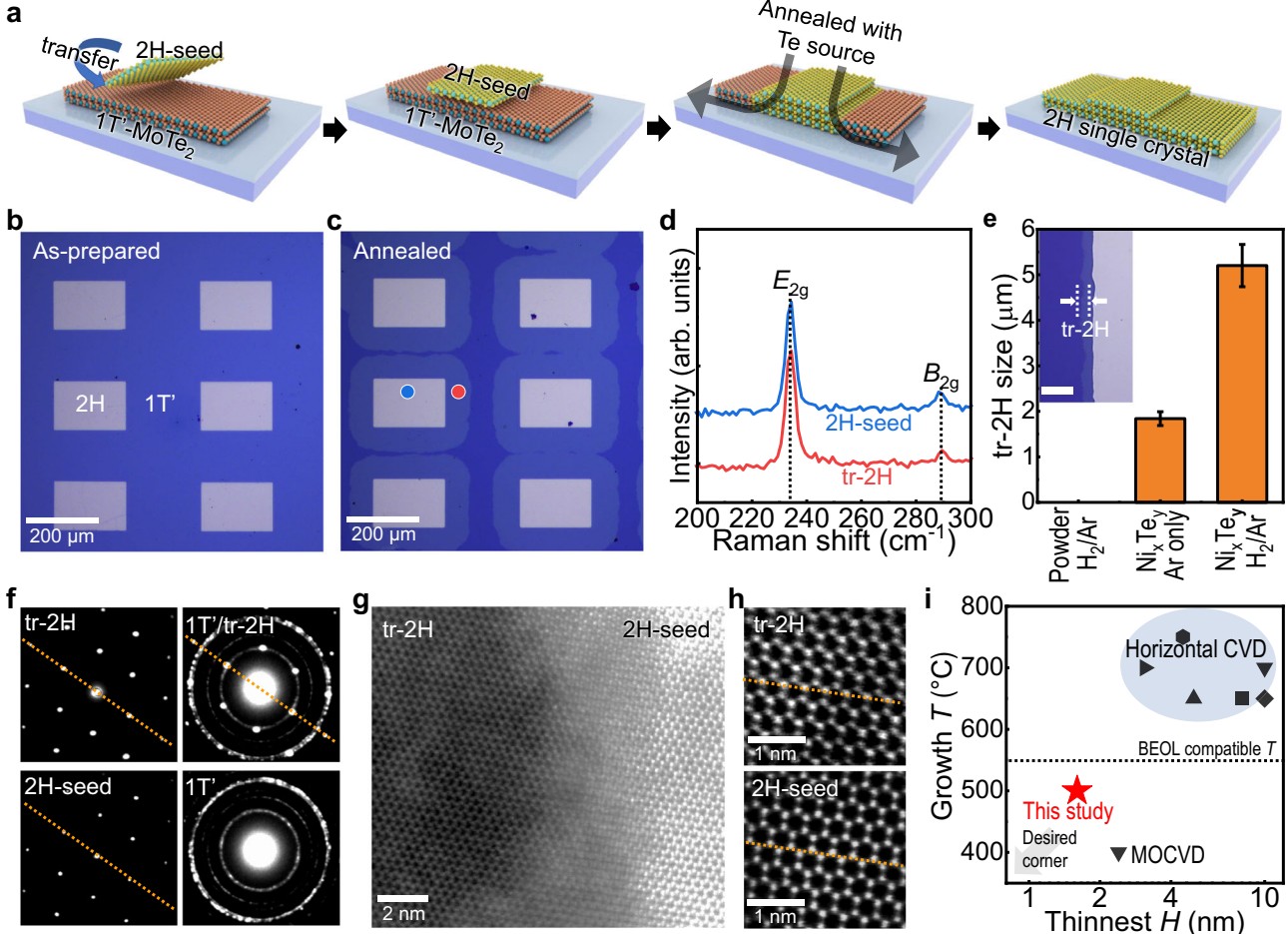

**Fig. 3 | Seed growth of 2H-MoTe₂ single crystal conducted at a low growth temperature ($T$) of ~500 °C. a** Schematic showing the abnormal grain growth of the 2H crystal from the seed. **b** Image of the transferred 2H-seed on top of the preformed 1T′ thin film. **c** The corresponding optical microscopy image of the seeded thin films after heating at 500 °C for 1 h using Te-gas confined reactor in Fig. 1a. **d** The corresponding Raman spectra of the seeded single crystal (blue, 2H-seed) and 1T′-to-2H transitioned (red, tr-2H) regions marked in (**c**). **e** Dimensional evolution of the tr-2H area (the distance between the dashed lines in the inset with a scale bar of 5 μm) in the sample heat-treated at 500 °C for 10 min, depending on the various ambient conditions, i.e., use of Te powder or $Ni_xTe_y$ stack as Te source and mixed H₂/Ar and only Ar as carrier gas. The error bars indicate the standard

deviations of 10 different measurements in each condition. **f–h** TEM analysis of the polymorphic regions in the sample prepared at $T$ = 500 °C, **f** SAED patterns captured from different regions in the film. The dash-dotted lines indicate that both 2H-seed and tr-2H regions have the same oriented planes. **g** Representative high-angle annular dark-field scanning-TEM (HAADF-STEM) image of the 2H-seed/tr-2H interface region, and **h** corresponding STEM images showing the comparable atomic structures of the 2H-seed and tr-2H with the same crystallographic orientations. **i** Lowest possible growth $T$ of 2H-MoTe₂ and the thickness of the resultant film. Previously reported values for large-scale thin films[18–22,35,37] are presented for comparison.

TMDs, particularly compared to previous approaches for synthesizing 2H-MoTe₂ thin films (Fig. 3i) (see Supplementary Fig. 3 and Supplementary Table 1 for more details).

### On-chip arrays of vdW-integrated *p*-type MoTe₂ transistors

Wafer-scale, polymorph-controlled synthesis allows the layer-by-layer integration of 1T′-semimetallic and 2H-semiconducting MoTe₂ for high-performance FET arrays. Figure 4a and Supplementary Fig. 14a schematically show the device fabrication process. The 3D metal (Au) deposited onto 1T′-MoTe₂ was used as the masking layer for the plasma etching of the exposed 2D metal to define the source/drain patterns. Then, the integrated 3D/2D metallic pattern layers were picked up and transferred to the single-crystal 2H-MoTe₂ using polymeric supporting layers (Supplementary Fig. 14a). Such 3D/2D metallic integration ensured a high-yield process because the 2D semimetal layer mechanically secured by the thicker 3D metal could relatively avoid the formation of macroscopic defects such as wrinkles and cracks during the transfer process. This resulted in the high-density fabrication of FET arrays on a centimeter-scale chip (Fig. 4b, c). Au was the

most excellent 3D contact pad for 1T′-MoTe₂ to ensure passivation of the underlying layers against oxidation during device fabrication. Further, Au helped enhance carrier transport by introducing more carriers to the semimetal while avoiding the formation of intermetallic $AuTe_x$ or non-stoichiometric $MoTe_{2-x}$ (Supplementary Fig. 15; the interfacial effect of 3D metal deposition on 1T′-MoTe₂ is further discussed in the following section). Compared with the recent research on vdW integration using mechanically exfoliated flakes for 2D/2D MSJ FETs[29,32], our study is more significant because two different synthetic 2D thin films were combined for FET arrays with a higher yield (e.g., ~490% increase in the number of FETs on a chip).

The cross-sectional STEM image shows an ultraclean, sharp interface of the fabricated 1T′/2H-MoTe₂ heterostructure (Fig. 4d and Supplementary Fig. 16). The high quality is further corroborated by the STEM-EDS analysis, wherein no signs of degradation of the MoTe₂ layers were observed following the process of fabrication (Supplementary Fig. 16). Additionally, our method of stacking polymorphic layers has a significant advantage over the conventional method of deposition for 3D metals in fabricating MoTe₂-based MSJs such as Ti

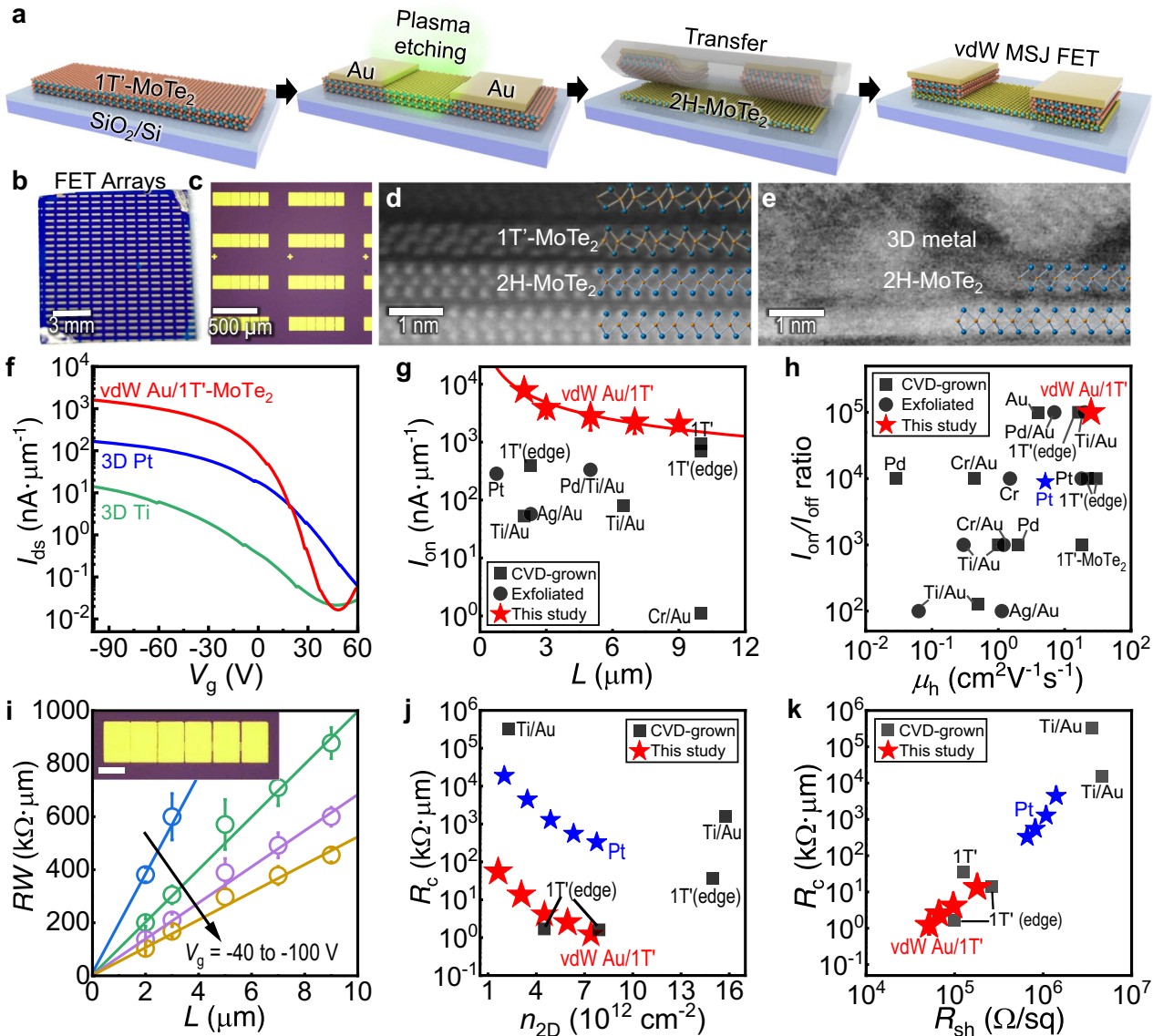

**Fig. 4 | Integration of 2H-MoTe₂ transistors with vdW Au/1T′-MoTe₂ contact electrodes. a** Schematic of the MoTe₂-based transistor fabrication. **b** Optical image of the field-effect transistor (FET) arrays on a ~1 × 1 cm² SiO₂/Si substrate. **c** Optical microscopy image of the fabricated FETs with transfer length method (TLM) patterns. **d, e** Cross-sectional scanning transmission electron microscope (STEM) images of the 2H-MoTe₂ transistors with (**d**) a 2D semimetal contact using 1T′-MoTe₂ and (**e**) conventional 3D metal contact electrodes. **f** Representative transfer characteristics (gate voltage ($V_g$) vs. drain current ($I_{ds}$)) of 2H-MoTe₂ FETs with different contacts, i.e., 1T′-MoTe₂ (red), Pt (blue), and Ti (green). **g, h** Benchmarking plots of the 2H-MoTe₂ with various contact electrodes, fabricated onto a bottom gate dielectric SiO₂ layer. Performance comparison include FETs with vertical contacts prepared by chemical vapor deposition (CVD)[18,20,21,34–36,42,59,64] (squares) or mechanical exfoliation[23,24,39,44,45,60] (circles), as well as FETs with lateral edge contacts using the 1T′ structure[18,20,34,36]. **g** Comparison of on-state current density ($I_{on}$) at a $V_{ds} = -1$ V depending on the channel length (L). **h** Comparison of the $I_{on}/I_{off}$ and the field-effect hole mobility ($\mu_h$) values. **i** TLM plots of the 1T′/2H-MoTe₂ junction FET with different $V_g$, showing the linear dependence of the total resistance ($RW$) on L. The error bars result from the averaging of at least five different TLMs. Inset represents an optical microscopy image of TLM patterns for the 2H-MoTe₂ FETs with vdW Au/1T′ contact electrodes (scale bar: 20 μm). **j** Contact resistance ($R_c$) of 1T′-MoTe₂- (red) and Pt- (blue) contacted 2H-MoTe₂ FETs, as a function of carrier concentration ($n_{2D}$) induced by applied $V_g$. **k** $R_c$ values of Au/1T′/2H-MoTe₂ (red) and Pt/2H-MoTe₂ FETs (blue), depending on $R_{sh}$. For comparison, those previously reported for 2H-MoTe₂ FETs[34–36,59,64,66,67] are indicated in (**j**) and (**k**).

and Pt (We specifically select the Pt as a contact electrode because Pt is widely known for its high WF of ~5.64 eV[58] so that the superior *p*-type transport is expected in the Pt-contacted MSJ, whereas Ti has the WF value of ~4.4 eV which is promising for *n*-type contact). The high-energy process for deposition and chemical interaction between 3D metals with 2D MoTe₂ can result in the formation of void-like defects, alloys, and glassy layers, as indicated by the STEM images (Fig. 4e) and the XPS characterizations (Supplementary Fig. 17).

Possibly affected by the disorder-free interface, the junction transistors with vdW Au/1T′-MoTe₂ contact electrodes demonstrated a higher switching performance than those composed of 3D metals in

transfer curves ($I_{ds}$–$V_g$) (Fig. 4f). The superior linearity of output characteristics ($I_{ds}$–$V_{ds}$) also indicated that the *p*-type ohmic contact was achieved by the vdW Au/1T′-MoTe₂ electrodes (Supplementary Fig. 14b). By contrast, the use of 3D metals (Pt and Ti) as contact electrodes resulted in nonlinear output characteristics, suggesting the existence of a substantial contact barrier at the 3D/2D MSJ interfaces (Supplementary Fig. 14c, d). More than 50 transistors with vdW-integrated 1T′/2H-MoTe₂ MSJs on a large-area chip (>1 × 1 cm²), which exhibited comparable and adequate *p*-type dominant transport behavior at room temperature, were measured. Notably, the transistors with a channel length (L) of ~5–9 μm demonstrated an

exceptionally high device performance, that is, high on-to-off current ratios ($I_{on}/I_{off}$ ratio = (1.3 ± 0.6) × $10^5$ on average), on-state current density ($I_{on}$ = 2.00 ± 0.92 µA·µm$^{-1}$ at $V_{ds}$ = −1 V), and field-effect hole mobilities ($\mu_h$ = 21.0 ± 3.3 cm$^2$V$^{-1}$s$^{-1}$) in the p-type 2H-MoTe$_2$ transistors. Supplementary Fig. 14e–g depict the histograms and distribution fits of the statistical data obtained from the FET arrays, suggesting the reproducibility of the vdW junction transistors. Compared with the devices in previous studies that used 2H-MoTe$_2$ as a p-type semiconductor to fabricate transistors[18,20,21,23,24,34–36,39,42,44,45,59–64], our devices exhibited substantially high $I_{on}$ (at $V_{ds}$ = −1 V), higher $I_{on}/I_{off}$, and field-effect hole mobilities ($\mu_h$) values, as shown in Fig. 4g, h and Supplementary Table 3.

$I_{on}$ increased as $L$ reduced, and it approached ~7.8 ± 1.4 µA·µm$^{-1}$ for the 2-µm-length channel (Fig. 4g and Supplementary Fig. 18a, b). The reciprocal relationship between $I_{ds}$ and $L$ ($I_{ds} \propto 1/L$) matches well with the $I_{on}$ values extracted in Fig. 4g, indicating ohmic behavior. The comparison of reported $I_{on}$ values of 2H-MoTe$_2$ transistors at the carrier densities of the 2D channel ($n_{2D}$) induced by $V_g$ (Supplementary Fig. 18c) suggested more effective hole transport in our device than in the literature[20,34,35,59]. In addition, the $I_{on}$ values obtained in this study are higher than those achieved for CVD-grown WSe$_2$ (Supplementary Fig. 19a). They were also comparable to one of the best p-type 2D transistors based on mechanically exfoliated WSe$_2$ multilayers (Supplementary Fig. 19b and Supplementary Table 4), indicating the high channel conductivity of our MoTe$_2$ polymorphic transistors. In theory, 2H-MoTe$_2$ can behave as a better unipolar p-type channel for CMOS devices than other group-VI 2D TMDs while suppressing n-type transport, given its $E_{VBM}$ and band alignment with respect to the $E_F$ of the metal contact (Supplementary Fig. 20).

The high-quality thin-film production method described in this study was instrumental in achieving high performances of FETs. Even using 3D contact electrodes (i.e., Pt), high $I_{on}/I_{off}$ ratio (~$10^4$) and $\mu_h$ (~5.1 ± 2.8 cm$^2$V$^{-1}$s$^{-1}$) values (blue star in Fig. 4h) could be obtained for our single-crystal 2H-MoTe$_2$ transistors, which was significantly better than those for the majority of the mechanically exfoliated samples (black circles in Fig. 4h). This may be attributed mainly to the absence of oxidation-related defects in our MoTe$_2$ (see XPS in Fig. 1g and Supplementary Fig. 4), since the oxidation induces the increase in the off-state current ($I_{off}$) and thereby degrading the $I_{on}/I_{off}$ ratio (<$10^3$) (Supplementary Fig. 21a, b). In addition, the amorphous structure and non-stoichiometric 2H-MoTe$_2$ layers resulted in relatively minor performance enhancements, as indicated by the performance of the directly synthesized 1T'/2H-MoTe$_2$ MSJ transistor (Supplementary Fig. 22) in this study. Because the locally residual 1T' metallic phase in a 2H film can similarly reduce the $I_{on}/I_{off}$ ratio[12], a synthesis technique capable of facilitating a complete phase transition to 2H is required for achieving a better device performance. Moreover, the abnormal grain-grown 2H-MoTe$_2$ single crystals could avoid the GB scattering mechanism of the carriers, which is beneficial for sustaining the high conductivity of the active layer compared with that of polycrystalline MoTe$_2$ (Supplementary Fig. 21c, d). Furthermore, the high $\mu_h$ of our 6-layer 2H-MoTe$_2$ (~29.5 cm$^2$V$^{-1}$s$^{-1}$), exceeding those with comparable or greater thicknesses, indicates its high crystalline quality (Supplementary Fig. 18d). In addition to the 2H phase, the as-synthesized 1T'-MoTe$_2$ displayed a low sheet resistance ($R_{sh} \approx$ 7.0 kΩ·sq$^{-1}$), which is comparable to that of the mechanically exfoliated samples (Supplementary Fig. 23), suggesting that the carrier was injected effectively through the semi-metallic layer.

Our method to fabricate the heterostructure via the conventional lithography technique enabled us to obtain the transfer length method (TLM) patterns (inset in Fig. 4i) for calculating the $R_c$ of the heterojunction. TLM patterns could not be realized via layer-by-layer assembly in previous efforts using 2D (semi-)metallic TMD flakes[10,29,35,65], thus indicating the improved scalability and reproducibility of our approach. The total resistance ($RW$) of the vdW Au/1T'-MoTe$_2$-contacted FET demonstrated its linear dependence on the $L$. Consequently, the $2R_c$ at the $y$-intercept was calculated from Fig. 4i (see Method for details). Figure 4j shows the representative TLM-driven $R_c$ of the FETs based on the $V_g$-induced $n_{2D}$, where the major carrier of the hole was injected through the vdW Au/1T'-MoTe$_2$ (red) and 3D Pt (blue) contact electrodes. Here, we estimated $R_c$ for the 3D Pt-contacted 2H-MoTe$_2$ MSJ FET as a counterpart of the vdW Au/1T'-MoTe$_2$ contact because hole transport through Pt demonstrated the best characteristics among our characterized 3D metal contacts (see Fig. 4f and Supplementary Fig. 14c, d). At an induced $n_{2D}$ of ~7.4 × $10^{12}$ cm$^{-2}$ (at $V_g$ = −100 V), the Au/1T'/2H-MoTe$_2$ system showed a low $2R_c$ of ~2.4 ± 1.0 kΩ·µm, substantially lower than that of Pt/2H-MoTe$_2$ (≈660 ± 82 kΩ·µm at ~7.8 × $10^{12}$ cm$^{-2}$) (Supplementary Fig. 24a). Because the TLM-driven $2R_c$ of this 3D/2D/2D system included the $R_c$ of the 3D/2D metal-pad/1T'-MoTe$_2$ interface (~0.5 kΩ·µm; Supplementary Fig. 23c), the actual $R_c$ at between the 2H- and 1T'-MoTe$_2$ (which is the true interface between the semimetal and the semiconductor) approached ~0.7 ± 0.5 kΩ·µm at $n_{2D}$ of ~7.4 × $10^{12}$ cm$^{-2}$. For a fair evaluation of the calculated $R_c$, we also plotted graphs for the reported values of $R_c$ for 2H-MoTe$_2$ FETs[18,20,24,34–36,59,61,66,67] depending on $n_{2D}$ and $R_{sh}$ in Fig. 4j and k, respectively (see their comprehensive comparisons in Supplementary Table 3). The efficient switching of a transistor necessitates a significant shift in $R_{sh}$ at a low $R_c$ (and high $I_{on}/I_{off}$ ratio as depicted in Supplementary Fig. 24b, c) and our vdW-integrated device fulfills this requirement. Given its $n_{2D}$, our 2H-MoTe$_2$ MSJ FET with a TLM-extracted $2R_c$ of ~2.4 ± 1.0 kΩ·µm exhibited the lowest recorded value among the p-type 2H-MoTe$_2$-based FETs that exhibited significant switching behavior (i.e., $I_{on}/I_{off}$ > $10^3$) (Fig. 4j and Supplementary Fig. 24b). Furthermore, the on-state $R_{sh}$ of our 2H-MoTe$_2$ was ~44.3 ± 2.3 kΩ·sq$^{-1}$, which is the smallest CVD-grown MoTe$_2$ FET analyzed using TLM in prior studies[18,34–36,59] (Fig. 4k). This result indicates that our channel material shows a higher quality (see Supplementary Fig. 24c–e for more comparative studies) because $R_{sh}$ represents material-dependent properties that exclude contributions from device dimensions and contact properties. Despite their low $R_c$, reports on FETs with poor $I_{on}/I_{off}$ ratios or $R_{sh}$ highlight the importance of a high-quality active layer for achieving high performance (i.e., free of partial oxides or metallic phases; their effects are discussed in Supplementary Figs. 21 and 22).

## Near-zero SBH at Fermi-level-tuned contact interface

The selection of various 3D metals (e.g., Au, Pt, and Ag) as contact pads on vdW 1T'-MoTe$_2$ electrodes provided controllability of the transfer characteristics of 2H-MoTe$_2$ MSJ FETs (Fig. 5a and Supplementary Fig. 25a, b). The hole injection efficiency of the MSJ was maximum when vdW Au/1T'-MoTe$_2$ was used, and it was followed by vdW Pt/1T' and vdW Ag/1T' contacts, as indicated by their various current densities at $V_g$ below 0 V (Fig. 5a). Furthermore, the threshold voltage ($V_{th}$) was shifted by changing the 3D metal in the vdW 3D/2D metallic system. Given the same fabrication process and channel material, the $V_{th}$ shift indicates the tunability of carrier injection from the contact owing to the reduced $R_c$ and SBH rather than the doping effect of the channel. The modulation of the contact property could be caused by the change in the WF values of the vdW semimetal of 1T'-MoTe$_2$ in response to the deposited 3D metals based on UPS measurements (Fig. 5b). For example, the vdW Au/1T'- and vdW Pt/1T'-MoTe$_2$ surfaces exhibited large WF values of ~5.0 eV and ~5.6 eV, respectively, compared to that of pristine 1T'-MoTe$_2$ (WF ≈ 4.5 eV), which is favorable for hole transport. However, despite the largest WF modulation by Pt, the performance enhancement of the FET by the vdW Pt/1T'-MoTe$_2$ contact was not significant compared to that of vdW Au/1T'-MoTe$_2$ (Fig. 5a). This was because of the degraded vdW Pt/1T'-MoTe$_2$ interface due to the formation of PtTe$_x$ and non-stoichiometric MoTe$_{2-x}$, as indicated in the XPS study (Fig. 5c and Supplementary Fig. 15). The XPS Te $3d$ scans of vdW Ag/1T'- and vdW Au/1T'-MoTe$_2$ revealed that the

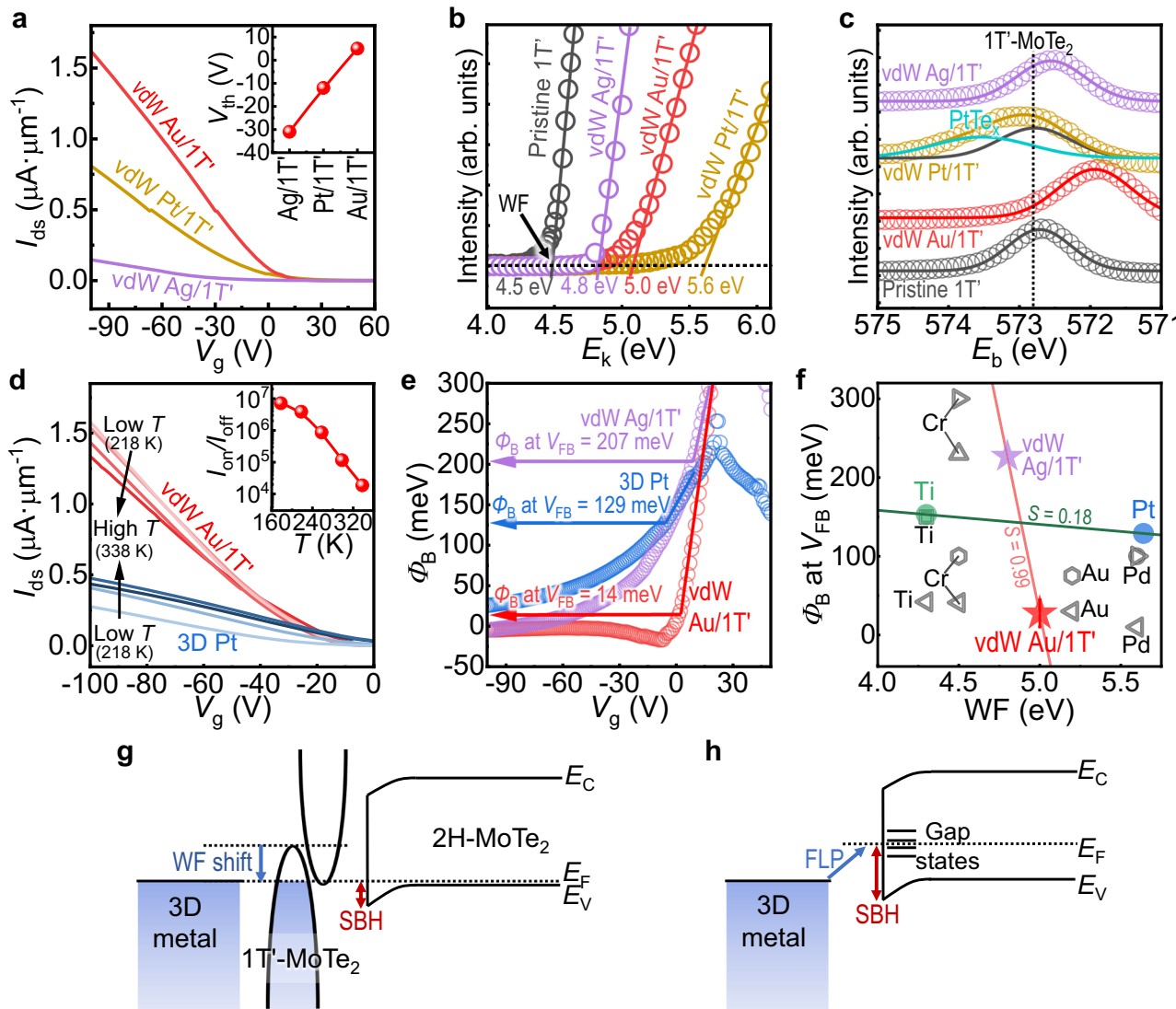

**Fig. 5 | Fermi-level-modulated van der Waals (vdW) semimetal contact electrodes. a** Transfer characteristics of 2H-MoTe$_2$ FETs with vdW Au/1T'- (red), Pt/1T'- (gold), and Ag/1T'-MoTe$_2$ (purple) contacts. Inset illustrates threshold voltage ($V_{th}$) shifts by metallized 3D metals on vdW semimetal contacts. **b, c** Ultraviolet photoelectron spectroscopy (UPS) and X-ray photoelectron spectroscopy (XPS) characterization of the interfaces at 1T'-MoTe$_2$ deposited by 3D metals; Ag (purple), Au (red), and Pt (gold). Spectra of pristine 1T'-MoTe$_2$ are displayed as gray curves. **b** UPS spectra of the 3D metal/1T'-MoTe$_2$ surfaces in secondary electron edge region. Work function (WF) can be extracted at the $x$-intercept of linear regime (solid lines) as kinetic energy ($E_k$) is the difference between the energy of ultraviolet photons (-21.21 eV for the He I radiation) and binding energy ($E_b$). **c** Normalized XPS spectra of Te 3$d$ core level acquired from the 3D metal/1T'-MoTe$_2$ interfaces. Dotted line indicates $E_b$ of pristine 1T'-MoTe$_2$. **d** Typical $I_{ds}$–$V_g$ of 2H-MoTe$_2$ FETs contacted with vdW Au/1T'-MoTe$_2$ (red) and 3D Pt (blue) at various temperatures ($T$). Inset shows the $I_{on}/I_{off}$ ratio with $T$. **e** Thermionic barrier height ($\Phi_B$) values measured at various $V_g$ for hole transfer with 3D Pt (blue), vdW Ag/1T'-MoTe$_2$ (purple), and vdW Au/1T'-MoTe$_2$ contacts (red) in 2H-MoTe$_2$ FET. The Schottky barrier heights (SBHs), i.e., $\Phi_B$ values at the flat band voltages ($V_{FB}$), are indicated by arrows. **f** Comparison of SBH values of our 2H-MoTe$_2$ FETs (colored stars) with reported ones[23,34,36,60–63] (binned black) depending on WFs of different metal contacts. Fermi level pinning (FLP) factors, $S$ (=d$\Phi_B$/d$WF$), for vdW semimetal and 3D contacts are displayed. **g, h** Band structures depicting formation of SBH and alignment of Fermi level ($E_F$) in 2H-MoTe$_2$ FETs with (**g**) vdW semimetal and (**h**) 3D metal contact. $E_c$ and $E_v$ indicate energy levels of conduction and valance band edge, respectively.

energy shifted to lower values with minimal peak broadening, indicating doping of 1T'-MoTe$_2$ induced by charge transfer from Ag or Au and a lack of interface problems. This suggests that the selection of non-reactive 3D metal pads is highly important for achieving the desired performance of 2D FETs with vdW 2D (semi-)metal contacts.

To further investigate the contact properties of the FETs, electrical measurements were conducted at various temperatures (178 < $T$ < 338 K). Figure 5d shows the representative transfer characteristics of the 2H-MoTe$_2$ FETs in contact with the Fermi-level-modulated vdW semimetal contact (i.e., vdW Au/1T'-MoTe$_2$) and 3D Pt (which showed the best $p$-type performance among the tested 3D metal contacts). We first observed an increase in the $I_{on}/I_{off}$ ratios as the $T$ decreased, approaching ~10$^7$ at 178 K (inset of Fig. 5d; the corresponding semi-log

scale curve is shown in Supplementary Fig. 25a), as a result of the suppressed thermionic-assisted emission of the carrier for the $I_{off}$ at a low $T$. While the current levels of Pt/2H-MoTe$_2$ FETs demonstrated a decrease in the $I_{on}$ as the $T$ decreased, the 1T'-MoTe$_2$-contacted transistors did not exhibit such deterioration. Conversely, their $I_{on}$ values increased significantly as the $T$ decreased from ~338 K to ~218 K (Fig. 5d and Supplementary Fig. 25h). This increase in the channel conductivity was attributed to the insulator-to-metal transition (MIT) over $T$. The metallic response of $I_{on}$ of 2H-MoTe$_2$ at low $T$ suggests that the thermionic-assisted transport over a certain barrier height ($\Phi_B$) at the 1T'/2H junction interface was negligible, allowing carriers to tunnel through it to exhibit its intrinsic transport nature. Hence, the MIT behavior of our FETs implies the realization of a genuine ohmic contact

for holes. Additionally, the carriers in vdW Au/1T'/2H-MoTe₂ FETs followed the phonon-limited transport at temperatures between 218 K and 338 K, showing the power-law dependence of $\mu \propto T^{-0.92}$ (Supplementary Fig. 25e)[68]. In contrast, the $\mu$ values of the Pt-contacted FETs degraded vastly as the $T$ decreased, which indicates that the high $\Phi_B$ majorly restricted the carrier transport through the 3D metal at the turn-on state because of the relatively high contribution from thermionic emission at the contact interface.

According to the thermionic emission model, the information of $I_{ds}$ depending on the $T$ additionally allow the calculation of the effective thermionic barrier height, i.e., $\Phi_B$ values, as follows:

$$I_{ds} = \left[ AA^* T^{3/2} \exp\left( -\frac{q\Phi_B}{k_B T} \right) \right] \left[ \exp\left( \frac{qV_{ds}}{k_B T} \right) - 1 \right] \quad (1)$$

where $A$ is the area of the heterojunction, and $A^*$ is the effective Richardson–Boltzmann constant. From the above-mentioned equation, the value of $\Phi_B$ could be extracted from the slope of the linear fit to the Arrhenius plot (ln $(I_{ds}/T^{3/2})$ vs. $1000/T$) as depicted in Supplementary Fig. 25f. The vdW Au/1T'/2H-MoTe₂ FET demonstrated a positive slope (negligible $\Phi_B$) in comparison to the 3D Pt-contacted FETs, which demonstrated a negative slope (existence of $\Phi_B$) at the $V_g$ near the $V_{th}$. The $\Phi_B$ of the 2H-MoTe₂ FETs, extracted from the slope, was further plotted as a function of $V_g$ in Fig. 5e (and Supplementary Fig. 25g). The behavior of $\Phi_B$ was observed to be dependent on the $V_g$ with regards to the flat band voltage ($V_{FB}$; the point at which the linear relationship between $V_g$ and $\Phi_B$ ends as the $p$-type FET turns on by decreasing the $V_g$) because the band bending after the flat band condition ($V_g < V_{FB}$) induces tunneling transport through the narrowed barrier width in addition to thermionic emission over the SBH (Supplementary Fig. 26). Consequently, the $\Phi_B$ at the exact $V_{FB}$ is interpreted as the "accurate SBH" for the thermionic emission. The SBH (i.e., $\Phi_B$ at $V_{FB}$) of vdW-integrated Au/1T'/2H-MoTe₂ FET approached almost zero (-14 meV; and -27.4 ± 17.3 meV in average in Supplementary Fig. 25j), which was much smaller than that of 3D metal Pt/2H MoTe₂ FET (-129 meV) as depicted in Fig. 5e. The significant band bending by further inducing $V_g$ gave rise to the negative values of $\Phi_B$, which also signifies the transparency of the $\Phi_B$ at the tunneling transport regime at $V_g$ below 0 V. This negligible SBH achieved by vdW Au/1T'-MoTe₂ contact is reproducible, regardless of the channel length ($L$) as displayed in Supplementary Fig. 25h–j. Furthermore, the obtained near-zero SBH (i.e., $\Phi_B$ at $V_{FB}$) was the lowest value among the reports for 2H-MoTe₂ FETs, as summarized in Fig. 5f and Supplementary Table 3.

Ideally, the band alignment between metal and semiconductor determines SBH by following the Schottky–Mott rule for $p$-type semiconductor[69]. This is based on the following equation:

$$SBH = E_g + \chi - WF, \quad (2)$$

where $E_g$ is the bandgap of 2H-MoTe₂ (~0.89 eV) and $\chi$ is the electron affinity (~4.12 eV) (Supplementary Figs. 7c, d). However, the SBH of the 2H-MoTe₂ FETs was not effectively modulated by the WFs of the vertical 3D metal contacts. For example, our 2H-MoTe₂ FET contacted to 3D Pt metal contact (with a WF value of ~5.6 eV) demonstrated a sizeable SBH for holes (-129 meV) (Fig. 5e, blue). However, considering the above theory, the ideal SBH should be zero for the metal with a WF higher than ~5.01 eV. In addition, even for the low-WF metal contact using Ti (WF ≈ 4.4 eV), the interface has a sufficiently low SBH (-188 meV) to exhibit a $p$-type transport (Supplementary Fig. 25g). The large discrepancy between the SBHs and WFs suggests that the Fermi level pinning (FLP) could be attributed to the defect-induced gap states (see the defects at the 3D metal/MoTe₂ interface in Fig. 4e). Hence, the shifted charge neutrality resulted in the strong FLP for various metals, and even the indicator for the strength of FLP, $S$ [= d(SBH)/d(WF)],

approached almost zero for our Pt or Ti-contacted FETs ($S \approx 0.18$; blue line in Fig. 5f). We can also observe the similar FLP phenomena in the previous studies on MoTe₂ FETs using Ti[34,63], Cr[23,60,61,63], Pd[23,62,63], and Au[36,60,63] contacts (Fig. 5f), that is, the reported FETs demonstrated $p$-type transport and similar SBHs for holes (-110 ± 85 meV), regardless of the WFs of the corresponding metal electrodes.

In comparison, the MSJ interfaces of the 2H-MoTe₂ FETs with vdW 3D-metal/1T'-MoTe₂ contact electrodes were close to the Schottky–Mott limit because their SBH values were effectively modulated by their different WF values. The experimentally obtained averaged SBH of the vdW Au/1T'-MoTe₂-contacted 2H-MoTe₂ MSJs was -27.4 ± 17.3 meV (Supplementary Fig. 25j), which is comparable to that of the calculated value (-10 meV) using Eq. 2 and the WF of the doped vdW 1T'-MoTe₂ contact (-5.0 eV; Fig. 5b). In addition, a larger SBH could be achieved by metallization using the vdW Ag/1T'-MoTe₂ contact, which approached -207 meV (Fig. 5e), in response to its smaller WF (-4.8 eV; Fig. 5b). The pinning factor, $S$, calculated for the FETs in contact with the Fermi-level-modulated vdW 1T'-MoTe₂ metal contacts was -0.99, which is close to the ideal value (-1) for the Schottky–Mott limit, indicating partially depinned vdW interfaces.

Figure 5g, h describes the proposed roles of the semimetallic 1T'-MoTe₂ electrode in forming an FLP-free interface with a tunable SBH in terms of band alignment and charge transfer as follows: First, unlike the 3D metal (Fig. 5h), the vdW integration for the MSJ interface could avoid defect production at the underlying 2H-MoTe₂ surface (i.e., defect-induced gap state). In addition, since the density of state (DOS) of the 2D semimetal is smaller than that of the 3D metal, there are few opportunities to generate a metal-induced gap state, obviating the need for FLP[16]. Most importantly, the small DOS of 1T' MoTe₂ facilitates the changes in its doping level in response to the $E_F$ of the 3D metal adsorbed on the top surface (Fig. 5g), a phenomenon similar to that observed in the semi-metallic graphene. In this respect, the Au-metallized 1T'-MoTe₂ can have a large WF value that is promising for a $p$-type contact, and its ultraclean vdW contact with a 2D semiconductor can allow an FLP-free MSJ, thereby facilitating a scalable technology to produce FETs with low SBHs. This strategy also provides versatility and universality in obtaining FLP-free MSJ interfaces and high-performance $p$-type 2D MSJ FETs. For example, we applied the method to MSJ FETs with another 2D semiconductor, WSe₂ (Supplementary Fig. 27). Compared with the $n$-type feature of multilayer WSe₂ FETs fabricated by evaporation of the 3D metal contact (i.e., Au; Supplementary Fig. 27b, d)[70,71], the interfacial defect-free, Fermi-level-tuned vdW Au/1T'-MoTe₂ contact electrodes allowed to achieve $p$-type transport in WSe₂ MSJ FETs (Supplementary Fig. 27a, c). Thus, our FLP-free, vdW contact scheme to control the $n$- and $p$-type polarity is a practical approach to realize 2D integrated circuitry.

## Discussion

We demonstrated phase- and position-controlled growth of 2H-MoTe₂ single crystals, as well as 3D-metal-assisted vdW integration of two polymorphs, to fabricate high-performance $p$-type MSJ FETs arrays on a large scale. The Te-rich atmosphere facilitated the synthesis of high-quality MoTe₂ and their phase transition from 1T' to 2H, where the 2H single-crystal domains could expand and cover the 4-inch-wafer SiO₂/Si by abnormal grain growth. Our growth technique could be extended to construct bilayer MoTe₂ films on a large scale (~20 mm) without notable micro-voids or impurities, which is the thinnest film that can be obtained using the CVD mode on a multimeter scale. Furthermore, the spatial arrangement of energetically favorable 2H-seed layers enabled the lateral solid-phase epitaxy of single-crystal 2H-patterns, which promises controllability of the crystallographic orientation of the 2D semiconductor on any amorphous substrate at a low temperature of ~500 °C. Besides, arrays of high-performance vdW-integrated MSJ FETs were fabricated by standard photolithographic patterning and transferring the 3D-metal-deposited 2D 1T'-MoTe₂

structures onto 2D 2H·MoTe$_2$ single crystals. Compared to previous reports, the single-crystal 2H·MoTe$_2$ MSJ FETs in this study performed better in terms of $I_{on}/I_{off}$ (-1.3 × 10$^5$) and $\mu_h$ (-29.5 cm$^2$V$^{-1}$s$^{-1}$). The significantly high on-state conductivity ($I_{on}$ ≈7.8 μA·μm$^{-1}$) was contributed by the high quality of MoTe$_2$ crystallinity, that is, the absence of oxides, metallic impurities, and non-stoichiometric or amorphous structures. Furthermore, the formation of an ultraclean and atomically precise interface between 1T′ and 2H MoTe$_2$ prevented the formation of gap states while creating an FLP-free interface. This also allowed us to tune the height of the thermionic emission barrier based on the electronic states of various 3D metals on the 2D semimetal. For example, metallization of 1T′ semimetal with Au provided a high WF value (-5.0 eV) of vdW metal electrodes, significantly suppressing the barrier height for holes to transport at the contact interface (-14 meV at $V_{FB}$).

Given the relative ease of obtaining an $n$-type 2D transistor, our study represents considerable progress toward scalable fabrication of $p$-type 2D transistors with enhanced hole transport. For example, 2D semiconductors are vulnerable to extrinsic $n$-type doping by chalcogen vacancies and dielectric layers, but our approach may assist in avoiding them. Synthesis of $p$-type 2D selenides and tellurides often necessitates high growth temperatures (>700 °C) (primarily due to the fact that Se and Te have lower vapor pressures than S), which degrades 2D channel quality while producing thermal-induced chalcogen vacancies. Se and Te vacancies in 2D chalcogenides are considered $n$-type dopants[70–72], which provoke the FLP near the conduction band when a 2D semiconductor comes in contact with a metal electrodes[72]. In addition, widely used dielectric layers such as SiO$_2$[1] and AlO$_x$[73] can result in $n$-doping of the 2D active layer. In this regard, our study guarantees the prevention of an unintended $n$-doping effect by the high-quality growth of a 2D single-crystal semiconductor on an arbitrary amorphous substrate at a low temperature (-500 °C).

Furthermore, the conventional $p$-type high-WF 3D metal contacts (e.g., Au, Pt, and Pd) have high melting points (-1064–1768 °C), which requires a high-energy deposition process that may damage a 2D MSJ interface and increase its $R_c$ (Note that this is in contrast to the case for $n$-type ohmic metal contacts for MoS$_2$ (e.g., In and Bi) with smaller melting points (-157–271 °C)[16,74], which yields a defect-less contact interface.). Accordingly, our approach for on-chip fabrication of ultraclean high-quality vdW MSJ transistor arrays has significance in terms of realizing negligible contact barriers for improved $p$-type transport. The importance of our results is also reflected in the obtained ultralow $R_c$ of -0.7 kΩ·μm between 2D semimetal and 2D semiconductor (where the $R_c$ value was extracted by excluding the $R_c$ of 3D metal/2D semimetal). This $R_c$ value is at their lowest level yet recorded for $p$-type few-layered 2D transistors based on 2H·MoTe$_2$ and even WSe$_2$ (Supplementary Tables 3 and 4). Therefore, by combining material synthesis, device manufacturing, and interface engineering, our study has considerable potential to broaden the range of 2D electronic applications.

## Methods

### Polymorphic MoTe$_2$ growth

The synthesis of large-area MoTe$_2$ films was carried out utilizing a typical hot-wall furnace system set up within a Te-gas restricted system consisting of sandwiched Mo and Ni$_x$Te$_y$ substrates. To prepare the Ni$_x$Te$_y$ precursor for growth, a DC magnetron sputtering technique was used to create a Ni layer (65 nm) atop SiO$_2$/Si, followed by powder-based tellurization at 500 °C for 10 min In the case of the Mo precursor, a Mo film (1–20 nm) was formed on the other SiO$_2$/Si wafer using DC sputtering under optimum deposition conditions (less than 5% uniformity throughout the wafer). Afterwards, the two prepared precursors (Ni$_x$Te$_y$/substrate and Mo/substrate) were sandwiched without any powder precursor at the center of the furnace. The MoTe$_2$ film was prepared via tellurization of the Mo precursor by

simply heating the reactants at different $T$ and atmospheric pressure with H$_2$/Ar (100/500 sccm) as the carrier gas in an 8-inch quartz tube for the designated growth time ($t$ = 0–90 min, depending on the experimental conditions). For seed growth, a 10-nm-thick 2H·MoTe$_2$ film with ultra-large single-crystalline domains was introduced as seed arrays, which was synthesized in the same manner as in Fig. 1. The 2H·MoTe$_2$ thin film was patterned and dry transferred to pre-grown 1T′ MoTe$_2$, followed by heating to 500 °C with the Ni$_x$Te$_y$ stack, using the same growth setup for the polymorphic thin film. For the dry transfer, the 0.4 M PMMA was coated onto 2H·MoTe$_2$/SiO$_2$, and then attached to a one-sided thermal release tape. The prepared "tape/PMMA/2H·MoTe$_2$" could be detached from the SiO$_2$ substrate by mechanical forces applied from the side of the sample. The structure was picked-up and manually transferred to the desired substrate of 1T′-MoTe$_2$/SiO$_2$/Si substrate. A -300 g weight was placed on the top of the structure. The thermal release tape could be detached by heating to -150 °C for -5–10 min ensuring the vdW adhesion between the 1T′-MoTe$_2$ and 2H·MoTe$_2$. The PMMA was removed by dipping in acetone solution (CMOS grade, J.T. Baker) for -20 min, followed by rinsing with isopropanol alcohol. To prevent the thermal budget effect, no heat treatment at temperatures above 150 °C was performed during the transfer.

### Structural characterization

Micro-Raman measurements were performed with an Ar ion laser (514.5 nm), which was focused using an objective lens with ×40 magnification (numerical aperture of 0.6). The scattered light was collected with the same objective lens, dispersed with a Jobin-Yvon Horiba iHR550 spectrometer (2400 grooves/mm), and detected with a back-illuminated charge-coupled-device array detector. Rayleigh-scattered light was rejected using volume holographic filters (Optigrate). To minimize the local heating of the samples, the laser power was maintained at 100 μW. The spectral resolution was less than 1 cm$^{-1}$. The 514.5 nm line of the Ar laser was employed for polarized Raman studies. The polarization direction of the linearly polarized incident laser beam was rotated to the desired direction using a half-wave plate. The angle of the analyzer was chosen such that photons with polarization parallel to the incoming beam may pass through (parallel configuration). Another achromatic half-wave plate was placed in front of the spectrometer to keep the polarization direction of the signal constant with respect to the groove direction of the grating. The spectroscopical ellipsometry (Woollam Ellipsometry M-2000 with detector spectral range of 371–1687 nm) method has been performed and the data were fitted using the CompleteEase software and a multi-Lorentzian model. The multi-incidence measurement mode (angle of incidence (AOI): 60°, 65°, and 70°) with sampling times of 3 s was used to measure the MoTe$_2$ films. Ellipsometry was also performed on the SiO$_2$/Si substrate to increase the accuracy of the results (Supplementary Fig. 28). The calculation of $\alpha$ in Fig. 1j was based on the measured $k$ values from the ellipsometry measurement (Fig. 1i), using the relationship $\alpha = 4\pi k/\lambda$, where $\lambda$ is wavelength of incident light. XRD patterns were obtained using a Bruker AXS D8 machine equipped with a Cu K source. AFM images were obtained using the Bruker Dimension AFM in the tapping mode. High-resolution STEM images, SAED patterns, and EDS were obtained using the aberration-corrected FEI Titan[3] G2 60-300 at an acceleration voltage of 200 kV. STEM images were obtained using a high-angle annular dark-field (HAADF) detector to collect semi-angles from 50.5 to 200 mrad. A Wiener filter was used to subtract the noises of high-resolution STEM images. The commercially available software TEMPAS (Total Resolution) was employed for multi-slice STEM image simulation. TEM images and diffraction patterns (Supplementary Figs. 8 and 9) were captured utilizing a Tecnai G2 F20 X-Twin system at an acceleration voltage of 200 kV. The single-crystalline MoTe$_2$ film in Supplementary Fig. 13b, c was characterized

by SEM (FEI Verios 460) and EBSD (AMTEK, Inc., Hikari). XPS and UPS measurements were performed by the ESCALAB 250XI system (Thermo Fisher, K-alpha) equipped with a micro-focused monochromatic Al X-ray source. The XPS calibration was accomplished to align the binding energy of 284.5 eV for the C 1s line.

### vdW integration of junction transistor and measurements

To fabricate the vertical 1T'/2H heterostructure, large-area synthesis was carried out at various growth $T$ and $t$ for each phase, such as $T = 700\,°C$, $t = 60$ min, and $T = 500\,°C$, $t = 30$ min for the 2H and 1T' structures, respectively. The Au/1T'-MoTe₂ patterns were created by depositing arrays of Au layers (40 nm) by conventional photolithography and an e-beam evaporator (Temescal FC-2000). To reduce the material damage and interface degradation during the high-energy deposition process, the deposition rate was reduced to 0.1 Å/s in ultra-high vacuum ($10^{-9}$ Torr). Then, the reactive ion etching (RIE) technique using $SF_6$ and $O_2$ plasma removed the exposed 1T'-MoTe₂, except for the underlying structure behind the Au patterns. The dry transfer method (the same process used for seed growth from the 2H-MoTe₂ pattern) enabled the formation of the Au/1T'-MoTe₂/2H-MoTe₂ junction (Supplementary Fig. 14). After the dry process of integrated 3D/2D metals, the 2H-MoTe₂ channel width was defined using the RIE process. Electrical characterization was carried out in a cryogenic probe station (Lakeshore CRX-4K) equipped with a Keithley 4200-SCS detector at 138–300 K and a high vacuum reaching $10^{-6}$ torr. A back-gate voltage was applied to the FETs via the 300-nm-thick $SiO_2$ on the Si layer of the substrate. The $n_{2D}$ was calculated using the following equation; $n_{2D} = C_{ox}(V_g - V_{th})/q$, where the $C_{ox}$ is a capacitance of oxide dielectric. The average device-to-device variation ($C_V = \sigma/\mu$, where $\sigma$ denotes the standard deviation and $\mu$ the mean value) was estimated to be ~11 ± 5% for our TLM devices (Fig. 4). The non-uniformity of the 2D material, contact and dielectric interface, or local charge impurities can lead to device-to-device variations[75]. The error bars in Fig. 4i are the regression standard errors resulting from the linear fitting of the TLM plots, which were scaled with the square root of the reduced Chi-squared statistics for at least five different TLM sets. All statistically estimated values this study are denoted as 'average ± standard deviation,' except for the fitted values ('mean ± standard error').

## Data availability

Relevant data supporting the key findings of this study are available within the article and the Supplementary information file. All raw data generated during the current study are available from the corresponding authors upon request.

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

## Acknowledgements
This work was supported by the 2020 research Funds (1.200041.01 and 1.200095.01) of UNIST, Institute for Basic Science (IBS-R019-G1), and

National Research Foundation (NRF) of Korea (Grant Nos. 2017M3A7B8065377, 2021R1A2C2094674, 2021R1A3B1077184, and 2021R1A6A3A14038492) funded by the Ministry of Science, ICT, and Future Planning. D.J. and J.L. acknowledge partial support from Asian Office of Aerospace Research and Development of the Air Force Office of Scientific Research (AFOSR) grant nos. FA2386-20-1-4074 and FA2386-21-1-4063 and GHz-THz program grant no. FA9550-21-1-0035. C.Y.C. acknowledge the National Science Foundation Graduate Research Fellowship Program (NSF GRFP) under grant number DGE-1845298.

## Author contributions

S.S. prepared materials and performed most of the experiments with assistance from S.J., J.Y., J.H., Y.H.J., J.K. and C.J; A.Y., M.C. and Z.L. conducted the (S)TEM characterizations; Y.C. and H.C. performed the Raman measurements; J.L., C.Y.C. and D.J. conducted ellipsometry characteristics; S.S. and S.-Y.K. wrote the manuscript with the input of all other authors; All authors discussed the results and commented on the manuscript; all the authors revised and commented on the manuscript; S.-Y.K. supervised the project.

## Competing interests

The authors declare no competing interests.
