## [Peer Review File · Nature Communications]

Fabrication of p-type 2D single-crystalline transistor arrays with Fermi-level-tuned van der Waals semimetal electrodesEditorial Note: This manuscript has been previously reviewed at another journal that is not operating a transparent peer review scheme. This document only contains reviewer comments and rebuttal letters for versions considered at *Nature Communications*.

REVIEWER COMMENTS

Reviewer #1 (Remarks to the Author):

The authors has addressed all my questions and has significantly improved the manuscript. I'm now convinced by the authors and I recommend this paper to be published on Nature Communications.

Reviewer #3 (Remarks to the Author):

In this manuscript, the authors developed an large-scale p-type MoTe₂ transistors with Fermi-level tuned van der Waals semimetal electrodes. The MoTe₂ transistors revealed good device performances with high on/off ratio (10^5) and low contact resistance (1.2 kohm·um). Given the difficulties of making good p-type contacts in 2D FETs, I believe it is very timely and could attract the broad readers in 2D device community. I have carefully gone through the whole submitted materials including the rebuttal letter replying to the reviewer's comments. I believe that the authors have answered all the reviewer's comments sufficiently. They provided substantial addition of comparisons to show their work's superiority and novelty such as low temperature nucleation at 500 oC, single crystallinity of 2H-MoTe₂, and Fermi-level tuning of 1T' MoTe₂ by gold metal deposition. This might not be groundbreaking as the reviewers commented, however, it provides a new insight on Fermi-level modification of semimetal electrodes by metal deposition for p-type 2D devices since previous studies have focused on only n-type devices. Overall, I would like to recommend its publication in Nature Communications.

Minor comments:

1) For equation (1), the thermionic emission equation typically uses $T^{3/2}$ for 2D semiconductors (e.g., Adv. Mater. 2022, 34, 2108425).

2) “The experimentally obtained averaged SBH of the Au/1T'-MoTe₂-contacted 2H-MoTe₂ MSJs was $\sim 21.8 \pm 8.3$ meV (Fig. 5b)” on Page 16 seems to incorrectly indicate the figure number. It should indicate Figure S22j.

Reviewer #4 (Remarks to the Author):

Using NixTe_y alloy as a Te source, Song et al. systemically studied the synthesis of MoTe₂ thin films (polymorphic phase). Impressively, the pure 2H phase MoTe₂ thin films can be increased in size up to 4 inches, down to bilayer in thickness, and down to 500 °C in synthesis temperature. However, the phase-controlled synthesis of MoTe₂ thin films has been extensively studied, including the 2H nucleation and growth kinetics with the temperature-time-transformation (TTT) phase diagram (J. Am. Chem. Soc. 141, 2128-2134, 2019). Due to the existence of previous research results and some misleading interpretations in the current submission, I think this submission lacks novelty for publication in Nature Communications.

(1) I agree with Reviewer #2 that MoTe₂ is not “one of the best p-type 2D transistor” materials in terms of on-current and mobility as the authors claim.

(i) The estimated on-current approaching ~ 2.6 mA/ μ m is inaccurate when the channel length is reduced to 5 nm due to the existence of the contact resistance. As a high-performance p-type semiconductor, it is critical to experimentally demonstrate the significant increase in on-current when the channel length is reduced to sub-100 nm.

(ii) The recently developed thermally evaporated Te layers (Nature Nanotech. 15, 53-58, 2020) outperform p-type MoTe₂ based on scalability, electronic performance, and synthesis temperature.

(iii) I agree with Reviewer #2 that the content and the background of the submitted work do not match. Synthesis of large-scale p-type 2D semiconductors at low temperatures targeting integrated circuits. However, the complex contact electrode preparation method is contradictory for this application, especially the involved transfer. The authors point out that this method can be used to drive micro-LEDs in manufacturing lines. However, a high on-state current is critical in this application but was not demonstrated experimentally.

(2) The “2H-seed” growth method has been demonstrated in previous work (Science 372, 195-200, 2021). In the submitted work, the seed array is introduced to trigger multi-site growth, but the manuscript does not introduce how the seed array is transferred. If the seed arrays do not have the same orientation, the resulting thin film is polycrystalline, which has no differences with multiple nucleation and growth. If the seed arrays have the same orientation, one should start with a large-scale single-crystalline thin film, a tricky goal in itself. Therefore, the authors stated “specifically, the use of the 2H seed patterns with the same crystal orientation enables the wafer-scale single-crystal film devoid of and GBs.”, which is only based on imagination without corresponding experimental results.

(3) Some interpretations are misleading.

(i) In the abstract, the authors claimed that they fabricated “4-inch single-crystalline 2H-MoTe₂ wafers”. From my scrutiny of the manuscript, I think the 4-inch wafer is from multiple nucleations, so in the submitted work, it should be poly-crystalline.

(ii) The comparison in the work should be based on a fair comparison, especially in the introduction section, some updated data for MoTe₂ such as on-current and mobility should be provided, not just compared with poorer reported values.

(iii) In conclusion, a contact resistance value of 0.7 kΩ·μm suddenly appeared (for the first time), much lower than the value reported in the main text.

(iv) Supplementary Table 1 is inappropriate because the 4-inch thin film is not based on the bilayer and no device characterization has been performed on bilayer MoTe₂.

(v) In the response, the comparison to the growth time of previous work is unfair. The previous work was grown from single nuclei, while the submitted work was nucleated from multi nuclei, growth, and collapse into a continuous 4-inch continuous 2H MoTe₂ thin film.

(4) On page 4, the authors stated “%(Te/Mo)~2.0 and 1.9 for 1T’ and 2H phases, respectively”, which I think may be a mistake, since the authors state elsewhere “Te-deficient environment to form from the 1T’ structure”.

(5) I’m curious whether the Te supply method or the confined environment promotes the abnormal growth of 2H phase grains. A control experiment can be designed to determine

whether the abnormal 2H phase grain growth can occur at low temperatures using a face-to-face (use a bare substrate without NiTe₂ facing a Mo film) structure to provide a confined environment by using a Te powder as the precursor.

(6) How did the authors state that “Pt is widely known for its exceptionally high WF of ~6.2 eV”? The Pt work function from the wiki reports a value of 5.12-5.93 eV.

Overall Responses to the Reviewers' Comments:

We extend our gratitude to the Reviewers for their insightful comments and recommendations regarding our manuscript, "*Fabrication of p-type 2D single-crystalline transistor arrays with Fermi-level-tuned van der Waals semimetal electrodes.*" Their input has been instrumental in enhancing the overall quality of our work. We have thoroughly revised our article in light of the Reviewers' feedback. We are confident that this revision will significantly improve our work and help prepare the paper for potential publication in *Nature Communications*. For ease of tracking, we have numbered the comments of the Reviewers.

Reviewer #1's Comments (blue) Followed by Our Response (black):

The authors has addressed all my questions and has significantly improved the manuscript. I'm now convinced by the authors and I recommend this paper to be published on Nature Communications.

Response to the comment: We would like to thank you for your recommendation of our manuscript for its publication. Your inquiries and ideas were quite helpful in improving the overall quality of the manuscript. We really appreciate your time and work in providing us with such extensive and insightful advice.

Reviewer #3's Comments (blue) Followed by Our Response (black):

In this manuscript, the authors developed an large-scale p-type MoTe₂ transistors with Fermi-level tuned van der Waals semimetal electrodes. The MoTe₂ transistors revealed good device performances with high on/off ratio (10^5) and low contact resistance (1.2 kohm-um). Given the difficulties of making good p-type contacts in 2D FETs, I believe it is very timely and could attract the broad readers in 2D device community. I have carefully gone through the whole submitted materials including the rebuttal letter replying to the reviewer's comments. I believe that the authors have answered all the reviewer's comments sufficiently. They provided substantial addition of comparisons to show their work's superiority and novelty such as low temperature nucleation at 500 oC, single crystallinity of 2H-MoTe₂, and Fermi-level tuning of 1T' MoTe₂ by gold metal deposition. This might not be groundbreaking as the reviewers commented, however, it provides a new insight on Fermi-level modification of semimetal electrodes by metal deposition for p-type 2D devices since previous studies have focused on only n-type devices. Overall, I would like to recommend its publication in Nature Communications.

Response to the general comment: We are grateful for your careful evaluation of our manuscript and for your recommendation for publication. Although the study may not break new ground in all aspects, we appreciate your recognition that it provides valuable insights into the fabrication and implementation of high-quality, high-performing p-type 2D devices. We

have prepared thoughtful and comprehensive responses to your suggestions, which we would be honored for you to review. Thank you once again for your time and consideration.

1) For equation (1), the thermionic emission equation typically uses $T^{3/2}$ for 2D semiconductors (e.g., *Adv. Mater.* 2022, 34, 2108425).

Response to Point #1: Thank you for your suggestion. Using the revised formula (Equation R1), we recalculated all relevant barrier heights and found no significant differences compared to the values in our manuscript.

$$I_{ds} = \left[AA^* T^{3/2} \exp\left(-\frac{q\Phi_B}{k_B T}\right) \right] \left[\exp\left(\frac{qV_{ds}}{k_B T}\right) - 1 \right] \quad (\text{R1})$$

The newly calculated effective thermionic barrier height (Φ_B) of vdW Au/1T' contact is negligible in contrast to that of 3D Pt, as indicated by the slopes in the Arrhenius plot (**Figure R1a**). The relationship between V_g and Φ_B reveals that the SBH values for vdW Au/1T', vdW Ag/1T', and 3D Pt are approximately 14, 207, and 129 meV, respectively (**Figure R1b**). The obtained SBH values of the metal-semiconductor junction (MSJ) are plotted as a function of the work function (WF) of the contact metals in **Figure R1c**. The SBH values of MoTe₂ transistors with Fermi-level-tuned vdW semimetal contact electrodes can be largely predicted by the Schottky-Mott rule with an S value of ~ 0.99 (red line in **Figure R1c**). Furthermore, the SBHs of the 2H-MoTe₂ transistors fabricated by the direct deposition of 3D metals Ti and Pt are ~ 153 meV and ~ 129 meV, respectively, which are close to each other regardless of the WF values of the metals. When considering the slope, it is evident that the strength of the Fermi-level pinning is substantial ($S = 0.18$, green line in **Figure R1c**).

Figure R1. Calculations of the effective thermionic barrier height (Φ_B) using a 2D model. (a) Arrhenius plots of the vdW Au/1T'/2H-MoTe₂ (red) and Pt/2H-MoTe₂ (blue) MSJ FETs. The extracted Φ_B values are 152.2 meV and -1.6 meV for Pt/2H-MoTe₂ and 1T'/2H-MoTe₂ FETs at n_{2D} of $\sim 6 \times 10^{12} \text{ cm}^{-2}$ ($\Delta V_g \approx 1$ V), respectively. (b) Φ_B values measured at various V_g for hole transfer with 3D Pt (blue), vdW Ag/1T'-MoTe₂ (purple), and vdW Au/1T'-MoTe₂ contacts (red) in a 2H-MoTe₂ FET. The SBHs, i.e., Φ_B values at the flat band voltages, are 129 meV, 207 meV, and 14 meV for Pt/2H-MoTe₂, vdW Ag/1T'/2H-MoTe₂, and vdW Au/1T'/2H-MoTe₂ FETs, respectively. (c) Comparison of SBH values of our 2H-MoTe₂ MSJ FETs

(colored stars) with reported ones¹⁻⁷ (binned black), depending on the WFs of different metal contacts. All the data presented here were determined by the T -dependent Arrhenius approach in (a). Indicators of FLP strength, S ($= d\Phi_B/dWF$), for vdW semimetal and 3D contacts are displayed, where $S = 1$ for SBHs at the ideal Schottky–Mott limit and $S = 0$ for those at the Bardeen limit.

- **Revisions to manuscript:** To reflect the Reviewer’s comment, we have replaced Figs. 5e, f and Supplementary Figs. 25f, g, i, and j. Lines 1, 4, 15, 17 on page 16, and lines 2, 4, 10, 17, 20, 22 on page 17 have been modified to show the newly obtained SBH values and pinning factors calculated using the 2D model.

2) “The experimentally obtained averaged SBH of the Au/1T’-MoTe₂-contacted 2H-MoTe₂ MSJs was $\sim 21.8 \pm 8.3$ meV (Fig. 5b)” on Page 16 seems to incorrectly indicate the figure number. It should indicate Figure S22j.

Response to Point #2: We thank the Reviewer for pointing it out. We have realized that the figure number was incorrect in our initial submission.

- **Revisions to manuscript:** We have corrected this error in lines 17-18 on Page 17.

Reviewer #4’s Comments (blue) Followed by Our Response (black):

Using NixTe_y alloy as a Te source, Song et al. systemically studied the synthesis of MoTe₂ thin films (polymorphic phase). Impressively, the pure 2H phase MoTe₂ thin films can be increased in size up to 4 inches, down to bilayer in thickness, and down to 500 °C in synthesis temperature. However, the phase-controlled synthesis of MoTe₂ thin films has been extensively studied, including the 2H nucleation and growth kinetics with the temperature-time-transformation (TTT) phase diagram (J. Am. Chem. Soc. 141, 2128-2134, 2019). Due to the existence of previous research results and some misleading interpretations in the current submission, I think this submission lacks novelty for publication in Nature Communications.

Response to the general comment: We sincerely thank the Reviewer for taking the time to review our paper and providing constructive feedback to improve our manuscript. We have revised the manuscript accordingly by following the Reviewer’s suggestion in the following point-by-point responses. We assigned numbers to the Reviewers’ remarks to make it easier to keep tracking them. Furthermore, in response to your *general comment* concerning the novelty issue, we would like to convey the significance of our research compared to the previous reports as follows:

The phase-controlled synthesis of MoTe₂ thin films has been extensively investigated in Refs.^{8,9} [J. Am. Chem. Soc. **141**, 2128 (2019); *Science* **372**, 195 (2021)], which were referenced by the Reviewer in this Response. Ref.⁹ explored the phase transformation of MoTe₂ from the 1T’ to 2H phase and its mechanism by employing a time–temperature–transformation diagram. Meanwhile, Ref.⁸ reported on the controlled 2D seed growth of 2H-MoTe₂ on an amorphous substrate using an alumina-passivated film with a small hole during the heating process. These

studies are pioneering and groundbreaking because they experimentally demonstrated a novel methodology for 2H-MoTe₂ synthesis.

Despite their promising strategies, the methods discussed in Refs.^{8,9} face several challenges in terms of scalable fabrication, crystalline quality, production rate, and electronic properties. In Ref.⁹, the use of Te powder in horizontal chemical vapor deposition (CVD) led to a compositional gradient across the substrate from the gas inlet to the outlet, thus limiting the growth scalability to less than 1 cm². The transistor reported in Ref.⁹ also exhibited a limited on-state current density ($I_{\text{on}} < 0.2 \mu\text{A}/\mu\text{m}$ at $V_{\text{ds}} = 0.5 \text{ V}$) and an on-to-off current ($I_{\text{on}}/I_{\text{off}}$) ratio of $\sim 10^4$. It is noted that a statistical analysis of the electrical properties was not conducted in that study⁹. Meanwhile, the seed 2H growth discussed in Ref.⁸ required a long time ($> 72 \text{ h}$) to produce a 1-inch 2H-MoTe₂ wafer. Moreover, the additional removal of the Al₂O₃ passivation layer by H₃PO₄/H₂O etchant solution at 180 °C is needed for its further application to electronic devices. Given the vulnerability of MoTe₂ to oxidation, the wet etching process could substantially degrade the material quality. For example, a small I_{on} of $\sim 93.8 \text{ nA}/\mu\text{m}$ at $V_{\text{ds}} = 0.1 \text{ V}$ and an $I_{\text{on}}/I_{\text{off}}$ of $\sim 10^4$ were reported in Ref.⁸.

In the current study, we present a novel method to obtain 2H-MoTe₂ with better crystalline quality and electrical properties and with a more uniform and larger 2H coverage across the wafers, which is beyond all existing synthetic methods⁸⁻¹⁵ (as summarized in **Table R1**). We achieved continuous, high-quality 2H-phase MoTe₂ on a 4-inch substrate by the newly introduced “Te-confined growth” method in this study. Introducing Te in a vertical configuration with confinement makes it possible to control the Te flux precisely and uniformly. This results in fewer nucleation sites for 2H-phase grains and the formation of larger single domains across the 4-inch-scale wafer, with a high production rate of 50.5 mm/h (calculated as the lateral dimension of the wafer divided by the required growth time) at a growth temperature of 700 °C. Compared to previous reports^{8,10-15} based on powder-based horizontal CVD, this is a significant development because the uniform Te flux prevents variations in the 2H nucleation density and domain size across the wafer (as illustrated in Supplementary Figs. 1e, 3 and Figs. 3d, g; please see Response to Point #5). We also demonstrated how the Te concentration and H₂ carrier gas could enhance the crystal growth rate while suppressing the initial 2H nucleation in the recrystallization kinetics toward a high growth rate (Fig. 3d and Supplementary Fig. 11), which has not been investigated in previous studies on the solid-to-solid phase transition of MoTe₂ including in Ref.⁹. Furthermore, we proposed a new phase transition mechanism accompanied by abnormal grain growth, where the migration of defects (e.g., grain boundaries, dislocations, and vacancies) in 1T'-MoTe₂ relieves its internal strain energy by producing large strain-free 2H grains (Supplementary Fig.10). By adapting the knowledge of abnormal grain growth¹⁶⁻¹⁸, the suppression of the random nucleation of abnormal 2H-phase grains became possible and single-crystal arrays were produced instead. This was achieved by carefully selecting the growth temperature of 500 °C (Fig. 3). Unlike the seed growth method proposed in Ref.⁸, our approach does not require an additional passivation layer, which may ensure better MoTe₂ crystalline quality.

Table R1. Comparison of the crystal quality and electronic properties of 2H-MoTe₂ obtained by CVD. Except for this study and Refs.^{8,12}, the growth was conducted using powder-based horizontal CVD^{9-11,13-15}. The production rate (i.e., throughput) in the unit of [mm/h] is determined based on the lateral dimensions of the wafer [mm] and the time required [h], which should be distinguished from a “growth rate” of a single-crystalline domain⁹. The total conductance ($G_{tot}=I_{on}/V_{ds}$, normalized by channel width) of 2H-MoTe₂ is calculated using the maximum current density of the transistor without consideration of R_c . “N/A” indicates no low-temperature measurements were conducted to investigate the metal-insulator-transition (MIT) behavior of the MoTe₂ channel.

Growth aspects				Electrical properties				Ref.	
Growth mode		Production rate (mm/h)	Growth Temp. (°C)	Possible impurities or residues	Contact metal	G_{tot} ($\mu\text{S}/\mu\text{m}$)	I_{on}/I_{off}		MIT behavior at low temp.
Te-confined mode	Multiple abnormal grain growth	50.5	700	No	vdW Au/1T'	7.8	$> 10^5$	Yes	This study
	Seed growth	0.06	500						
MoO _{3-x} -induced 2H nucleation by powder-based CVD		10	650	No	Pd	0.02	10^3	No	10
					1T' edge contact	1	10^3	No	
Solid-to-solid phase transition by powder-based CVD		7	650	No	1T' edge contact	0.5	10^4		
MOCVD		61.5	400	1T' features in the Raman spectrum	Pd	1.5×10^{-5}	10^4	N/A	12
Random 2H nucleation by powder-based CVD		6.5	700	Oxide peaks in XPS	Ti/Au	0.11	10^2	N/A	13
Seed growth by single nuclei		0.35	680	Required to remove the Al ₂ O ₃ layer	1T' edge contact	0.94	10^4	No	8
MoO ₃ -induced 2H nucleation by powder-based CVD		18	700	Oxide peaks in XPS	Ti/Au	0.2	10^3	N/A	14
Solid-to-solid phase transition by powder-based CVD		0.49	620	Oxide peaks in XPS	1T' edge contact	0.4	10^4	N/A	9

Solid-to-solid phase transition by powder-based CVD			No	Vertical 1T' contact	0.6	10^3	N/A	¹⁹
Flux-modulated growth by powder-based CVD	N/A	585	No	1T' edge contact	0.396	10^3	No	¹

Because of high crystalline quality, our MoTe₂ did not show any signals of oxide impurities or 1T' residues in XPS or Raman measurements compared to those reported previously^{9,12-14} (**Table R1**). The electronic structure of as-grown MoTe₂, extracted by its UPS, is also very similar to that of mechanically exfoliated flakes^{20,21} or resulted from computational simulation^{22,23}, which could not be achieved via synthesis methods²⁴⁻²⁶ (Supplementary Table 2). Furthermore, the high quality of MoTe₂ is indicated by its high adsorption coefficient ($\alpha > 10^4 \text{ cm}^{-1}$) in the visible (VIS) and near-infrared (NIR) regions, which is similar to those of mechanically exfoliated single crystals (red curves in **Figure R2**). MoTe₂ has a smaller bandgap (~1.1 eV), which is in contrast to other 2D transition metal dichalcogenides (TMDs) with larger bandgaps²⁷. This characteristic enables MoTe₂ to absorb light from a broader spectral range, where the α value is even higher than that of bulk semiconductors with a similar band gap, such as Si²⁸ or Ge²⁹ (**Figure R2**), which are conventionally used as photodetectors.

Figure R2. Large absorption coefficient (α) of our MoTe₂ film in the broad range ($\lambda = 300\text{--}1,300 \text{ nm}$). The α values were compared with those reported for other bulk (e.g., Si²⁸ and Ge²⁹) and 2D TMD semiconductors²⁷ (e.g., mechanically exfoliated multilayer MoS₂, WSe₂, and MoTe₂) to display its higher absorption efficiency. The calculation of α was based on the measured k values from the ellipsometry measurement in Fig. 1i, using the relationship $\alpha = 4\pi k/\lambda$.

The large α values are the result of the lowest direct transition in MoTe₂ having an energy close to the indirect band gap which is not the case for Si and Ge³⁰. This suggests that our MoTe₂

has great potential for use in optical communication devices, including saturable absorbers³¹, modulators³², and photodetectors³³. Specifically, we believe that the MoTe₂ FETs can be used as high-performance photodetectors to cover the NIR range, which cannot be achieved with other 2D TMDs with larger band gaps. Under NIR illumination, the photoresponsivity of the transistors can be enhanced by a photo-gating effect. In addition, asymmetric contact barriers formed using different drain/source electrodes can further enhance their rectification ratio³⁴. Our fermi-level-tuned 1T'-MoTe₂ can serve as an efficient vdW contact for hole transport, which promises fast photoresponse due to fewer charge traps at the MSJ interface. To the best of our knowledge, this is the first measurement of α values in large-area, CVD-grown MoTe₂, indicating that the large-area growth and device fabrication methods proposed in this paper may provide exceptional possibilities for various optoelectronic devices.

The exceptional electrical quality of our crystals is further suggested by their high on-state conductance ($G_{\text{tot}} > 7.8 \mu\text{S}/\mu\text{m}$; which is normalized by the channel width) and $I_{\text{on}}/I_{\text{off}}$ ratio ($> 10^5$) of FET, which is better than in any previous report for CVD-grown MoTe₂ (**Table R1**). For example, compared to those reported in Refs.^{8,9}, the total G_{tot} of our 2H-MoTe₂ was 8–19 times better. In addition, we successfully controlled the contact barrier of the interface between the 2H-MoTe₂ channel and the Fermi-level modulated vdW semimetal electrodes for high-performance transistors (Fig. 5). Remarkably, we found that the prevention of interfacial issues at the 3D/2D metallic system is essential to obtain the low SBH and R_c of transistors (Fig 4i, Fig. 5c and Supplementary Fig. 15). The obtained R_c value ($\sim 0.7 \text{ k}\Omega \cdot \mu\text{m}$ if the R_c at 3D/2D Au/1T'-MoTe₂ is excluded) is the lowest one among those reported for *p*-type MoTe₂ (Fig. 4i, j). It should be noted that the Fermi level shift and interfacial defect formation by 3D metal deposition on 2D semimetal have often been overlooked in the considerations of 2D metals for contact electrodes, indicating the importance of our study.

Specifically, our low-temperature electrical measurements revealed an insulator-to-metal transition (MIT) of 2H-MoTe₂ channel conductivity. The MoTe₂ transistor exhibited a crossover from an insulating phase, where I_{ds} increased with rising temperature, to a metallic regime, whereas I_{ds} reduced with increasing temperature, as carrier concentration increased as V_g approaches -100 V (**Figure R3a**). The observed MIT is attributed to the negligible thermionic barrier height at our vdW MSJ interface, allowing the exhibition of the intrinsic transport properties of the MoTe₂ channel³⁵⁻³⁷. The MIT behavior is not simply exhibited in one good device, but appeared in all measured vdW-integrated Au/1T'/2H-MoTe₂ transistors, which validates its reproducibility (Supplementary Fig. 25h). Additionally, we observe an increase in mobility at low temperatures, suggesting that transport was limited by phonons in the channel rather than contact barriers (**Figure R3b**). To the best of our knowledge, this is the first experimental realization of MIT behavior and phonon-limited mobility in two-terminal *p*-type 2D transistors based on CVD-grown MoTe₂, as summarized in **Table R1**. These phenomena have not yet been reported for *p*-type 2D MoTe₂ transistors due to the impact of thermionic barrier height, even though many studies have claimed to achieve negligible R_c and SBH at the contact interface using 1T' MoTe₂ semimetal^{1,8-11,19,38}. This suggests that our study is the only one that achieves a genuine ohmic contact for hole transport in 2D 2H-MoTe₂ transistors.

Figure R3. (reproduced from the manuscript) Low-temperature measurements of the 2D MoTe₂ transistors with vdW Au/1T' or 3D metal contact electrodes. (a) Typical I_{ds} - V_g of 2H-MoTe₂ MSJs contacted with vdW Au/1T'-MoTe₂ (red) and 3D Pt (blue) at various temperatures. The inset shows the evolution of the I_{on}/I_{off} ratio with the measured T . As T decreases to ~ 178 K, the I_{on}/I_{off} ratio approaches $\sim 10^7$. **(b)** Two-terminal hole mobilities of 2H-MoTe₂ FETs with vdW Au/1T'-MoTe₂ (red), 3D Ti (green), and 3D Pt (blue) contact electrodes as a function of temperature. The temperature dependence of the mobility, $\mu \propto T^{-\gamma}$, indicates the dominant phonon scattering of the FET.

- **Revisions to manuscript:** We have revised the manuscript and Supplementary Information to ensure that the above-mentioned novelties are well presented as follows:
 - ① In the introduction, lines 28-29 on page 3 have been modified to highlight the growth scheme of the thin film with ultra-large single-crystalline domains, using the vertically introduced Te flux.
 - ② We have replaced Supplementary Table 1, and added a more detailed caption in Supplementary Table 1 for the fair comparison of the crystalline quality of the CVD-grown MoTe₂, displaying production rate, possible impurities or residues, and electrical properties.
 - ③ We have inserted lines 13-15 and 16-18 on page 6 and Figure 1j for novelty in terms of the high optical absorption coefficient of MoTe₂ in the NIR spectral range. We have also added the notes in Supplementary Fig. 7 to highlight the potential of MoTe₂ for various optoelectronic devices.
 - ④ Explanations of the MIT behavior of the channel conductivity at low T have been added to lines 16-17, 19-21, and 26-28 on page 15. We also added a column to Supplementary Table 1 to compare MIT's findings with other relevant literature.

(1) I agree with Reviewer #2 that MoTe₂ is not “one of the best p-type 2D transistor” materials in terms of on-current and mobility as the authors claim.

Response to Point #1: We also understand that the explanation of “one of the best p -type 2D transistors” looks overstated. However, compared to MoS₂ or WSe₂, the growth of tellurides is complex and less investigated, which may cause this material system to be underestimated

for its use in electrical or optoelectronic applications. Additionally, typical MoTe₂ metal contact electrodes exhibit Fermi-level pinning to the mid-gap with poor hole conductivity and high SBH values in the literatures. Several attempts have been made to obtain high-performance *p*-type MoTe₂ transistors by implementing phase-engineered 1T' contact electrodes through laser and local heating^{39,40}; nevertheless, these techniques have limited scalability in terms of laser resolution or thermal budget. In addition, the effect of the bulk 3D metal pad, which must be deposited on top of 1T'-MoTe₂, has not been carefully investigated, resulting in small μ_h and I_{on} values for the transistor.

To address this issue, we introduced Fermi-level-tuned vdW semimetal contact electrodes, which allowed us to tune the SBH. Moreover, the I_{on} in our MoTe₂ ($\sim 7.8 \pm 1.4 \mu\text{A} \cdot \mu\text{m}^{-1}$) is one of the best among those reported for 2D MoTe₂^{1,2,13,19,41-43} and WSe₂^{34,44-49} prepared by CVD (**Figure R4**), making it a promising channel candidate for high-performance *p*-type 2D transistors. In addition, most reports on *p*-type 2D semiconductors have relied on mechanically exfoliated or CVD-grown irregular flakes, which raises questions about their reproducibility and large-area applications as opposed to our study. All data points in **Figure R4** are based on irregularly synthesized flakes, except for the data points from MOCVD⁴⁹ and this study.

We also note that MoTe₂ can behave as a better *p*-type channel than other group-VI 2D TMDs in CMOS, given its band structure (**Figure R5**). Typical group-VI 2D TMDs other than 2H-MoTe₂ present a challenge in suppressing electron transport owing to their valence band maximum located at >5.0 eV. This results in a high SBH for holes (Φ_{hole} at V_{FB}) in these 2D TMDs, which limits the hole conductivity. In contrast, MoTe₂ has a lower SBH owing to its higher location of E_{VBM} , making its hole conductivity higher (**Figure R5a**). Notably, the ratio between Φ_{hole} and $\Phi_{electron}$ of MoTe₂ is also the smallest among the group-VI TMDs (**Figure R5c**), which implies that the possibility of hole transport is greater. Consequently, MoTe₂-based 2D transistors tend to exhibit *p*-type unipolarity (instead of ambipolarity) in their transfer characteristics, promising a low power-delay product per bit in CMOS owing to the smaller off-state currents in *p*-type MOS and greater efficiency in high-low and low-high transitions.

Figure R4. Benchmarking plots of the on-state current density (I_{on}) of our 2H-MoTe₂ transistors with different *p*-type 2D transistors based on CVD-grown WSe₂ and MoTe₂. (a) Comparison of the I_{on} values (at $V_{ds} = -1$ V) of synthetic 2H/1T'-MoTe₂ transistors in this study with reported values for *p*-type CVD-grown WSe₂^{34,44-49} dependent on channel L . Our data were fitted with the reciprocal relationship between I_{ds} and L (i.e., $I_{ds} \propto 1/L$). (b)

Extracted I_{on} of our 2H-MoTe₂ transistors with vdW Au/1T' contacts and their comparisons with reported values^{1,2,13,19,41-43} depending on the channel length (L). I_{on} values at a V_{ds} of -1 V are compared.

Figure R5. Estimation of hole injection efficiency depending on the 2D group-VI TMDs. (a) (Left) Band structures of group VI-TMDs⁵⁰ and (right) work functions (WFs) of conventional 3D metals (black) and vdW Au/1T'-MoTe₂ (red) used in this study. (b) Calculated thermionic barrier height for holes at the flat band voltage (Φ_B), depending on the valence band maximum (E_{VBM}) of each 2D semiconducting TMD. The estimation assumed the ideal Schottky-Mott model for a metal contact with a WF of ~ 5.0 eV. (c) The ratio between the Φ_B values for holes and electrons ($\Phi_{hole}/\Phi_{electron}$) as a function of E_{VBM} .

- **Revisions to manuscript:** To reflect the Reviewer's comment, and to highlight the smaller Schottky barrier height of MoTe₂ compared to those of other group-VI 2D TMDs, we have inserted lines 21-24 on page 12 and Supplementary Fig. 20. The relevant explanations have been added in the note of Supplementary Fig. 20.

(i) The estimated on-current approaching ~ 2.6 mA/ μm is inaccurate when the channel length is reduced to 5 nm due to the existence of the contact resistance. As a high-performance p-type semiconductor, it is critical to experimentally demonstrate the significant increase in on-current when the channel length is reduced to sub-100 nm.

Response to Point #1-i: Thank you for your comment. We agree that our estimation, conducted during the previous revision, did not consider the effects of R_c on I_{on} . For ultrashort-channel transistors in the on-state under a high applied V_g , the contribution of the channel resistance (R_{ch}) is negligible^{51,52}, which results in the total resistance (R_{on}) dictated only by the R_c , as shown in the following Equation⁵²:

$$R_{on} = R_{ch} + 2R_c \approx 2R_c \quad (\text{R2})$$

In our study, the R_c value at the interface between 1T' and 2H MoTe₂ was found to be ~ 0.7 k $\Omega \cdot \mu\text{m}$, which leads to the estimated I_{on} of ~ 0.71 mA/ μm at $V_{ds} = -1$ V for an ultrashort channel transistor. This I_{on} (~ 0.71 mA/ μm) is smaller than our initial calculated value (~ 2.6 mA/ μm); hence, we would like to retract our claim and therefore removed the claim from the text.

Nevertheless, the estimated I_{on} value of ~ 0.71 mA/ μm is still a high value, comparable to those reported for the p-type 2D transistors with ultrashort channels of black phosphorene^{53,54} and WSe₂⁵², as summarized in **Table R2**. Only one report⁵² exhibits higher I_{on} (~ 1.72 mA/ μm) than our estimated value at the comparable V_{ds} . Note that Ref.⁵² was published while our manuscript was under review; therefore, it should be disregarded from the evaluation at *Nature Communications* (please see the editorial, available on the website: <https://www.nature.com/articles/s41467-020-17817-x>).

Table R2. Comparisons of I_{on} in p-type 2D transistors with an ultrashort channel.

Fabrication or extraction method	2D channel	Metal contact	Channel length (nm)	I_{on} (mA/ μm)	Ref.
Photolithograph	MoTe ₂ (~ 5 nm)	vdW Au/1T'	2,000	0.008 at V_{ds} of -1.0 V	This study
Calculation			0	0.71 at V_{ds} of -1.0 V	
e-beam lithography	Black phosphorene (< 12.5 nm)	Ni/Au	100	1.2 at V_{ds} of -2.5 V	53
Two-times e-beam lithography + angle evaporation	Black phosphorene (< 10 nm)	Au	20	0.17 at V_{ds} of -0.1 V	54
Two-step epitaxial growth	WSe ₂ (bilayer)	VSe ₂	20	1.72 at V_{ds} of -1.0 V	52

We would like to highlight that the processes used in this study were based on photolithography with a limited line width of ~ 1.5 – 2.0 μm (in the system we used), which is not optimal for the fabrication of ultrashort channel transistors. We agree that the experimental demonstration of ultrashort channel p-type 2D transistors is a big issue in the society and it will be desirable to have an in-depth study on the issue. For the study, e-beam lithography and other

complicated processes are essentially required. We tried to fabricate ultrashort channel transistors using the above-mentioned processes during this limited revision period, however, the task was not completed because optimization of the fabrication process is costly and time consuming. We respectfully request that the Reviewer consider our situation. Instead, we would like to emphasize that our photolithography process yielded higher I_{on} ($\approx 7.8 \pm 1.4 \mu\text{A}/\mu\text{m}$) in the 2 μm -long channels than any other existing study on p -type MoTe_2 (**Figure R4b**). Furthermore, only three reports on “ p -type” 2D transistors with channels smaller than 100 nm have been published at this moment (**Table R2**), emphasizing that the failure of experimental demonstration during revision should not be used to discredit this study.

- **Revisions to manuscript:** To correct the estimation of the I_{on} of ultrashort channel MoTe_2 FET in the manuscript, we have modified line 14 on page 12.

(ii) The recently developed thermally evaporated Te layers (Nature Nanotech. 15, 53-58, 2020) outperform p-type \$\text{MoTe}_2\$ based on scalability, electronic performance, and synthesis temperature.

Response to Point #1-ii: Thank you for your comment. However, we respectfully hold a different view regarding the statement that Te as an ultrathin body film in Ref.⁵⁵ (the stated reference) is superior in terms of electronic performance, scalability, and processing accessibility compared to MoTe_2 in our study.

First, the reported Te film is polycrystalline, with domains sizes of $\sim 30 \mu\text{m}^2$ and no texture or oriented planes. For example, the XRD patterns exhibited the (100), (101), (102), (110), (111), (200), (201), and (202) planes (Supplementary Fig. 2 in Ref.⁵⁵), indicating that the material should be classified as a bulk 3D semiconductor rather than a 2D semiconductor. The use of such 3D materials in ultrathin body films may result in deteriorated carrier mobility and inadequate device performance owing to surface scattering and unsaturated dangling bonds.

Second, the reported field-effect mobility (μ_{FE}) and I_{on}/I_{off} ratio of Te films with a thickness of $\sim 5 \text{ nm}$ are $< 20 \text{ cm}^2\text{V}^{-1}\text{s}^{-1}$ and $\sim 10^4$, respectively (Fig. 2j in Ref.⁵⁵), which are smaller than those of our MoTe_2 transistors with the same channel thickness (μ_{FE} of $\sim 30 \text{ cm}^2\text{V}^{-1}\text{s}^{-1}$ and I_{on}/I_{off} ratio $> 10^5$).

Third, the 3D body nature and high dielectric constant of the Te film ($\sim 27-66$)^{56,57} cause short channel effects worse than the 2D MoTe_2 with a smaller dielectric constant (~ 20)⁵⁸, given the transistor characteristic length (λ) dictated by the dielectric constant of the semiconductor body (ϵ_b) as shown below⁵⁹:

$$\lambda = \sqrt{\frac{t_b t_{ox} \epsilon_b}{\epsilon_{ox}}}$$

where t_b , t_{ox} , and ϵ_b are the thicknesses and the dielectric constant of the semiconductor body, respectively. Therefore, the larger ϵ_b of Te limits its use in ultra-scaled devices compared with that of MoTe_2 .

Finally, in terms of process compatibility, the deposition of Te film described in Ref.⁵⁵ necessitates the use of cooled nitrogen gas for a target substrate in order to reduce the process temperature to -80 °C, making the process complicated and challenging to integrate.

(iii) I agree with Review #2 that the content and the background of the submitted work do not match. Synthesis of large-scale p-type 2D semiconductors at low temperatures targeting integrated circuits. However, the complex contact electrode preparation method is contradictory for this application, especially the involved transfer. The authors point out that this method can be used to drive micro-LEDs in manufacturing lines. However, a high on-state current is critical in this application but was not demonstrated experimentally.

Response to Point #1-iii: Thank you for your comment. We would like to mention that the stacking of 2D materials necessitates a transfer approach to utilize the ultraclean nature of the vdW contact interface, at least in this work. Nevertheless, there are some viable options for producing vdW contacts using the synthesis method instead of transfer, where the process includes the vertical growth of TMDs without any interfacial defects^{52,60-62}. We believe that the low-energy deposition of MoO_x or MoI₂ as precursors for 1T'-MoTe₂ is a possible solution for preparing ultraclean vdW contact electrodes in a synthetic manner devoid of performance degradations^{52,60}. The use of thermally degradable buffer layers^{61,62} is another option for realizing vdW 1T' MoTe₂ contacts.

However, it was not possible to implement ultraclean vdW contacts by direct synthesis during this short revision period. This task is also a very big issue in the society and it will be desirable to have an in-depth study on the issue. Nevertheless, we believe that the importance and novelty of both low-temperature growth for *p*-type 2D semiconductors and low-energy-transfer processes for high-performance transistors are sufficiently present in this study. The high-density fabrication of vdW-contacted MoTe₂ transistor arrays on a large area SiO₂/Si substrate (> 1 × 1 cm²; Fig. 4b) and their statistical analysis with histograms and distribution fits (Supplementary Figs. 14e-g) have demonstrated that our method is a feasible technique for large-area applications. We respectfully request that the reviewer consider this point in the manuscript.

Regarding the statement about micro-LEDs, we apologize for any misunderstandings in our previous communication. We would like to clarify that we intended to emphasize the availability of mass transfer used in the process for micro-LEDs. This technology is commercially available⁶³ and there are no significant limitations for the transfer process to be applied to 2D materials. Layer-transfer techniques for the monolithic integration of dissimilar materials have become promising for application in electronic devices⁶⁴.

- **Revisions to manuscript:** Following the Reviewer's comment, we have inserted Supplementary Note in Supplementary Fig. 22 to suggest some viable options to produce the van der Waals contacts using the synthesis method instead of transfer.

(2) The "2H-seed" growth method has been demonstrated in previous work (Science 372, 195-200, 2021). In the submitted work, the seed array is introduced to trigger multi-site growth, but

the manuscript does not introduce how the seed array is transferred. If the seed arrays do not have the same orientation, the resulting thin film is polycrystalline, which has no differences with multiple nucleation and growth. If the seed arrays have the same orientation, one should start with a large-scale single-crystalline thin film, a tricky goal in itself. Therefore, the authors stated “specifically, the use of the 2H seed patterns with the same crystal orientation enables the wafer-scale single-crystal film devoid of and GBs.”, which is only based on imagination without corresponding experimental results.

Response to Point #2: Thank you for your valuable comment. In this study, we have introduced abnormally grown 10-nm-thick 2H-MoTe₂ single crystalline domains as seed arrays, which were synthesized in the same manner as in Fig. 1. Although the thin film had large single-crystalline domains (several hundreds of micrometers), subsequent seed growth might produce single-crystalline MoTe₂ arrays of various crystal orientations due to multiple instances of nucleation in the film, as the Reviewer pointed it out. However, it is noted that this selective growth approach ensured the single-crystalline nature of each conformal MoTe₂ pattern as confirmed by STEM study in Fig. 3, even if the orientation direction of each domain varied. This fact indicates that our approach benefits electronic or optoelectronic devices that require high-quality single crystals after isolating individual crystals by etching away unnecessary areas. Our method for obtaining single-crystalline arrays is significant, as recent research published in *Nature* (Ref.⁶⁵, published when this paper was under review) also emphasizes the importance of growing single-crystal arrays with different crystal orientations on amorphous substrates using a non-epitaxial method.

In addition, to address the Reviewer’s comment, the initial seed layers were made of mechanically exfoliated single-crystalline flakes. The MoTe₂ mother crystal (obtained from 2D Semiconductors) had a single crystalline property over a large area (> ~5 mm). Different flakes transferred on the tape had the same crystal orientation when adhesive tape was simultaneously applied and removed from the crystal. This tape/flake sample was then transferred onto a preformed 1T'-MoTe₂ film (with a thickness of 20 nm), followed by seed growth and conversion to the 2H phase at 500 °C. The low-temperature growth at 500 °C allowed the suppression of random 2H nucleation during the seed growth process. The resulting fully grown 2H film was single-crystal in nature along the unidirectional crystalline orientation of the exfoliated seeds (at least across an area of ~2 × 2 mm², as shown in **Figure R6a-c**). The resulting MoTe₂ was patterned using photolithography for reuse as a seed array, as shown in **Figure R6a**. Inverse pole figure mapping (**Figure R6b**) and electron backscattered diffraction (EBSD) patterns (**Figure R6c**) confirmed that the conformally formed patterns were single crystals with (0001) texture and the same alignment or thin films with a very small-angle GB (< ± 1.34°). Moreover, TEM analysis showed that the newly grown “tr-2H” regions were all oriented in a similar direction, indicating the single-crystalline nature.

Our results imply that we have more room for developing wafer-scale single-crystal films by “2H-seed” engineering using CVD process at low temperatures (550 °C or less), that is, centimeter-scale or larger grain sizes in the resulting 2D films can be achieved by producing 2H seed patterns with the same crystal orientation sufficiently separated from one another with high controllability.

- **Revisions to manuscript:** We have provided a more detailed description of the synthesis conditions for the seed layers used in Figure 3 in the Methods section at page 20, lines 17-18. The advantages and possibilities of this growth mode are also discussed on page 10, lines 5-7. Additionally, we have presented the possibility of forming ultrasmall-angled GBs through seed layers with the same crystal orientation as those observed in Figure R6 on page 10, lines 7-14. Figure R6 has been included in the manuscript as Supplementary Figure 13.

Figure R6. Characterizations of the large-scale single-crystalline MoTe₂ achieved via seed growth. (a–c) Electron backscattered diffraction (EBSD) characterization of the 2H-seed patterns with the same crystalline orientation. (a) OM image of 2H-seed layers on the SiO₂/Si substrate. (b) Representative inverse pole figure map of the 2H-seed patterns along the direction normal to surface. The uniform red color indicates the 2H-MoTe₂ oriented along the [0001] direction. (c) EBSD patterns captured for the different structures in (a), demonstrating Kikuchi patterns of MoTe₂ aligned in similar orientations. Measurements were taken randomly for each pattern with ~500–1,000 μm separations in the order indicated by the arrow in (a), suggesting a single-crystalline nature across an area of ~2 × 2 mm². (d, e) TEM

characterization of the 2H-seed and newly transitioned area (tr-2H) of 2H-MoTe₂. (d) Low-magnified TEM image of 2H-seed (dashed rectangular) and tr-2H area surrounding the 2H-seed, transferred onto the TEM grid. (e) Corresponding SAED patterns captured from the tr-2H regions (positions 1–12) in (d) captured using the TEM aperture of ~40 μm. The slight difference in orientations of the patterns for the 10–12 regions may be attributed to the bending of the film induced by the TEM grid.

(3) Some interpretations are misleading.

(i) In the abstract, the authors claimed that they fabricated “4-inch single-crystalline 2H-MoTe₂ wafers”. From my scrutiny of the manuscript, I think the 4-inch wafer is from multiple nucleations, so in the submitted work, it should be polycrystalline.

Response to Point #3-i: We appreciate your insight regarding a specific point in the abstract. We have taken the necessary steps to revise the sentence to prevent it from being perceived as an exaggeration.

- **Revisions to manuscript:** Following the Reviewer’s comment, we have modified the abstract, lines 31-32 on page 1, as follows: “...we fabricated 4-inch 2H-MoTe₂ wafers with ultra-large single-crystalline domains and spatially-controlled single-crystalline arrays...” As part of the abstract revision process to limit the abstract to 150 words, we have revised lines 26, 27, 33, and 34 on page 1, as well as lines 1 and 2 on page 2 to avoid overemphasizing the findings presented in the manuscript.

(ii) The comparison in the work should be based on a fair comparison, especially in the introduction section, some updated data for MoTe₂ such as on-current and mobility should be provided, not just compared with poorer reported values.

Response to Point #3-ii: Thank you for your comment. We have thoroughly reviewed the updated data; and we found that we have excluded 6 references from the comparisons in our paper (which were published in 2021–2023)^{40,66-69}, as summarized in **Table R3**. We found that all the research results showed transistor performance with lower I_{on} and hole mobility (μ_h) than our MoTe₂ devices. We also reviewed the Introduction, but no significant revisions were required as no new or updated data on MoTe₂ transistors with high-work function 3D metal electrodes was found.

Table R3. Benchmarking table for 2H-MoTe₂-based FETs reported in the papers recently published in 2021–2023. We found six references that were not mentioned in the original manuscript.

Prep. method for 2H-MoTe ₂	Contact metal	I_{on}/I_{off} ratio	μ_h (cm ² V ⁻¹ s ⁻¹)	I_{on} at $V_{ds} = -1$ V (nA·μm ⁻¹)	Ref.
CVD	vdW Au/1T'-MoTe ₂	2.9×10^5	29.5	7,814	This study
CVD	N/A	~10 ⁴	N/A	~3.3	66

CVD	Pd/Au	3.9×10^3	6.7	N/A	67
CVD	Ti/Au	$\sim 10^{0.5}$	4.39	~ 180	68
CVD	Pd	10^5	1.0	41	69
Mechanical exfoliation	1T'-enriched MoTe ₂ (annealing)	1.32×10^3	N/A	~ 5	40

- **Revisions to manuscript:** In response to the Reviewer's suggestion, we have modified lines 11 and 12 on page 3 to specify that MoTe₂ transistors with 3D metal contacts are limited to an I_{on}/I_{off} of 10^4 and an I_{on} of 1 μ A/ μ m at $V_{ds} = -1$ V.

(iii) In conclusion, a contact resistance value of 0.7 $k\Omega \cdot \mu\text{m}$ suddenly appeared (for the first time), much lower than the value reported in the main text.

Response to Point #3-iii: Thank you for your comment. We would like to clarify that the reported R_c values of 1T'-MoTe₂/2H-MoTe₂ MSJs in Refs.^{2,8,11,19} include both the R_c of 3D-metal-pads/1T'-MoTe₂ and 1T'/2H-MoTe₂ interfaces, which do not strictly represent the contact property of pure MSJ. This has already been stated in the main text on page 13 as follows: “Because the R_c of this MSJ system included the R_c of the 3D-metal-pad/1T'-MoTe₂ interface (~ 0.5 $k\Omega \cdot \mu\text{m}$; Supplementary Fig. 23c), the actual R_c at between the 2H- and 1T'-MoTe₂ approached $\sim 0.7 \pm 0.5$ $k\Omega \cdot \mu\text{m}$ at n_{2D} of $\sim 7.4 \times 10^{12}$ cm^{-2} .”

We also acknowledge that this may be a bit confusing because different values were used for the Abstract and Introduction; therefore, we have corrected them.

- **Revisions to manuscript:** We have made corrections to the minimum contact resistance value in the abstract and introduction, specifically on line 36 of page 1 and line 7 of page 4, to reflect the R_c value of ~ 0.7 $k\Omega \cdot \mu\text{m}$. On the other hand, we have retained the description of how the R_c was extracted on page 13. To clearly distinguish the “TLM-derived $2R_c$ ” from the “actual R_c at the semimetal/semiconductor interface”, we express the TLM-derived one as $2R_c$ on page 13, lines 28-32, and page 14, line 6.

(iv) Supplementary Table 1 is inappropriate because the 4-inch thin film is not based on the bilayer and no device characterization has been performed on bilayer MoTe₂.

Response to Point #3-iv: Thank you for your comment. We found that the content added during the revision process was exaggerated. Therefore, we removed some columns and added more notes regarding the comparisons.

- **Revisions to manuscript:** To avoid unnecessary exaggeration, we have excluded information on the thickness of the synthesized 2H-MoTe₂ in Supplementary Table 1.

(v) In the response, the comparison to the growth time of previous work is unfair. The previous work was grown from single nuclei, while the submitted work was nucleated from multi nuclei, growth, and collapse into a continuous 4-inch continuous 2H MoTe₂ thin film.

Response to Point #3-v: We appreciate the valuable comments regarding the comparison. However, we do not believe that this is entirely unfair because the comparisons are directly related to the throughput of obtaining a fully connected 2H-MoTe₂ thin film on a wafer scale.

Although the mentioned study (published in *Science*⁸), presents an interesting aspect of single-crystal formation via single nuclei, the electrical conductivity of the resulting MoTe₂ is approximately 10 times lower than our reported values. A fast growth rate for a large area and strong electrical features are significant parameters for the growth of thin films. Our method for obtaining a 4-inch continuous MoTe₂ thin film has the advantages of a faster growth rate and higher quality. We also note that the compared studies^{8,10-15} except Ref.⁸ are based on thin films synthesized from multiple nuclei.

- **Revisions to manuscript:** In the revised manuscript, we have utilized the term 'production rate' to refer to the rapid production of the large-area substrates presented in this study, and we have distinguished this from the 'growth rate,' which pertains to the synthesis growth rate of single crystal domains. In response to this rationale, we have modified line 2 on page 5 and have included a definition of the production rate, as well as comparisons with other studies on the CVD-grown MoTe₂, in Supplementary Table 1.

(4) On page 4, the authors stated “%(Te/Mo)~2.0 and 1.9 for 1T’ and 2H phases, respectively”, which I think may be a mistake, since the authors state elsewhere “Te-deficient environment to form from the 1T’ structure”.

Response to Point #4: Thank you for your comment. There was no clear correlation between the XPS and ambient Te flux during growth. The accuracy of XPS atomic concentration is ± 4–15 % depending on materials⁷⁰, indicating that the percentage (Te/Mo) of as-grown MoTe₂ crystals falls within the expected range for an ideal stoichiometry.

(5) I’m curious whether the Te supply method or the confined environment promotes the abnormal growth of 2H phase grains. A control experiment can be designed to determine whether the abnormal 2H phase grain growth can occur at low temperatures using a face-to-face (use a bare substrate without NixTey facing a Mo film) structure to provide a confined environment by using a Te powder as the precursor.

Response to Point #5: Thank you for your interesting suggestion. To reflect the Reviewer’s suggestion, we investigated the abnormal grain growth of the 2H phase under different synthesis conditions for three different cases (A, B, and C). The growth in all three cases was conducted using the same Mo precursor (~3 nm) at the same temperature of ~650 °C, and schematic diagrams of each process are depicted in **Figures R8a, d, and g**.

The color uniformity of the as-grown MoTe₂ confirmed that our method was suitable for obtaining uniform films, as shown in Case A (**Figure R8b**). In addition, circular-shaped 2H domains of approximately 30–100 μm with a coverage of ~31.6 % (over 1 × 1 cm² substrate) were evenly distributed inside the 1T' thin film in Case A, indicating abnormal grain growth (**Figure R8c**). In contrast, in Case B, little 2H phase nucleation occurred, with a 2H coverage of ~0.004%, implying that the supplied Te was insufficient for 2H nucleation to occur (**Figure R8f**). In Case C, grain growth in the 2H phase larger than several micrometers did not occur (**Figure R8i**). The 1T' and 2H phases in the thin film were intermixed, and their densities varied across the film, presumably because of the non-uniform Te flux (as seen by the ambiguous optical contrast in **Figures R8h, i**). Many regions had polycrystalline 2H-MoTe₂ domains, which were very small (< 1 μm), indicating that grain growth was inhibited.

- **Revisions to manuscript:** Following the Reviewer's suggestion, we conducted additional experiments to confirm the advantages of our work, particularly the uniformity and 2H coverage, in comparison to the powder-based CVD. The results have been presented in lines 16-21 on page 8, and we have inserted Figure R8 as Supplementary Fig. 11. Additionally, we have included the relevant discussions in the notes of Supplementary Fig. 11.

Figure R8. Comparison of the abnormal grain growth of the 2H phase using different growth schemes. (a–c) Case A, which involves stacked Mo and Ni_xTe_y substrates for the vertical introduction of Te (used in this study); (d–f) Case B, which is horizontal CVD using the Te powder (1,000 mg) to introduce Te to the Mo film; and (g–i) Case C, which uses a gas-confined reactor prepared with a Mo film covered with the SiO₂/Si substrate and Te supplied using the powder (100 mg). (a, d, and g) Schematics of the different growth processes for Cases A–C, respectively, where the thickness of the red arrows indicates the extent of the Te flux. (b, e, and h) Optical images of the samples grown using (b) Case A, (e) B, and (h) C. (c, f, and i) Corresponding OM images taken at the center of each sample. The inset on the top right corner of (f) shows the locally transformed 2H region (scale bar: 20 μm), even though most areas of Case B consisted of the 1T' structure. The small dark regions in (i) indicate the local 1T'-MoTe₂.

(6) How did the authors state that “Pt is widely known for its exceptionally high WF of ~6.2 eV”? The Pt work function from the wiki reports a value of 5.12-5.93 eV.

Response to Point #6: Thank you for your valuable comment. It should be corrected to ~5.64 eV for polycrystalline Pt⁷¹.

- **Revisions to manuscript:** Following the Reviewer’s comment, we have corrected the WF value from ~6.2 to ~5.64 eV on line 20 of page 11.

References used in this response

- 1 Ma, R. *et al.* MoTe₂ Lateral Homojunction Field-Effect Transistors Fabricated using Flux-Controlled Phase Engineering. *ACS Nano* **13**, 8035-8046 (2019).
- 2 Sung, J. H. *et al.* Coplanar semiconductor–metal circuitry defined on few-layer MoTe₂ via polymorphic heteroepitaxy. *Nat. Nanotech.* **12**, 1064 (2017).
- 3 Kim, C. *et al.* Fermi Level Pinning at Electrical Metal Contacts of Monolayer Molybdenum Dichalcogenides. *ACS Nano* **11**, 1588-1596 (2017).
- 4 Haratipour, N. & Koester, S. Multi-layer MoTe₂ p-channel MOSFETs with high drive current. *72nd Dev. Res. Conf.*, 171-172 (2014).
- 5 Townsend, N. J., Amit, I., Craciun, M. F. & Russo, S. Sub 20 meV Schottky barriers in metal/MoTe₂ junctions. *2D Mater.* **5**, 025023 (2018).
- 6 Aftab, S. *et al.* Formation of an MoTe₂ based Schottky junction employing ultra-low and high resistive metal contacts. *RSC Adv.* **9**, 10017-10023 (2019).
- 7 Yin, L. *et al.* Ultrahigh sensitive MoTe₂ phototransistors driven by carrier tunneling. *Appl. Phys. Lett.* **108**, 043503 (2016).
- 8 Xu, X. *et al.* Seeded 2D epitaxy of large-area single-crystal films of the van der Waals semiconductor 2H MoTe₂. *Science* **372**, 195-200 (2021).
- 9 Xu, X. *et al.* Millimeter-Scale Single-Crystalline Semiconducting MoTe₂ via Solid-to-Solid Phase Transformation. *J. Am. Chem. Soc.* **141**, 2128-2134 (2019).
- 10 Zhang, Q. *et al.* Simultaneous synthesis and integration of two-dimensional electronic components. *Nat. Electron.* **2**, 164-170 (2019).
- 11 Xu, X. *et al.* Scaling-up Atomically Thin Coplanar Semiconductor–Metal Circuitry via Phase Engineered Chemical Assembly. *Nano Lett.* **19**, 6845-6852 (2019).
- 12 Kim, T. *et al.* Wafer-Scale Epitaxial 1T', 1T'–2H Mixed, and 2H Phases MoTe₂ Thin Films Grown by Metal–Organic Chemical Vapor Deposition. *Adv. Mater. Interfaces* **5**, 1800439 (2018).
- 13 Zhang, X. *et al.* Low Contact Barrier in 2H/1T' MoTe₂ In-Plane Heterostructure Synthesized by Chemical Vapor Deposition. *ACS Appl. Mater. Interfaces* **11**, 12777-12785 (2019).
- 14 Zhou, L. *et al.* Large-Area Synthesis of High-Quality Uniform Few-Layer MoTe₂. *J. Am. Chem. Soc.* **137**, 11892-11895 (2015).
- 15 Yang, L. *et al.* Tellurization Velocity-Dependent Metallic–Semiconducting–Metallic Phase Evolution in Chemical Vapor Deposition Growth of Large-Area, Few-Layer MoTe₂. *ACS Nano* **11**, 1964-1972 (2017).
- 16 Kusama, T. *et al.* Ultra-large single crystals by abnormal grain growth. *Nat. Commun.* **8**, 354 (2017).
- 17 Yamamoto, T. & Sakuma, T. Fabrication of Barium Titanate Single Crystals by Solid-State Grain Growth. *J. Am. Ceram. Soc.* **77**, 1107-1109 (1994).

- 18 Messing, G. L. *et al.* Templated Grain Growth of Textured Piezoelectric Ceramics. *Crit. Rev. Solid State Mater. Sci.* **29**, 45-96 (2004).
- 19 Yang, S. *et al.* Large-Scale Vertical 1T'/2H MoTe₂ Nanosheet-Based Heterostructures for Low Contact Resistance Transistors. *ACS Appl. Nano Mater.* **3**, 10411-10417 (2020).
- 20 Zheng, X. *et al.* Enormous enhancement in electrical performance of few-layered MoTe₂ due to Schottky barrier reduction induced by ultraviolet ozone treatment. *Nano Res.* **13**, 952-958 (2020).
- 21 Cho, Y. *et al.* Fully Transparent p-MoTe₂ 2D Transistors Using Ultrathin MoO_x/Pt Contact Media for Indium-Tin-Oxide Source/Drain. *Adv. Funct. Mater.* **28**, 1801204 (2018).
- 22 Sun, Y. *et al.* Interface-mediated noble metal deposition on transition metal dichalcogenide nanostructures. *Nat. Chem.* **12**, 284-293 (2020).
- 23 Zhang, C. *et al.* Systematic study of electronic structure and band alignment of monolayer transition metal dichalcogenides in Van der Waals heterostructures. *2D Mater.* **4**, 015026 (2016).
- 24 He, Q. *et al.* Molecular Beam Epitaxy Scalable Growth of Wafer-Scale Continuous Semiconducting Monolayer MoTe₂ on Inert Amorphous Dielectrics. *Adv. Mater.* **31**, 1901578 (2019).
- 25 Diaz, H. C., Ma, Y., Chaghi, R. & Batzill, M. High density of (pseudo) periodic twin-grain boundaries in molecular beam epitaxy-grown van der Waals heterostructure: MoTe₂/MoS₂. *Appl. Phys. Lett.* **108**, 191606 (2016).
- 26 Park, Y. J., Katiyar, A. K., Hoang, A. T. & Ahn, J.-H. Controllable P- and N-Type Conversion of MoTe₂ via Oxide Interfacial Layer for Logic Circuits. *Small* **15**, 1901772 (2019).
- 27 Munkhbat, B., Wróbel, P., Antosiewicz, T. J. & Shegai, T. O. Optical Constants of Several Multilayer Transition Metal Dichalcogenides Measured by Spectroscopic Ellipsometry in the 300–1700 nm Range: High Index, Anisotropy, and Hyperbolicity. *ACS Photonics* **9**, 2398-2407 (2022).
- 28 Green, M. A. Self-consistent optical parameters of intrinsic silicon at 300K including temperature coefficients. *Solar Energy Mater. Sol. Cells* **92**, 1305-1310 (2008).
- 29 Amotchkina, T., Trubetskoy, M., Hahner, D. & Pervak, V. Characterization of e-beam evaporated Ge, YbF₃, ZnS, and LaF₃ thin films for laser-oriented coatings. *Appl. Opt.* **59**, A40-A47 (2020).
- 30 Ruppert, C., Aslan, B. & Heinz, T. F. Optical Properties and Band Gap of Single- and Few-Layer MoTe₂ Crystals. *Nano Lett.* **14**, 6231-6236 (2014).
- 31 Liu, M., Liu, W. & Wei, Z. MoTe₂ Saturable Absorber With High Modulation Depth for Erbium-Doped Fiber Laser. *J. Light. Technol.* **37**, 3100-3105 (2019).
- 32 Cao, H. *et al.* Efficient and Fast All-Optical Modulator with In Situ Grown MoTe₂ Nanosheets on Silicon. *ACS Applied Nano Mater.* **6**, 838-845 (2023).

- 33 Bie, Y.-Q. *et al.* A MoTe₂-based light-emitting diode and photodetector for silicon photonic integrated circuits. *Nature Nanotechnol.* **12**, 1124-1129 (2017).
- 34 Wang, Y. *et al.* P-type electrical contacts for 2D transition-metal dichalcogenides. *Nature* **610**, 61-66 (2022).
- 35 Chuang, H.-J. *et al.* Low-Resistance 2D/2D Ohmic Contacts: A Universal Approach to High-Performance WSe₂, MoS₂, and MoSe₂ Transistors. *Nano Lett.* **16**, 1896-1902 (2016).
- 36 Mleczko, M. J. *et al.* Contact Engineering High-Performance n-Type MoTe₂ Transistors. *Nano Lett.* **19**, 6352-6362 (2019).
- 37 Cho, S. *et al.* Phase patterning for ohmic homojunction contact in MoTe₂. *Science* **349**, 625-628 (2015).
- 38 Sung, J. H. *et al.* Coplanar semiconductor–metal circuitry defined on few-layer MoTe₂ via polymorphic heteroepitaxy. *Nat. Nanotechnol.* **12**, 1064-1070 (2017).
- 39 Ryu, H. *et al.* Laser-Induced Phase Transition and Patterning of hBN-Encapsulated MoTe₂. *Small* **19**, 2205224 (2023).
- 40 Wang, Y. *et al.* Atomistic Observation of the Local Phase Transition in MoTe₂ for Application in Homojunction Photodetectors. *Small* **18**, 2200913 (2022).
- 41 Pradhan, N. R. *et al.* Field-effect transistors based on few-layered α -MoTe₂. *ACS Nano* **8**, 5911-5920 (2014).
- 42 Lee, R. S. *et al.* van der Waals Epitaxy of High-Mobility Polymorphic Structure of Mo₆Te₆ Nanoplates/MoTe₂ Atomic Layers with Low Schottky Barrier Height. *ACS Nano* **13**, 642-648 (2019).
- 43 Choi, D. *et al.* Directly grown Te nanowire electrodes and soft plasma etching for high-performance MoTe₂ field-effect transistors. *Appl. Surf. Sci.* **565**, 150521 (2021).
- 44 Vu, V. T. *et al.* One-Step Synthesis of NbSe₂/Nb-Doped-WSe₂ Metal/Doped-Semiconductor van der Waals Heterostructures for Doping Controlled Ohmic Contact. *ACS Nano* **15**, 13031-13040 (2021).
- 45 Li, S. *et al.* Tunable Doping of Rhenium and Vanadium into Transition Metal Dichalcogenides for Two-Dimensional Electronics. *Adv. Sci.* **8**, 2004438 (2021).
- 46 Gao, Y. *et al.* Ultrafast Growth of High-Quality Monolayer WSe₂ on Au. *Adv. Mater.* **29**, 1700990 (2017).
- 47 Tang, H.-L. *et al.* Multilayer Graphene–WSe₂ Heterostructures for WSe₂ Transistors. *ACS Nano* **11**, 12817-12823 (2017).
- 48 Chu, C.-H. *et al.* End-Bonded Metal Contacts on WSe₂ Field-Effect Transistors. *ACS Nano* **13**, 8146-8154 (2019).
- 49 Kozhakhmetov, A. *et al.* Scalable BEOL compatible 2D tungsten diselenide. *2D Mater.* **7**, 015029 (2020).

- 50 Liu, Y., Stradins, P. & Wei, S.-H. Van der Waals metal-semiconductor junction: Weak Fermi level pinning enables effective tuning of Schottky barrier. *Sci. Adv.* **2**, e1600069 (2016).
- 51 English, C. D., Shine, G., Dorgan, V. E., Saraswat, K. C. & Pop, E. Improved Contacts to MoS₂ Transistors by Ultra-High Vacuum Metal Deposition. *Nano Lett.* **16**, 3824-3830 (2016).
- 52 Wu, R. *et al.* Bilayer tungsten diselenide transistors with on-state currents exceeding 1.5 milliamperes per micrometre. *Nat. Electron.* **5**, 497-504 (2022).
- 53 Li, X. *et al.* High-speed black phosphorus field-effect transistors approaching ballistic limit. *Sci. Adv.* **5**, eaau3194 (2019).
- 54 Miao, J., Zhang, S., Cai, L., Scherr, M. & Wang, C. Ultrashort Channel Length Black Phosphorus Field-Effect Transistors. *ACS Nano* **9**, 9236-9243 (2015).
- 55 Zhao, C. *et al.* Evaporated tellurium thin films for p-type field-effect transistors and circuits. *Nat. Nanotechnol.* **15**, 53-58 (2020).
- 56 Young, K. F. & Frederikse, H. P. R. Compilation of the Static Dielectric Constant of Inorganic Solids. *J. Phys. Chem. Ref. Data* **2**, 313-410 (1973).
- 57 Balasubramaniam, T., Narayandass, S. K. & Mangalaraj, D. Electrical properties of thermally evaporated tellurium thin films. *Bull. Mater. Sci.* **20**, 79-92 (1997).
- 58 Laturia, A., Van de Put, M. L. & Vandenberghe, W. G. Dielectric properties of hexagonal boron nitride and transition metal dichalcogenides: from monolayer to bulk. *npj 2D Mater. Appl.* **2**, 6 (2018).
- 59 Liu, Y. *et al.* Promises and prospects of two-dimensional transistors. *Nature* **591**, 43-53 (2021).
- 60 Zhang, K. *et al.* Epitaxial substitution of metal iodides for low-temperature growth of two-dimensional metal chalcogenides. *Nat. Nanotechnol.*, in press, doi:10.1038/s41565-023-01326-1 (2023).
- 61 Kwon, G. *et al.* Interaction- and defect-free van der Waals contacts between metals and two-dimensional semiconductors. *Nat. Electron.* **5**, 241-247 (2022).
- 62 Kong, L. *et al.* Wafer-scale and universal van der Waals metal semiconductor contact. *Nat. Commun.* **14**, 1014 (2023).
- 63 Li, J., Luo, B. & Liu, Z. in *2020 21st International Conference on Electronic Packaging Technology (ICEPT)*.
- 64 Kum, H. *et al.* Epitaxial growth and layer-transfer techniques for heterogeneous integration of materials for electronic and photonic devices. *Nat. Electron.* **2**, 439-450 (2019).
- 65 Kim, K. S. *et al.* Non-epitaxial single-crystal 2D material growth by geometric confinement. *Nature* **614**, 88-94 (2023).
- 66 Pan, Y. *et al.* Direct Multitier Synthesis of Two-Dimensional Semiconductor 2H-MoTe₂. *ACS Appl. Electron. Mater.* **4**, 5733-5738 (2022).

- 67 Jia, X. *et al.* High-Performance CMOS Inverter Array with Monolithic 3D Architecture Based on CVD-Grown n-MoS₂ and p-MoTe₂. *Small in press*, 2207927, doi:<https://doi.org/10.1002/sml.202207927> (2023).
- 68 Li, J. *et al.* Controllable growth of large-area 1T', 2H ultrathin MoTe₂ films, and 1T'–2H in-plane homojunction. *J. Appl. Phys.* **131**, 185302 (2022).
- 69 Lin, C.-P. *et al.* Two-dimensional solid-phase crystallization toward centimeter-scale monocrystalline layered MoTe₂ via two-step annealing. *J. Mater. Chem. C* **9**, 15566–15576 (2021).
- 70 Brundle, C. R. & Crist, B. V. X-ray photoelectron spectroscopy: A perspective on quantitation accuracy for composition analysis of homogeneous materials. *J. Vac. Sci. Technol. A* **38**, 041001 (2020).
- 71 Haynes, W. M. *CRC handbook of chemistry and physics*. (CRC press, 2016).

REVIEWERS' COMMENTS

Reviewer #1 (Remarks to the Author):

I am delighted to see the significant improvements made in this manuscript. Based on my overall evaluation, I recommend the publication of this paper in Nature Communications.

However, I must acknowledge that there are areas for improvement regarding the benchmarks presented in this paper. Reviewer #4 raised a valid concern about potential unfair comparisons between different papers due to variations in device conditions. The "device dependent" quantities (where device geometries and configuration matters) and "materials dependent" quantities (which only depends on material itself) should be specifically pointed out in the benchmarks. Nevertheless, by comparing the "best performance" data points across various papers, this study demonstrates high I_{on} currents, on/off ratios, and low Schottky barriers among MoTe₂-based devices.

The central focus of this paper remains justified as it presents a novel method for producing large domain 2H-MoTe₂ samples. The authors achieve this by strategically placing a significant amount of Te, which is uniformly distributed and in close proximity to the target wafer.

The authors effectively demonstrate the high quality of their MoTe₂ samples through the following observations:

1. There exists a strong correlation between the crystal orientation of the "seed" and the post-transition 2H-MoTe₂ (tr-2H-MoTe₂) as depicted in Figure 3e and Supplementary Figure 13. By comparing the orientations of diffraction patterns at various locations, the authors have proved the same crystal orientations across the whole sample. One caveat is that this method doesn't pick up domain boundaries which separate the crystals with the same orientation (e.g. when two crystals with the same orientation grow toward each other, there might still be a 1D defect if there is a misalignment). The authors might just need to mention it in the paper.

2. Compared to other studies, the quality of the tr-2H-MoTe₂ stands out. This outcome aligns with their approach of compensating for the most common defect in MoTe₂, namely, Te vacancies, by supplying an excessive amount of Te.

With all the revisions done at the current stage, I believe the paper is in a good shape to be published.

Reviewer #3 (Remarks to the Author):

[Reviewer #3 only provided confidential comments to the editor.]

Reviewer #4 (Remarks to the Author):

I thank the authors for their careful revisions and removed over-the-top claims such as the extremely high on-current at the 5 nm channel limit. However, after scrutiny of the revision and response letter, I still find that the manuscript lacks the novelty to be published in a well-known journal like Nature Communications, and most of my previous concerns have not been well addressed.

(1) For the 4-inch wafer sample, which was grown at 700 °C, the nucleation and growth kinetics resulted in polycrystalline thin films with grain sizes comparable to those previously reported. Moreover, this method has been used to synthesize 2H-MoTe₂ in the authors' previous paper (Nature Electronics 3, 207-215, 2020). The so-called "abnormal grain growth" is the same mechanism as the "solid-to-solid phase transition", and the atomic rearrangement and recrystallization processes have been studied.

(2) Low-temperature growth at 500 °C was used only for seed-induced growth. However, seed arrays start with thin films grown at 700 °C.

(i) Single-crystal arrays are limited by the size of the single-crystal domains grown at 700 °C.

(ii) How did the authors control the nucleation sites to achieve ~5 mm-sized single crystal thin films as seed films? From all images presented in the manuscript and supplemental information, there are multiple nuclei on a chip-sized substrate.

(iii) If we already have a single crystal thin film of large size, what is the purpose to etch it into an array as a seed film, perform secondary growth, etch away the unwanted area (most likely the seed area because they are thicker), leaving a discontinuous film? In other words, is not the original 100% coverage seed film better than the array film that has undergone a complex process?

(3) I still think the comparison between production rate and growth rate is unfair. The production rate defined by the authors is the size of the substrate divided by the time required to achieve 100% 2H-MoTe₂ coverage. The growth rate is defined by the 2H-MoTe₂ domain expansion rate. From all the optical images presented in the manuscript and supplementary information, the method presented in this manuscript has similar growth rates to the previous reports. Furthermore, previously reported results would also yield a similar production rate for multiple nucleation, since it depends on the growth substrate size.

(4) On-current and on-off ratio are device parameters and should be compared under the same device geometry and measurement conditions. Mobility is a more intrinsic property related to film quality (should be Hall mobility, since field effect mobility is still a device parameter). Under the same device geometry, carrier concentration, and applied electrical field, the higher the mobility, the larger the on-current. Although the comparison between the calculated on-current at 0 channel length and the experimental results with finite channel length is unfair, 2H-MoTe₂ still provides a much smaller on-current (Table R2).

(5) In Fig. 25a, the on-current does not “increase significantly as the T decreased from ~338 K to ~218 K”. Furthermore, it is inappropriate to study the electron-phonon scattering using field-effect mobility, since it is largely affected by contact, and Hall mobility should be used instead.

Overall Response to Reviewers' Comments:

We express our sincere appreciation to the Reviewers for their insightful comments regarding our manuscript, "*Fabrication of p-type 2D single-crystalline transistor arrays with Fermi-level-tuned van der Waals semimetal electrodes*" Their input has been instrumental in enhancing the overall quality of our work. We have thoroughly revised our manuscript in light of the Reviewers' feedback and are confident that this revision will satisfy the reviewer's concerns. For ease of tracking, we have numbered the comments of the Reviewers.

Reviewer #1's Comments (blue) Followed by Our Response (black):

I am delighted to see the significant improvements made in this manuscript. Based on my overall evaluation, I recommend the publication of this paper in Nature Communications.

However, I must acknowledge that there are areas for improvement regarding the benchmarks presented in this paper. Reviewer #4 raised a valid concern about potential unfair comparisons between different papers due to variations in device conditions. The "device dependent" quantities (where device geometries and configuration matters) and "materials dependent" quantities (which only depends on material itself) should be specifically pointed out in the benchmarks. Nevertheless, by comparing the "best performance" data points across various papers, this study demonstrates high I_{on} currents, on/off ratios, and low Schottky barriers among MoTe₂-based devices.

The central focus of this paper remains justified as it presents a novel method for producing large domain 2H-MoTe₂ samples. The authors achieve this by strategically placing a significant amount of Te, which is uniformly distributed and in close proximity to the target wafer.

The authors effectively demonstrate the high quality of their MoTe₂ samples through the following observations:

1. There exists a strong correlation between the crystal orientation of the "seed" and the post-transition 2H-MoTe₂ (tr-2H-MoTe₂) as depicted in Figure 3e and Supplementary Figure 13. By comparing the orientations of diffraction patterns at various locations, the authors have proved the same crystal orientations across the whole sample. One caveat is that this method doesn't pick up domain boundaries which separate the crystals with the same orientation (e.g. when two crystals with the same orientation grow toward each other, there might still be a 1D defect if there is a misalignment). The authors might just need to mention it in the paper.
2. Compared to other studies, the quality of the tr-2H-MoTe₂ stands out. This outcome aligns with their approach of compensating for the most common defect in MoTe₂, namely, Te vacancies, by supplying an excessive amount of Te.

With all the revisions done at the current stage, I believe the paper is in a good shape to be published.

Response to Reviewer #1's comment: We express our sincere gratitude for your recommendation to publish this study. The suggestions provided by the reviewer greatly contributed to the preparation of a high-quality manuscript.

In response to the reviewer's feedback, we agree with the reviewer that it is important to clarify the influence of device geometry on the behavior of two-terminal field-effect transistors (FETs). We acknowledge that our previous Supplementary Table 1 might unintentionally imply that the electrical properties are entirely determined by the material properties after reviewing our manuscript based on the reviewer's suggestions. Therefore, we intend to address this issue by introducing a two-terminal on-state total sheet conductance ($G_{on,tot}$ in the unit of μS), which is normalized by the channel length (L) and width (W), as described by the following equation:

$$G_{on,tot} = \frac{I_{on}}{V_{ds}} \frac{L}{W}$$

By comparing $G_{on,tot}$ values, the potential underestimation of I_{on} values owing to variations in channel dimensions can be minimized, enabling fair comparison. Our 2H-MoTe₂ FETs, contacted with the Fermi-level-tuned 2D semimetal, exhibit the highest $G_{on,tot}$ value ($\sim 15.6 \mu\text{S}$) among large-scale 2H-MoTe₂ thin films grown using chemical vapor deposition (CVD) techniques¹⁻⁸ (in the range of approximately 4.5×10^{-4} to $9.8 \mu\text{S}$) (**Table R1**). In addition, we note that FETs¹⁻⁸ in **Table R1** employ a bottom gate of SiO₂ (except for Ref. ¹, which uses a HfO₂ bottom gate); hence, the effects of the gate capacitance and MoTe₂/dielectric interface on the $G_{on,tot}$ and on-to-off current ratios (I_{on}/I_{off} ratios) are comparable. We also note that short channel effects on FETs in Refs¹⁻⁸ are minimal because their channel lengths are larger than $\sim 2 \mu\text{m}$.

Moreover, we intend to eliminate the column in the revised table (**Table R1**) that compares the metal-to-insulator transition (MIT) behaviors of the FETs¹⁻⁸, which was presented in the previous Supplementary Table 1. This decision is driven by the understanding that MIT characteristics can be influenced by the material properties and also by the contact resistance of the device. We may prevent potential misinterpretations and ensure clarity in the data representation by deleting this column.

Table R1. Comparison of the CVD techniques for the growth of 2H-MoTe₂ thin films with dimensions larger than $\sim 0.5 \times 0.5 \text{ cm}^2$. Except in this study and Refs.^{3,5}, the thin films were grown using powder-based horizontal CVD^{1,2,4,6-8}. The production rate in the unit of [mm/h] is determined based on the lateral dimensions of the wafer [mm] and the time required [h], which should be distinguished from a “growth rate” of a single-crystalline domain⁸. The on-state total sheet conductance ($G_{\text{on,tot}} = (I_{\text{on}}/V_{\text{ds}})(L/W)$ in the unit of [μS]) of 2H-MoTe₂ is calculated when the transistor exhibits its maximum current. The reported FETs¹⁻⁸ utilizes the bottom gate of SiO₂ (except Ref.¹ which employs HfO₂ bottom gate); therefore, the effects of capacitance or MoTe₂/dielectric interface on the $G_{\text{on,tot}}$, and $I_{\text{on}}/I_{\text{off}}$ ratio values are similar. By comparing $G_{\text{on,tot}}$ values, the potential underestimation of I_{on} values owing to variations in channel dimensions can be minimized, enabling fair comparison.

Growth aspects		Electrical properties			Ref.
Production rate (mm/h)	Possible impurities or residues	Contact metal	$G_{\text{on,tot}}$ (μS)	$I_{\text{on}}/I_{\text{off}}$ ratio	
50.5	No	vdW Au/1T'	15.6	$> 10^5$	This study
10	No	1T' edge contact	9.75	$\sim 10^3$	1
7	No	1T' edge contact	8	$\sim 10^4$	2
61.5	1T' residues in Raman spectrum	Pd	4.5×10^{-4}	$\sim 10^4$	3
6.5	Oxide peaks in XPS	Ti/Au	1.3×10^{-2}	$\sim 10^2$	4
0.35	Required to remove Al ₂ O ₃ layer	1T' edge contact	5	$\sim 10^4$	5
18	Oxide peaks in XPS	Ti/Au	0.91	$\sim 10^3$	6
5	No	No transfer curve measurements			7
0.49 at $T = 620 \text{ }^\circ\text{C}$	Oxide peaks in XPS	1T' edge contact	6.9	$\sim 10^4$	8

A comparison of the normalized $G_{\text{on,tot}}$ values in **Table R1** provides a fair benchmark for 2D channels because it eliminates the dependence on the channel width and length. However, even after normalization, the $G_{\text{on,tot}}$ value can still be affected by poor contact interface, resulting in degradation. Therefore, it is crucial to compare the channel conductance while excluding the contribution of the contact resistance. In our manuscript, the on-state sheet resistance (R_{sh}) calculated using the transfer length method (TLM) is $\sim 44.3 \pm 2.3 \text{ k}\Omega/\text{sq}$ at $V_{\text{g}} = -100 \text{ V}$ (as observed from the slopes of TLM curves in Fig. 4i of the main text). Device geometries and contact resistance, which is considered a “material-dependent” quantity do not contribute to the on-state R_{sh} value. Remarkably, our synthesis method facilitates the production of 2H-MoTe₂ with the lowest R_{sh} ($44.3 \pm 2.3 \text{ k}\Omega/\text{sq}$), the highest $I_{\text{on}}/I_{\text{off}}$ ratio (> 2.9

$\times 10^5$), and the smallest layer numbers (~ 6 layers) when compared to TLM-analyzed CVD-grown MoTe₂ FETs in previous studies^{2,4,9-11} (**Figure R1a, b**).

Further, the relationship between the on-state R_{sh} ($\sim 44.3 \pm 2.3$ k Ω /sq) obtained through TLM and the V_g -induced carrier concentration ($n_{2D} \sim 7 \times 10^{12}$ cm⁻²) permits us to determine the “intrinsic” field-effect mobility (μ_i) of the MoTe₂ using the following relationship¹²⁻¹⁴:

$$\mu_i = \frac{1}{qR_{sh}n_{2D}}$$

The calculated value of μ_i is $\sim 20.2 \pm 1.1$ cm²V⁻¹s⁻¹; this represents an inherent channel property unaffected by contact resistance. In particular, this value closely aligns with the averaged two-terminal field-effect mobility ($\mu_{th} \approx 21.0 \pm 3.3$ cm²V⁻¹s⁻¹; inset of Supplementary Fig. 14e) reported in the main text, indicating that the 2D semimetal contact electrodes have a minimal impact on μ_{th} . Notably, the TLM-extracted μ_i value in our study ($\sim 20.2 \pm 1.1$ cm²V⁻¹s⁻¹) exceeds the calculated values for CVD-grown MoTe₂ in previous studies^{2,4,9-11} (**Figure R1c**; from the reports^{2,4,9-11}, the on-state R_{sh} is extracted from the slopes of TLM plots, and n_{2D} is obtained using parallel capacitance model as $n_{2D} = C_{ox}(V_g - V_{th})/q$). Given that all the compared FETs in Refs.^{2,4,9-11} utilize a bottom-gate SiO₂ dielectric layer, the μ_i and n_{2D} values are primarily influenced by material properties such as defect density and doping capacity, rather than the device configuration or dielectric interface.

During the revision process, we discovered that the reported R_{sh} values in Mo₆Te₆- and Te-contacted 2H-MoTe₂ FETs (~ 53 and ~ 0.5 k Ω /sq in Refs.^{15,16}, respectively) are significantly underestimated due to long channel length exceeding ~ 1 mm, and the absence of proper contact pads and channel definition process. Recognizing that these data points could hinder fair benchmarking, we decided to exclude them from the benchmarking figures.

This study demonstrates that the extracted $G_{on,tot}$, R_{sh} , and μ_i values reflect the intrinsic properties of the 2D 2H-MoTe₂ channel. Furthermore, we observed that these properties exhibited superior performance compared with previous studies that reported similar characteristics.

Figure R1. Electrical characterizations of 2H-MoTe₂ FETs with vdW Au/1T'-MoTe₂ and 3D Pt contact electrodes consisting of the TLM patterns. (a) I_{on}/I_{off} ratio and R_{sh} obtained using TLM patterns in our 2H-MoTe₂ FETs, compared to previous reports^{2,4,9-11}. (b)

Comparison of on-state R_{sh} values as a function of the number of 2H-MoTe₂ layers according to literature^{2,4,9-11}. (c) “Intrinsic” field-effect mobility (μ_i) extracted using TLM measurements plotted against the layer numbers in previous reports for 2H-MoTe₂^{2,4,9-11}.

- **Summary of revisions in the manuscript or Supplementary Information:** We have provided a detailed comparison and explanation of the conductance related to the existing 2H-MoTe₂ in the introduction of the main text on lines 5-6 of page 3. We explicitly mentioned the on-state sheet conductance and implied that the corresponding comparison can be found in Supplementary Table 1. Additionally, we have revised the explanation regarding the comparison of R_{sh} using TLM on lines 8-18 of page 14, highlighting that R_{sh} exhibits material-dependent characteristics rather than being solely determined by the device. Table R1 has been inserted as Supplementary Table 1, and Figure R1 has been inserted as Supplementary Fig 24c-e. Furthermore, we have included a detailed explanation of the comparison between the provided R_{sh} and μ_i in the caption of Supplementary Fig 24. We also have mentioned the possibility of presence of a 1D defect after regrowth process in the revised manuscript on lines 17-18 of page 10. These revisions are presented below.

- **Corresponding changes in manuscript text:** (page 3, line 4-) “Moreover, the electrical conductance of 2H-MoTe₂ synthesized using these methods¹⁸⁻²⁰ has been restricted (e.g., on-state sheet conductance less than $\sim 10 \mu S$ and on-to-off current ratio of $\sim 10^4$; Supplementary Table 1), indicating a need for further exploration of novel growth techniques.”

(page 10, line 13-) “The results show that there is further potential for developing wafer-scale single-crystal films by “2H-seed” engineering using CVD process at low temperatures (550 °C or less), that is, centimeter-scale or larger grain sizes in the resulting 2D films can be achieved by producing 2H seed patterns with the same crystal orientation sufficiently separated from one another with high controllability although there might still be a 1D defect if there is a slight misalignment.”

(page 14, line 8-) “Given its n_{2D} , our 2H-MoTe₂ MSJ FET with a TLM-extracted $2R_c$ of $\sim 2.4 \pm 1.0 \text{ k}\Omega \cdot \mu\text{m}$ exhibited the lowest recorded value among the p-type 2H-MoTe₂-based FETs that exhibited significant switching behavior (i.e., $I_{on}/I_{off} > 10^3$) (Fig. 4j and Supplementary Fig. 24b). Furthermore, the on-state R_{sh} of our 2H-MoTe₂ was $\sim 44.3 \pm 2.3 \text{ k}\Omega \cdot \text{sq}^{-1}$, which is the smallest CVD-grown MoTe₂ FET analyzed using TLM in prior studies^{18,34-36,59} (Fig. 4k). This result indicates that our channel material shows a higher quality (see Supplementary Figs. 24c-e for more comparative studies) because R_{sh} represents material-dependent properties that exclude contributions from device dimensions and contact properties. Despite their low R_c , reports on FETs with poor I_{on}/I_{off} ratios or R_{sh} highlight the importance of a high-quality active layer for achieving high performance (i.e., free of partial oxides or metallic phases; their effects are discussed in Supplementary Figs. 21 and 22).”

- **Corresponding changes in SI text:** (pages 28-29) “We utilized the TLM approach to determine the on-state R_{sh} , estimated to be $\sim 44.3 \pm 2.3 \text{ k}\Omega/\text{sq}$ at $V_g = -100 \text{ V}$ (as observed from the slopes of TLM curves in Fig. 4i and Supplementary Fig. 24a). The on-state R_{sh} value represents a “material-dependent” quantity and does not incorporate

contributions from device dimensions and contact resistance. Our analysis of on-state R_{sh} of 2H-MoTe₂ indicates that our synthesis method results in the lowest R_{sh} (44.3 ± 2.3 k Ω /sq), the highest I_{on}/I_{off} ratio ($> 2.9 \times 10^5$), and the smallest layer numbers (~ 6 layers), compared to those of TLM-analyzed CVD-grown MoTe₂ FETs in previous studies^{3,5,24,25,27} (Supplementary Fig. 24c, d).

Furthermore, the relationship between the on-state R_{sh} ($\sim 44.3 \pm 2.3$ k Ω /sq) obtained through TLM and the V_g -induced carrier concentration ($n_{2D} \sim 7 \times 10^{12}$ cm⁻²) permits us to determine the “intrinsic” field-effect mobility (μ_i) of our MoTe₂ using the following relationship⁶⁵⁻⁶⁷:

$$\mu_i = \frac{1}{qR_{sh}n_{2D}}$$

The calculated value of μ_i is $\sim 20.2 \pm 1.1$ cm²V⁻¹s⁻¹, representing an inherent channel property that is unaffected by contact resistance. Notably, this value closely matches the averaged two-terminal field-effect mobility ($\mu_h \approx 21.0 \pm 3.3$ cm²V⁻¹s⁻¹; inset of Supplementary Fig. 14e), indicating that our 2D semimetal contact electrodes have a minimal impact on μ_h . Moreover, the TLM-extracted μ_i value in our study ($\sim 20.2 \pm 1.1$ cm²V⁻¹s⁻¹) surpasses the calculated values for CVD-grown MoTe₂ in previous studies^{3,5,24,25,27} (Supplementary Fig. 24e; from the reports^{3,5,24,25,27}, the on-state R_{sh} value is extracted from the slopes of TLM plot, and n_{2D} is obtained using a parallel capacitance model as $n_{2D} = C_{ox}(V_g - V_{th})/q$). Given that all the compared FETs in Refs.^{3,5,24,25,27} utilize a bottom-gate SiO₂ dielectric layer, the μ_i or n_{2D} values are primarily influenced by material properties such as defect density and doping capacity, rather than the device configuration or dielectric interface.”

Reviewer #4's Comments (blue) Followed by Our Response (black):

I thank the authors for their careful revisions and removed over-the-top claims such as the extremely high on-current at the 5 nm channel limit. However, after scrutiny of the revision and response letter, I still find that the manuscript lacks the novelty to be published in a well-known journal like Nature Communications, and most of my previous concerns have not been well addressed.

Response to general comment: We are grateful for your response to this manuscript. With respect to novelty, we request a comprehensive evaluation of the distinct aspects of this study.

Although we acknowledge that few details in our study may not be exceptionally novel, we would like to emphasize the following achievements: (i) a synthesis method utilizing a Te-confined reactor for the production of high-quality 2H-MoTe₂, (ii) the formation of single-crystal arrays through abnormal grain growth, (iii) a detailed analysis of junction interface characteristics, and (iv) the utilization of Fermi-level tuned semimetals for contact electrodes. Combinations of these techniques, in addition to achieving exceptional high-quality 2H-MoTe₂ film and contact interfaces, have facilitated the realization of significantly superior *p*-type conductivity of 2H-MoTe₂, even without extremely short channel lengths or high-*k* dielectric gates. We have thoroughly reviewed your feedback and sincerely hope that our responses adequately address your concerns.

1. For the 4-inch wafer sample, which was grown at 700 °C, the nucleation and growth kinetics resulted in polycrystalline thin films with grain sizes comparable to those previously reported. Moreover, this method has been used to synthesize 2H-MoTe₂ in the authors' previous paper (Nature Electronics 3, 207-215, 2020). The so-called "abnormal grain growth" is the same mechanism as the "solid-to-solid phase transition", and the atomic rearrangement and recrystallization processes have been studied.

Response to Comment #1: Thank you for bringing up your concerns. While the size of abnormally grown 2H-phase MoTe₂ domains in this study may be similar to those in other studies, no study has homogeneously and uniformly formed them on a 4-inch wafer (~300–1,000 μm in Supplementary Fig. 2). This is significant because a uniform 2H nucleation density on a large-area substrate can only be achieved using our Te-confined reactor (Supplementary Fig. 11).

Furthermore, while the polycrystalline WTe₂ grain size in our previous study (Nature Electronics 3, 207-215, 2020) was restricted to hundreds of nanometers, the domain size of 2H-MoTe₂ in this study expanded to several hundred micrometers through the abnormal grain growth mechanism under the Te-rich atmosphere; thus, the focus of each study is different.

Regarding abnormal grain growth, we revealed that the growth kinetics can vary depending on the Te flux and the presence of H₂ (Fig. 3e and Supplementary Figs. 12b-e). In addition, we propose the possibility of seed growth at ~500 °C without a capping layer by utilizing the principles of abnormal grain growth (Fig. 3). This strategy further enables the conformal

growth of 2H-MoTe₂ single crystals arrays with the same in-plane crystal orientation (Supplementary Fig. 13).

2. Low-temperature growth at 500 °C was used only for seed-induced growth. However, seed arrays start with thin films grown at 700 °C.

(i) Single-crystal arrays are limited by the size of the single-crystal domains grown at 700 °C.

Response to Comment #2 and 2-(i): We appreciate the concerns suggested by the reviewer, but we believe that this approach offers multiple possibilities for high-quality 2H MoTe₂ and other 2D crystal synthesis. The seed layers can be used as single crystals regardless of their growth temperature. For example, the seed arrays shown in Fig. 3 are synthesized at 700 °C, whereas those used in Supplementary Fig. 13 are single-crystalline flakes exfoliated from bulk crystal.

We also note that single-crystal arrays cannot be limited by seed size. In principle, the subsequent growth of 2H MoTe₂ can expand beyond the size of the seeds. In addition, our seed growth method utilizes standard lithography, which allows the production of smaller patterns.

(ii) How did the authors control the nucleation sites to achieve ~5 mm-sized single crystal thin films as seed films? From all images presented in the manuscript and supplemental information, there are multiple nuclei on a chip-sized substrate.

Response to Comment #2-(ii): Thank you for the question. The “bulk” MoTe₂ mother crystal (purchased from HQ graphene) was a single crystal over a large area (~5 mm). Therefore, the flakes exfoliated on the tape had the same crystal orientation when the adhesive tape was applied and removed from the mother crystal.

- **Revision in Supplementary Information:** We have modified lines in the caption of Supplementary Fig. 13. This revision is presented as: “The “*bulk*” MoTe₂ mother crystal (obtained from *HQ graphene*) had a single crystalline property over a large area (> ~5 mm).”

(iii) If we already have a single crystal thin film of large size, what is the purpose to etch it into an array as a seed film, perform secondary growth, etch away the unwanted area (most likely the seed area because they are thicker), leaving a discontinuous film? In other words, is not the original 100% coverage seed film better than the array film that has undergone a complex process?

Response to Comment #2 (iii): Thank you for the question. As the reviewer suggested, it would be appropriate if uniform-thickness single crystals could maintain high quality and be synthesized from a single nucleus. However, existing research on the 2H-phase MoTe₂ is limited, which indicates the importance of our synthesis strategy.

3. I still think the comparison between production rate and growth rate is unfair. The production rate defined by the authors is the size of the substrate divided by the time required to achieve 100% 2H-MoTe₂ coverage. The growth rate is defined by the 2H-MoTe₂ domain expansion rate. From all the optical images presented in the manuscript and supplementary information, the method presented in this manuscript has similar growth rates to the previous reports. Furthermore, previously reported results would also yield a similar production rate for multiple nucleation, since it depends on the growth substrate size.

Response to Comment #3 (iii): Thank you for your concerns; however, we believe that our argument regarding the production rate is sufficiently cogent. This is because no previous studies have demonstrated the uniform nucleation and grain growth of 2H-MoTe₂ on a 4-inch substrate. Wafer-scale uniformity is inevitably compromised depending on the flow of the precursor in the CVD reactor (Supplementary Fig. 11). Consequently, the formation of phase-pure 2H-MoTe₂ over a large area is difficult; for example, powder-based horizontal CVD results in non-uniform nucleation of the 2H-phase across a large area (Supplementary Figs. 11d-i).

In addition, the intricate comparison of growth rates across different studies is challenging. For instance, the growth rate varies with the growth temperature, and the ability to observe the growth rate over time depends on the initial nucleation density. Moreover, few studies have reported these specific details. Because the growth temperatures were different, quantifying and comparing them under the same conditions was challenging. In contrast, the production rate can be compared among different studies as it is determined by the time required to obtain fully-grown 2H-MoTe₂ films on predetermined substrate sizes.

4. On-current and on-off ratio are device parameters and should be compared under the same device geometry and measurement conditions. Mobility is a more intrinsic property related to film quality (should be Hall mobility, since field effect mobility is still a device parameter). Under the same device geometry, carrier concentration, and applied electrical field, the higher the mobility, the larger the on-current. Although the comparison between the calculated on-current at 0 channel length and the experimental results with finite channel length is unfair, 2H-MoTe₂ still provides a much smaller on-current (Table R2).

Response to Comment #4: We appreciate the concerns expressed by the reviewer. Nonetheless, we would like to underscore that the obtained I_{on} and I_{on}/I_{off} ratio of our 2H-MoTe₂ FET are higher than those of any other 2H-MoTe₂ FETs reported in previous studies^{1,2,4,5,9-11,17-25}, specifically considering its channel length dependence (Fig. 4g, h and **Table R2**). The correlation between the carrier concentration (n_{2D}) and I_{on} further demonstrates the high performance of our FETs (Supplementary Figure 18c). In particular, this study did not utilize any specific device configurations that overestimate the performance indicators. For example, our FET device operation relies on the bottom gate of a low- k SiO₂ (300 nm) dielectric layer. Nearly every study listed in **Table R2** (yellow highlights) used bottom-gate SiO₂ in the same manner, except for the case of a high- k bottom gate^{1,17} or BN top gate²², which dictates that the effect of gate capacitance or MoTe₂/dielectric interface is comparable. In

addition, I_{on}/I_{off} ratio is independent of the device geometry unless it has an ultrashort channel length below 100 nm, which is not the case in the comparison presented in Table R2.

Table R2. Benchmarking table for the few-layered 2H-MoTe₂-based FETs. To illustrate the reproducibility of the data points, we demonstrated the average values of the tested devices in each report. The best result or the data point extracted solely by one-time measurement of each reference is presented in parentheses. In Ref.¹⁸, the as-fabricated devices were gently annealed in UHV to improve their contact properties. This may lead to a different contact interface compared with those in other studies that did not perform heat treatment following device fabrication. The R_c values for 1T'-MoTe₂ contacts^{2,5,10,11} presented here include those of 3D-metal-pads/1T'-MoTe₂ and 1T'/2H-MoTe₂ interfaces. The bottom gate SiO₂ is commonly used in previous studies^{1,2,4,5,9-11,17-25}, with the exception of cases involving a high-k bottom gate^{1,17} or BN top gate²².

Prep. method for 2H-MoTe ₂	Bottom gate dielectric	Contact metal	R_c extraction method	R_c (k Ω · μ m)	R_{sh} (k Ω /sq)	SBH (meV)	I_{on}/I_{off} ratio at RT	μ_h at RT (cm ² V ⁻¹ s ⁻¹)	I_{on} at $V_{ds} = -1$ V (nA/ μ m)	Ref.	
CVD	300 nm SiO ₂	1T'	TLM	1.2 \pm 0.5	44.3 \pm 2.3	27.4 (14)	1.3 \times 10 ⁵ (2.9 \times 10 ⁵)	21.0 \pm 3.3 (29.5)	7,820 \pm 1,405 (8,814)	This study	
CVD	90 nm SiO ₂	Ti/Au	N/A	N/A	N/A	N/A	(10 ³)	1	(40 at $V_d = 0.5$ V)	6	
	300 nm SiO ₂	Ti/Au	TLM	15,600 \pm 580	4,670 \pm 700	132	10-20	0.6-0.8	5.72 at $V_d = -0.1$ V	9	
		1T' (edge)	N/A	N/A	N/A	30 \pm 10	20	7-8	39.6 at $V_d = -0.1$ V		
	300 nm SiO ₂	1T' (vertical; directly synthesized)	TLM	36.4	(125)	N/A	(10 ³)	15 (25)	~1,000 at $V_d = 0.5$ V	10	
	35 nm Al ₂ O ₃	Cr/Au	N/A	N/A	N/A	N/A	(10 ³)	(0.44)	1.11	17	
	12 nm HfO ₂	Pd	N/A	N/A	N/A	N/A	(10 ³)	2	N/A	1	
	285 nm SiO ₂	1T' (edge)	N/A	N/A	N/A	N/A	250 (DFT)	N/A	N/A	N/A	4
		Ti/Au	TLM	326,500	35,200	N/A	(126)	(0.5)	N/A		
	300 nm SiO ₂	Au	Four-point probe	(409)	N/A	150	(10 ⁵)	(4.0)	N/A	11	
1T' (edge)		TLM (irregular line shape)	14	260	22	10 ⁵	16.2	N/A			
300 nm SiO ₂	Pd/Au	N/A	N/A	N/A	N/A	(10 ³)	(2)	N/A	2		

		1T' (edge)	TLM	1.7	100	N/A	(10^4)	20-24	N/A	
	SiO ₂	1T' (edge)	TLM	1.6	N/A	65	1.8×10^4	45 ± 2	94 at $V_d = 0.1$ V	5
MOCVD	300 nm SiO ₂	Pd	N/A	N/A	N/A	N/A	(4.8×10^4)	(0.029)	N/A	3
	SiO ₂	Mo ₆ Te ₆ (directly synthesized)	TLM	28,500	53	8.7	(10^3)	(1,139)	N/A	15
	300 nm SiO ₂	Te (directly synthesized)	TLM	12.3	0.5	N/A	(6×10^2)	(544)	N/A	16
Mechanical exfoliation	285 nm SiO ₂	Cr	Four-point probe	(10,000)	N/A	(230)	(10^3)	N/A	N/A	26
	300 nm SiO ₂	Ag/Au	N/A	N/A	N/A	N/A	(10^2)	(1.13)	(53)	19
	285 nm SiO ₂	Ti/Au	N/A	N/A	N/A	N/A	(2×10^3)	(0.3)	N/A	20
	285 nm SiO ₂	Ti/Au	N/A	N/A	N/A	N/A	10^2	0.063	N/A	21
	270 nm SiO ₂	Ti/Au (Annealed under UHV)	N/A	N/A	N/A	N/A	10^5	20	(7 at $V_d = 0.01$ V)	18
	100 nm SiO ₂	Pd/Ti/Au	N/A	N/A	N/A	(100)	(10^3)	N/A	(1,000 at $V_d = -3$ V)	27
	Top gate of 10-20 nm BN	Pt (hBN-capped)	Four-point probe	1,000	N/A	N/A	(10^4)	(18)	(285)	22
	SiO ₂	Ti	N/A	N/A	N/A	42	$(10$ at $T = 40$ K)	N/A	N/A	28
		Cr	N/A	N/A	N/A	41	$(10$ at $T = 40$ K)	N/A	N/A	
		Au	N/A	N/A	N/A	30	$(10$ at $T = 40$ K)	N/A	N/A	
		Pd	N/A	N/A	N/A	10	$(10$ at $T = 40$ K)	N/A	N/A	
	300 nm SiO ₂	Pd/Au	N/A	N/A	N/A	(90)	(10^5)	(7.0)	N/A	23
		Cr/Au	N/A	N/A	N/A	300	(10^3)	(1.2)	N/A	
300 nm SiO ₂	Cr	N/A	N/A	N/A	100	2.3×10^4 (2.5×10^4)	1.0 (1.5)	N/A	24	
	Au	N/A	N/A	N/A	75	1.5×10^4 (5.7×10^4)	0.5 (1.2)	N/A		

In addition, we would like to note that the excellent performance of our 2H-MoTe₂ FET can be independent of specific geometry, configurations, and contact resistance of the devices. This is confirmed by measuring the on-state sheet resistance (R_{sh}). In our manuscript, the on-

state R_{sh} value calculated using the TLM is $\sim 44.3 \pm 2.3 \text{ k}\Omega/\text{sq}$ at $V_g = -100 \text{ V}$ (as observed from the slopes of TLM curves in Fig. 4i of the main text). Device geometries and contact resistance, which is considered a “material-dependent” quantity do not contribute to the on-state R_{sh} value. Remarkably, the analysis of on-state R_{sh} of 2H-MoTe₂ indicates that our synthesis methods facilitate the lowest R_{sh} ($44.3 \pm 2.3 \text{ k}\Omega/\text{sq}$), the highest I_{on}/I_{off} ratio ($> 2.9 \times 10^5$), and the smallest layer numbers (~ 6 layers) when compared to TLM-analyzed CVD-grown MoTe₂ FETs in previous studies^{2,4,9-11} (**Figure R1a, b; copied below**).

Furthermore, the relationship between the on-state R_{sh} ($\sim 44.3 \pm 2.3 \text{ k}\Omega/\text{sq}$) obtained through TLM and the V_g -induced carrier concentration ($n_{2D} \sim 7 \times 10^{12} \text{ cm}^{-2}$) permits us to determine the “intrinsic” field-effect mobility (μ_i) of the MoTe₂ using the following relationship¹²⁻¹⁴:

$$\mu_i = \frac{1}{qR_{sh}n_{2D}}$$

Using this equation, we calculated μ_i to be $\sim 20.2 \pm 1.1 \text{ cm}^2\text{V}^{-1}\text{s}^{-1}$; this represents an inherent channel property unaffected by contact resistance. In particular, this value closely matches with the averaged two-terminal field-effect mobility ($\mu_{th} \approx 21.0 \pm 3.3 \text{ cm}^2\text{V}^{-1}\text{s}^{-1}$; inset of Supplementary Fig. 14e) reported in the main text, indicating that the 2D semimetal contact electrodes have minimal impact on μ_{th} . Notably, the TLM-extracted μ_i value in our study ($\sim 20.2 \pm 1.1 \text{ cm}^2\text{V}^{-1}\text{s}^{-1}$) exceeds the calculated values for CVD-grown MoTe₂ in previous studies^{2,4,9-11} (**Figure R1c**; from the reports^{2,4,9-11}, the on-state R_{sh} is extracted from the slopes of TLM plots, and n_{2D} is obtained using parallel capacitance model as $n_{2D} = C_{ox}(V_g - V_{th})/q$).

Figure R1 (copied from above). Electrical characterizations of 2H-MoTe₂ FETs with vdW Au/1T'-MoTe₂ and 3D Pt contact electrodes consisting of the TLM patterns. (a) I_{on}/I_{off} ratio and R_{sh} obtained using TLM patterns in our 2H-MoTe₂ FETs, compared to previous reports^{2,4,9-11}. (b) Comparison of on-state R_{sh} values as a function of the number of 2H-MoTe₂ layers according to literature^{2,4,9-11}. (c) “Intrinsic” field-effect mobility (μ_i) extracted using TLM measurements plotted against the layer numbers in previous reports for 2H-MoTe₂^{2,4,9-11}.

- **Summary of revisions in the manuscript or Supplementary Information:** We have revised the explanation regarding the comparison of R_{sh} using TLM on lines 8-18 of page 14 in the main text, highlighting that R_{sh} exhibits material-dependent characteristics rather than being solely determined by the device. Table R2 has been

inserted as Supplementary Table 3, and Figure R1 has been inserted as Supplementary Fig 24c-e. Furthermore, we have included a detailed explanation of the comparison between the provided R_{sh} and μ_i in the caption of Supplementary Fig. 24. These revisions are presented below.

- Corresponding changes in manuscript text:** (page 3, line 4-) “Moreover, the electrical conductance of 2H-MoTe₂ synthesized using these methods¹⁸⁻²⁰ has been restricted (e.g., on-state sheet conductance less than $\sim 10 \mu S$ and on-to-off current ratio of $\sim 10^4$; Supplementary Table 1), indicating a need for further exploration of novel growth techniques.”

(page 14, line 8-) “Given its n_{2D} , our 2H-MoTe₂ MSJ FET with a TLM-extracted $2R_c$ of $\sim 2.4 \pm 1.0 \text{ k}\Omega \cdot \mu\text{m}$ exhibited the lowest recorded value among the p-type 2H-MoTe₂-based FETs that exhibited significant switching behavior (i.e., $I_{on}/I_{off} > 10^3$) (Fig. 4j and Supplementary Fig. 24b). Furthermore, the on-state R_{sh} of our 2H-MoTe₂ was $\sim 44.3 \pm 2.3 \text{ k}\Omega \cdot \text{sq}^{-1}$, which is the smallest CVD-grown MoTe₂ FET analyzed using TLM in prior studies^{18,34-36,59} (Fig. 4k). This result indicates that our channel material shows a higher quality (see Supplementary Figs. 24c-e for more comparative studies) because R_{sh} represents material-dependent properties that exclude contributions from device dimensions and contact properties. Despite their low R_c , reports on FETs with poor I_{on}/I_{off} ratios or R_{sh} highlight the importance of a high-quality active layer for achieving high performance (i.e., free of partial oxides or metallic phases; their effects are discussed in Supplementary Figs. 21 and 22).”
- Corresponding changes in SI text:** (pages 28-29) “We utilized the TLM approach to determine the on-state R_{sh} , estimated to be $\sim 44.3 \pm 2.3 \text{ k}\Omega/\text{sq}$ at $V_g = -100 \text{ V}$ (as observed from the slopes of TLM curves in Fig. 4i and Supplementary Fig. 24a). The on-state R_{sh} value represents a “material-dependent” quantity and does not incorporate contributions from device dimensions and contact resistance. Our analysis of on-state R_{sh} of 2H-MoTe₂ indicates that our synthesis method results in the lowest R_{sh} ($44.3 \pm 2.3 \text{ k}\Omega/\text{sq}$), the highest I_{on}/I_{off} ratio ($> 2.9 \times 10^5$), and the smallest layer numbers (~ 6 layers), compared to those of TLM-analyzed CVD-grown MoTe₂ FETs in previous studies^{3,5,24,25,27} (Supplementary Fig. 24c, d).

Furthermore, the relationship between the on-state R_{sh} ($\sim 44.3 \pm 2.3 \text{ k}\Omega/\text{sq}$) obtained through TLM and the V_g -induced carrier concentration ($n_{2D} \sim 7 \times 10^{12} \text{ cm}^{-2}$) permits us to determine the “intrinsic” field-effect mobility (μ_i) of our MoTe₂ using the following relationship⁶⁵⁻⁶⁷:

$$\mu_i = \frac{1}{qR_{sh}n_{2D}}$$

The calculated value of μ_i is $\sim 20.2 \pm 1.1 \text{ cm}^2\text{V}^{-1}\text{s}^{-1}$, representing an inherent channel property that is unaffected by contact resistance. Notably, this value closely matches the averaged two-terminal field-effect mobility ($\mu_h \approx 21.0 \pm 3.3 \text{ cm}^2\text{V}^{-1}\text{s}^{-1}$; inset of Supplementary Fig. 14e), indicating that our 2D semimetal contact electrodes have a minimal impact on μ_h . Moreover, the TLM-extracted μ_i value in our study ($\sim 20.2 \pm 1.1 \text{ cm}^2\text{V}^{-1}\text{s}^{-1}$) surpasses the calculated values for CVD-grown MoTe₂ in previous studies^{3,5,24,25,27} (Supplementary Fig. 24e; from the reports^{3,5,24,25,27}, the on-state R_{sh} value is extracted from the slopes of TLM plot, and n_{2D} is obtained using a parallel capacitance model as $n_{2D} = C_{ox}(V_g - V_{th})/q$). Given that all the compared FETs in Refs.^{3,5,24,25,27} utilize a bottom-gate SiO₂ dielectric layer, the μ_i or n_{2D} values are primarily influenced by

material properties such as defect density and doping capacity, rather than the device configuration or dielectric interface.”

5. In Fig. 25a, the on-current does not “increase significantly as the T decreased from ~ 338 K to ~ 218 K”. Furthermore, it is inappropriate to study the electron-phonon scattering using field-effect mobility, since it is largely affected by contact, and Hall mobility should be used instead.

Response to Comment #5: Thank you for your concerns. Supplementary Fig. 25a shows a transfer curve graph on a logarithmic scale, which makes it difficult to recognize the significant on-current change. However, the corresponding linear scale transfer curve (Fig. 5d and Supplementary Fig. 25h) effectively demonstrates the changes in I_{ds} depending on T .

Investigation of the two-terminal field-effect mobility over T serves as a valuable tool for investigating phonon- or contact-limited scattering in the channel because this approach is applicable to 2D FETs with “ultrasmall” contact resistance, which is primarily observed in the n -type 2D FETs used in previous studies²⁹⁻³⁷. When the two-terminal mobility remained largely unaffected by the contact interface, it revealed the intrinsic properties of the channel, which were predominantly influenced by phonon scattering. We have provided evidence for this by presenting the results in Supplementary Fig. 25e that demonstrate increased mobility at lower T for our 2H-MoTe₂ FETs with Fermi-level-tuned semimetal electrodes. However, when there is substantial contact resistance (e.g., the Ti or Pt contacts depicted in Supplementary Fig. 25e), the mobility decreases. Therefore, the emergence of the band-like transport in our FETs suggests that the influence of the contact resistance and Schottky barrier on the transport is negligible. To the best of our knowledge, this is the first demonstration of phonon-limited behavior in two-terminal FETs with p -type 2D MoTe₂.

- **Summary of revisions in the manuscript or Supplementary Information:** We have clearly indicated which figures show how conductivity changes with temperature, on page 15, lines 17, 18, and 22 in the main text. To eliminate any potential misleading that MIT can be interpreted as “material-dependent” property, we have removed the column for the comparisons of MIT within Supplementary Table 1. Consequently, we have also removed the sentence referring to MIT in line 2 of page 16 in the main text.

References used in this Response.

- 1 Zhang, Q. *et al.* Simultaneous synthesis and integration of two-dimensional electronic components. *Nature Electronics* **2**, 164-170 (2019). <https://doi.org:10.1038/s41928-019-0233-2>
- 2 Xu, X. *et al.* Scaling-up Atomically Thin Coplanar Semiconductor–Metal Circuitry via Phase Engineered Chemical Assembly. *Nano Letters* **19**, 6845-6852 (2019). <https://doi.org:10.1021/acs.nanolett.9b02006>
- 3 Kim, T. *et al.* Wafer-Scale Epitaxial 1T', 1T'–2H Mixed, and 2H Phases MoTe₂ Thin Films Grown by Metal–Organic Chemical Vapor Deposition. *Advanced Materials Interfaces* **5**, 1800439 (2018).

- 4 Zhang, X. *et al.* Low Contact Barrier in 2H/1T' MoTe₂ In-Plane Heterostructure Synthesized by Chemical Vapor Deposition. *ACS Applied Materials & Interfaces* **11**, 12777-12785 (2019). <https://doi.org:10.1021/acsami.9b00306>
- 5 Xu, X. *et al.* Seeded 2D epitaxy of large-area single-crystal films of the van der Waals semiconductor 2H MoTe₂. *Science* **372**, 195-200 (2021). <https://doi.org:doi:10.1126/science.abf5825>
- 6 Zhou, L. *et al.* Large-Area Synthesis of High-Quality Uniform Few-Layer MoTe₂. *Journal of the American Chemical Society* **137**, 11892-11895 (2015). <https://doi.org:10.1021/jacs.5b07452>
- 7 Yang, L. *et al.* Tellurization Velocity-Dependent Metallic–Semiconducting–Metallic Phase Evolution in Chemical Vapor Deposition Growth of Large-Area, Few-Layer MoTe₂. *ACS Nano* **11**, 1964-1972 (2017). <https://doi.org:10.1021/acsnano.6b08109>
- 8 Xu, X. *et al.* Millimeter-Scale Single-Crystalline Semiconducting MoTe₂ via Solid-to-Solid Phase Transformation. *Journal of the American Chemical Society* **141**, 2128-2134 (2019). <https://doi.org:10.1021/jacs.8b12230>
- 9 Ma, R. *et al.* MoTe₂ Lateral Homojunction Field-Effect Transistors Fabricated using Flux-Controlled Phase Engineering. *ACS Nano* **13**, 8035-8046 (2019). <https://doi.org:10.1021/acsnano.9b02785>
- 10 Yang, S. *et al.* Large-Scale Vertical 1T'/2H MoTe₂ Nanosheet-Based Heterostructures for Low Contact Resistance Transistors. *ACS Applied Nano Materials* **3**, 10411-10417 (2020). <https://doi.org:10.1021/acsanm.0c02302>
- 11 Sung, J. H. *et al.* Coplanar semiconductor–metal circuitry defined on few-layer MoTe₂ via polymorphic heteroepitaxy. *Nature nanotechnology* **12**, 1064 (2017).
- 12 Schranghamer, T. F. *et al.* Ultrascaled Contacts to Monolayer MoS₂ Field Effect Transistors. *Nano Letters* **23**, 3426-3434 (2023). <https://doi.org:10.1021/acs.nanolett.3c00466>
- 13 English, C. D., Shine, G., Dorgan, V. E., Saraswat, K. C. & Pop, E. Improved Contacts to MoS₂ Transistors by Ultra-High Vacuum Metal Deposition. *Nano Letters* **16**, 3824-3830 (2016). <https://doi.org:10.1021/acs.nanolett.6b01309>
- 14 Mleczko, M. J. *et al.* Contact Engineering High-Performance n-Type MoTe₂ Transistors. *Nano Letters* **19**, 6352-6362 (2019). <https://doi.org:10.1021/acs.nanolett.9b02497>
- 15 Lee, R. S. *et al.* van der Waals Epitaxy of High-Mobility Polymorphic Structure of Mo₆Te₆ Nanoplates/MoTe₂ Atomic Layers with Low Schottky Barrier Height. *ACS Nano* **13**, 642-648 (2019). <https://doi.org:10.1021/acsnano.8b07720>
- 16 Choi, D. *et al.* Directly grown Te nanowire electrodes and soft plasma etching for high-performance MoTe₂ field-effect transistors. *Applied Surface Science* **565**, 150521 (2021). <https://doi.org:https://doi.org/10.1016/j.apsusc.2021.150521>
- 17 Park, Y. J., Katiyar, A. K., Hoang, A. T. & Ahn, J.-H. Controllable P- and N-Type Conversion of MoTe₂ via Oxide Interfacial Layer for Logic Circuits. *Small* **15**, 1901772 (2019). <https://doi.org:10.1002/smll.201901772>

- 18 Pradhan, N. R. *et al.* Field-effect transistors based on few-layered α -MoTe₂. *ACS nano* **8**, 5911-5920 (2014).
- 19 Zheng, X. *et al.* Enormous enhancement in electrical performance of few-layered MoTe₂ due to Schottky barrier reduction induced by ultraviolet ozone treatment. *Nano Research*, 1-7 (2020).
- 20 Lin, Y.-F. *et al.* Ambipolar MoTe₂ Transistors and Their Applications in Logic Circuits. *Advanced Materials* **26**, 3263-3269 (2014). <https://doi.org:10.1002/adma.201305845>
- 21 Chang, Y.-M. *et al.* Reversible and Precisely Controllable p/n-Type Doping of MoTe₂ Transistors through Electrothermal Doping. *Advanced Materials* **30**, 1706995 (2018). <https://doi.org:10.1002/adma.201706995>
- 22 Larentis, S. *et al.* Reconfigurable Complementary Monolayer MoTe₂ Field-Effect Transistors for Integrated Circuits. *ACS Nano* **11**, 4832-4839 (2017). <https://doi.org:10.1021/acsnano.7b01306>
- 23 Aftab, S. *et al.* Formation of an MoTe₂ based Schottky junction employing ultra-low and high resistive metal contacts. *RSC Advances* **9**, 10017-10023 (2019). <https://doi.org:10.1039/C8RA09656B>
- 24 Yin, L. *et al.* Ultrahigh sensitive MoTe₂ phototransistors driven by carrier tunneling. *Applied Physics Letters* **108**, 043503 (2016). <https://doi.org:10.1063/1.4941001>
- 25 Aftab, S. *et al.* Formation of an MoTe₂ based Schottky junction employing ultra-low and high resistive metal contacts. *RSC advances* **9**, 10017-10023 (2019).
- 26 Kim, C. *et al.* Fermi Level Pinning at Electrical Metal Contacts of Monolayer Molybdenum Dichalcogenides. *ACS Nano* **11**, 1588-1596 (2017). <https://doi.org:10.1021/acsnano.6b07159>
- 27 Haratipour, N. & Koester, S. Multi-layer MoTe₂ p-channel MOSFETs with high drive current. *72nd Device Research Conference*, 171-172 (2014).
- 28 Townsend, N. J., Amit, I., Craciun, M. F. & Russo, S. Sub 20 meV Schottky barriers in metal/MoTe₂ junctions. *2D Materials* **5**, 025023 (2018). <https://doi.org:10.1088/2053-1583/aab56a>
- 29 Wang, Y. *et al.* Van der Waals contacts between three-dimensional metals and two-dimensional semiconductors. *Nature* **568**, 70-74 (2019). <https://doi.org:10.1038/s41586-019-1052-3>
- 30 Jariwala, D. *et al.* Band-like transport in high mobility unencapsulated single-layer MoS₂ transistors. *Applied Physics Letters* **102** (2013). <https://doi.org:10.1063/1.4803920>
- 31 Chuang, H.-J. *et al.* High Mobility WSe₂ p- and n-Type Field-Effect Transistors Contacted by Highly Doped Graphene for Low-Resistance Contacts. *Nano Letters* **14**, 3594-3601 (2014). <https://doi.org:10.1021/nl501275p>
- 32 Wang, J. *et al.* High Mobility MoS₂ Transistor with Low Schottky Barrier Contact by Using Atomic Thick h-BN as a Tunneling Layer. *Advanced Materials* **28**, 8302-8308 (2016). <https://doi.org:https://doi.org/10.1002/adma.201602757>

- 33 Chuang, H.-J. *et al.* Low-Resistance 2D/2D Ohmic Contacts: A Universal Approach to High-Performance WSe₂, MoS₂, and MoSe₂ Transistors. *Nano Letters* **16**, 1896-1902 (2016). <https://doi.org:10.1021/acs.nanolett.5b05066>
- 34 Andrews, K., Bowman, A., Rijal, U., Chen, P.-Y. & Zhou, Z. Improved Contacts and Device Performance in MoS₂ Transistors Using a 2D Semiconductor Interlayer. *ACS Nano* **14**, 6232-6241 (2020). <https://doi.org:10.1021/acsnano.0c02303>
- 35 Liu, Y. *et al.* Toward Barrier Free Contact to Molybdenum Disulfide Using Graphene Electrodes. *Nano Letters* **15**, 3030-3034 (2015). <https://doi.org:10.1021/nl504957p>
- 36 Xiao, J. *et al.* Record-high saturation current in end-bond contacted monolayer MoS₂ transistors. *Nano Research* **15**, 475-481 (2022). <https://doi.org:10.1007/s12274-021-3504-y>
- 37 Miao, J. *et al.* A “Click” Reaction to Engineer MoS₂ Field-Effect Transistors with Low Contact Resistance. *ACS Nano* **16**, 20647-20655 (2022). <https://doi.org:10.1021/acsnano.2c07670>